# MISA: Memory-Efficient LLMs Optimization with Module-wise Importance Sampling

**Yuxi Liu**[*]
Peking University
yuxiliu666@stu.pku.edu.cn

**Renjia Deng**[*]
Peking University
2501210078@stu.pku.edu.cn

**Yutong He**
Peking University
yutonghe@pku.edu.cn

**Xue Wang**
Alibaba DAMO Academy
xue.wang@alibaba-inc.com

**Tao Yao**
Shanghai Jiao Tong University
taoyao@sjtu.edu.cn

**Kun Yuan**[†]
Peking University
kunyuan@pku.edu.cn

## Abstract

The substantial memory demands of pre-training and fine-tuning large language models (LLMs) require memory-efficient optimization algorithms. One promising approach is layer-wise optimization, which treats each transformer block as a single layer and optimizes it sequentially, while freezing the other layers to save optimizer states and activations. Although effective, these methods ignore the varying importance of the modules within each layer, leading to suboptimal performance. Moreover, layer-wise sampling provides only limited memory savings, as at least one full layer must remain active during optimization. To overcome these limitations, we propose **M**odule-wise **I**mportance **SA**mpling (**MISA**), a novel method that divides each layer into smaller modules and assigns importance scores to each module. MISA uses a weighted random sampling mechanism to activate modules, provably reducing gradient variance compared to layer-wise sampling. Additionally, we establish an $\mathcal{O}(1/\sqrt{K})$ convergence rate under non-convex and stochastic conditions, where $K$ is the total number of block updates, and provide a detailed memory analysis showcasing MISA's superiority over existing baseline methods. Experiments on diverse learning tasks validate the effectiveness of MISA. Source code is available at: https://github.com/pkumelon/MISA.

## 1 Introduction

Large language models (LLMs) have emerged as a cornerstone of modern artificial intelligence, driving groundbreaking advancements in diverse domains such as machine translation, commonsense reasoning, and mathematical problem-solving. Despite their impressive capabilities, fine-tuning these models for downstream tasks introduces significant challenges [45, 46, 49], primarily due to the substantial memory overhead required to store optimizer states, gradients, and intermediate activations. These memory footprints are particularly acute in resource-constrained environments, where the full-parameter fine-tuning of billion-scale models often exceeds the available hardware capacity, thereby impeding the practical deployment and scalability of LLMs [69, 60, 14].

---

[*]Equal contribution.
[†]Corresponding author.

39th Conference on Neural Information Processing Systems (NeurIPS 2025).

To address these challenges, parameter-efficient fine-tuning (PEFT) methods, such as Low-Rank Adaptation (LoRA) [24] and its variants [35, 12], have gained considerable attention. These methods freeze the majority of pre-trained parameters and optimize only small, low-rank matrices integrated into transformer layers, leading to substantial reductions in memory usage. However, while these approaches are highly memory-efficient, they inherently constrain the model's adaptability. By optimizing only a sparse subset of parameters, such methods often lead to suboptimal performance compared to full-parameter fine-tuning, as task-specific features crucial to downstream performance may reside within the frozen portions of the network [33, 66].

Recent research has explored block-coordinate descent (BCD) [55, 59, 62], a classical strategy for high-dimensional optimization, as a promising alternative for fine-tuning LLMs, resulting in the layer-wise optimization methods such as BAdam [37], HIFT [36], LIFT [75], LISA [41], and BlockLLM [47]. Unlike LoRA, these layer-wise methods optimize transformer blocks sequentially while keeping the remaining layers frozen. By iteratively updating all transformer blocks, layer-wise optimization effectively enables full-parameter updates, thereby preserving the expressive capacity of the original model. Empirical studies demonstrate that layer-wise optimization consistently outperforms LoRA in performance [37, 41]. Furthermore, by skipping gradient computations for frozen layers, layer-wise optimization eliminates the need to store their intermediate activations, offering greater memory efficiency than LoRA.

**Motivating questions.** All existing layer-wise LLM approaches [37, 36, 75, 41, 47] adopt the traditional BCD optimization paradigm, in which the model weights are divided into manageable blocks and updated iteratively. Although BCD provides a structured framework for handling the high dimensionality of LLM weights, fully realizing its potential in practical LLM fine-tuning requires addressing the following fundamental open questions.

*Q1. How to effectively partition the LLM's weights into blocks for optimization?*

The choice of partitioning strategy can significantly impact optimization efficiency and convergence. Current methods [37, 36, 75, 41, 47] treat transformer layers as homogeneous units, overlooking the differing significance of internal modules—such as multi-head attention, feed-forward networks, and normalization layers—within each layer. By uniformly updating all parameters in a layer, these methods risk over-adapting less impactful modules while under-training critical ones, resulting in suboptimal optimization performance. Thus, more effective partitioning strategies must be explored.

*Q2. How to effectively sample each block to achieve fast empirical convergence?*

The performance of layer-wise methods is highly influenced by the block sampling strategies. However, current approaches primarily rely on cyclic [37] or uniform [36] sampling patterns, which overlook the varying importance across layers. LISA [41] keeps the Embedding and LLM head layers active but assigns equal sampling probabilities to all transformer layers, ignoring their varying significance. BlockLLM [47] prioritize more frequent updates to critical layers but ultimately focus on a small set of fixed blocks, failing to adequately explore the remaining layers. An effective sampling strategy should maintain a balance between thorough exploration of the parameter space and efficient exploitation of the most promising optimization directions to ensure rapid convergence.

*Q3. How to establish convergence guarantees under practical LLM settings, incorporating Adam optimizer, stochastic gradients, and multiple updates per sampled block?*

Theoretical convergence of layer-wise LLM optimization remains understudied. Traditional BCD optimization literature typically focuses on block-wise GD/SGD convergence [59, 61, 22]. Recent works [37, 74, 41] provide convergence guarantees under restrictive assumptions, such as noiseless gradients or single updates per sampled block. These analyses fail to reflect practical layer-wise LLM optimization, where Adam optimizer, stochastic gradients, and multiple updates per block are standard practice. A rigorous convergence analysis under these realistic conditions is essential for broader adoption of layer-wise optimization methods.

**Main contributions.** To address the aforementioned open questions, we propose a novel **M**odule-wise **I**mportance **SA**mpling (**MISA**) method. MISA partitions the LLM's weights into smaller modules, which serve as blocks for optimization. Additionally, we assign importance scores to each module and develop an effective strategy that dynamically samples modules based on real-time importance metrics. Our main contributions are summarized as follows:

  **C1. Module-wise optimization.** We define a module as a matrix parameter within a transformer layer associated with weight gradients. Empirically, we observe that internal modules within

transformer layers exhibit heterogeneous importance. Theoretically, we demonstrate that decomposing each layer into smaller modules preserves more information in gradient, thus motivating module-wise optimization. This addresses **Question 1**. Furthermore, we find this fine-grained update strategy eliminates the need to load a full layer into memory, making it more memory-efficient than layer-wise LLM optimization approaches [37, 36, 75, 41, 47].

C2. **Improved importance sampling.** Traditional sampling strategies often rely on heuristics, which can be suboptimal for LLM optimization. To improve this, we parameterize the gradient variance as a function of the sampling probability for each module and maximize it to optimize the sampling strategy. Additionally, we introduce a strategy to balance importance and uniform distributions, ensuring both comprehensive exploration of all modules and efficient exploitation of critical ones, addressing **Question 2**.

C3. **Convergence guarantees.** We demonstrate that MISA achieves a convergence rate of $\mathcal{O}(1/\sqrt{K})$, where $K$ represents the total number of block updates. Our theoretical guarantees are derived under practical LLMs training scenarios, incorporating Adam optimization, stochastic gradients, and multiple updates per sampled block, thus addressing **Question 3**. Conventional BCD analysis relies on the assumption that block gradients are unbiased estimators of the full gradient—an assumption that fails when performing multiple updates within the same block. Our new analysis addresses this limitation by establishing fundamental connections between block-level gradients and the full gradient.

**Experimental results.** MISA demonstrates strong empirical performance.

E1. **Fine-tuning.** We evaluated MISA on different LLMs across three benchmarks: Commonsense Reasoning [25], Math Reasoning [25], and Instruction Following, encompassing a total of 16 datasets. We compared MISA against PEFT and layer-wise optimization methods, including LoRA [24], DoRA [35], BAdam [37], and LISA [41]. Under comparable memory constraints, MISA outperformed all baselines. An illustration of MISA's superiority in fine-tuning tasks is provided in Table 1, with more detailed results presented in Section 4.

E2. **Pre-training.** We trained the LLaMA2 130M and 350M variant [32] on the C4 dataset [46]. In 350M model training, MISA achieved a perplexity of 22.11 after 2.7B training tokens, significantly outperforming GaLore's 24.34 [71] , and approaching Adam's 21.3.

Table 1: Comparison of fine-tuning methods on LLaMA3-8B across eight commonsense reasoning tasks. The "ChatGPT" row presents results from ChatGPT obtained using Zero-shot CoT [58] with the GPT-3.5-turbo API. "Hella." refers to the HellaSwag dataset [65], and "Wino." refers to the Winogrande dataset [50]. Memory usage is reported without the application of additional memory-saving techniques such as gradient checkpointing or flash attention. The symbol $\delta$ denotes the proportion of parameters updated in each training iteration.

| Model | Method | Mem.(GB) | BoolQ | PIQA | SIQA | Hella. | Wino. | ARC-e | ARC-c | OBQA | Avg.↑ |
|---|---|---|---|---|---|---|---|---|---|---|---|
| ChatGPT | - | - | 73.1 | 85.4 | 68.5 | 78.5 | 66.1 | 89.8 | 79.9 | 74.8 | 77.0 |
| LLaMA3-8B | FT | 150.5 | 75.1 | 89.2 | 80.4 | 96.2 | 88.3 | 92.4 | 82.5 | 89.8 | 86.7 |
| | LoRA | 35.7 | 70.8 | 85.2 | 79.7 | 92.5 | 84.9 | 88.9 | 78.7 | 84.4 | 82.5 |
| | DoRA | 54.1 | 74.6 | 89.3 | 79.9 | 95.5 | 85.6 | 90.5 | 80.4 | 85.8 | 85.2 |
| | LISA | 56.3 | 74.6 | 88.1 | 81.5 | **96** | 86.4 | **92.5** | 81.7 | 86.2 | 85.9 |
| | BAdam | 34.1 | 74.2 | 87.1 | **81.6** | 95 | 84.6 | 91.2 | 79.8 | 84.8 | 84.8 |
| | **MISA($\delta = 1\%$)** | **30.7** | 74.1 | 88.6 | 80.7 | 95 | 85.6 | 92 | 81 | 86.2 | 85.4 |
| | **MISA($\delta = 3\%$)** | 34.4 | **75.4** | **90.5** | 81.4 | 95.9 | **88.2** | 92.2 | **82.4** | **87** | **86.6** |

# 2 Module-wise Importance Sampling

## 2.1 Problem Formulation

We consider the following problem:

$$\min_{\theta \in \mathbb{R}^d} \mathbb{E}_\xi[F(\theta; \xi)] \tag{1}$$

where $F(\theta; \xi)$ is a loss function that depends on a random variable $\xi$, and $\theta$ represents the set of $d$ trainable parameters. Conventional algorithms update all elements of $\theta$ simultaneously, leading to high memory and computational costs that hinder the training of LLMs on low-end hardware. In this work, we propose an alternative formulation of the problem in (1). Specifically, we introduce a block-wise representation of $\theta$, defined as $\theta = (\theta_1, \theta_2, \ldots, \theta_B)$, where $\theta_b \in \mathbb{R}^{d_b}$ represents the $b$-th

block of weights. Here, we have $d = \sum_{b=1}^{B} d_b$. The resulting problem formulation is then given by:
$$\min_{\theta_1,...,\theta_B} f(\theta_1, \theta_2, ..., \theta_B) := \mathbb{E}_\xi[F(\theta_1, \theta_2, ..., \theta_B; \xi)].$$

## 2.2 Block Sampling Strategy

At each time step, block-wise LLM training updates only a subset of parameters $\{\theta_i\}$. This approach inherently reduces the number of trainable parameters, resulting in lower memory and computational requirements. However, it also introduces greater variability in the gradient's unbiased estimator, which can potentially hinder convergence performance. This section will derive an effective importance sampling strategy to enhance block-coordinate optimization.

**Block coordinate descent.** Let $g_b^n := [\nabla f(\theta^n)]_b \in \mathbb{R}^{d_b}$ denote the $b$-th block gradient of $\nabla f(\theta^n)$, where $b \in [B]$ is the block index and $n \geq 0$ is the iteration index. Additionally, we define the matrix $U_b := [0; \cdots ; I_b; \cdots ; 0] \in \mathbb{R}^{d \times d_b}$, where the $b$-th block of $U_b$ is the identity matrix $I_b \in \mathbb{R}^{d_b \times d_b}$, and all other blocks are zero. Suppose each block gradient $g_b^n$ is sampled with probability $p_b$, the BCD algorithm will iterate as follows
$$\theta^{n+1} = \theta^n - \alpha U_b g_b^n, \quad \text{where block } b \text{ is sampled with probability } p_b.$$
where $\alpha$ is the learning rate. Assume $f(\theta)$ is $L$-smooth, we have

$$f(\theta^{n+1}) = f(\theta^n - \alpha U_b g_b^n) \leq f(\theta^n) - \alpha \langle \nabla f(\theta^n), U_b g_b^n \rangle + \frac{L\alpha^2}{2} \| U_b g_b^n \|^2$$

$$\stackrel{(a)}{=} f(\theta^n) - \alpha(1 - \frac{L\alpha}{2}) \| g_b^n \|^2 \stackrel{(b)}{\leq} f(\theta^n) - \alpha \| g_b^n \|^2 / 2$$

where (a) holds because $g_b^n = U_b^\top \nabla f(\theta^n)$ and (b) holds when $\alpha \leq 1/L$. Taking expectation on $b$,

$$\mathbb{E}[f(\theta^n) - f(\theta^{n+1})] \geq \frac{\alpha}{2} \sum_{b=1}^{B} p_b \| g_b^n \|^2.$$

**Importance sampling.** An ideal update $\theta^{n+1}$ should maximize the expected decrease $\mathbb{E}[f(\theta^n) - f(\theta^{n+1})]$, ensuring the most significant descent in the objective function. This can be achieved by

$$\max_{\{p_b\}_{b=1}^{B}} \sum_{b=1}^{B} p_b \| g_b^n \|^2, \quad \text{s.t.} \quad \sum_{b=1}^{B} p_b = 1, \quad p_b \geq 0.$$

Intuitively, this strategy prioritizes blocks with larger expected gradient norms, as they contribute more significantly to optimization progress. However, excessively prioritizing important blocks can lead to a scenario where only a few blocks are updated frequently. This results in over-exploitation at the expense of exploration. To encourage broader exploration, we constrain the sampling probabilities to stay close to a uniform distribution by incorporating a Kullback–Leibler (KL) divergence penalty:

$$\max_{\{p_b\}_{b=1}^{B}} \sum_{b=1}^{B} p_b \| g_b^n \|^2 - (1/\eta)\mathrm{KL}(p_b, q_B), \quad \text{s.t.} \quad \sum_{b=1}^{B} p_b = 1, \quad p_b \geq 0, \quad (2)$$

where $q_B = 1/B$ denotes the uniform distribution, and $\eta > 0$ is a coefficient controlling the trade-off between exploration and exploitation. As $\eta \to 0$, the KL divergence penalty dominates, and each $p_b$ approaches uniform sampling; as $\eta \to +\infty$, the penalty vanishes, recovering standard importance sampling. The following proposition provides a closed-form solution to problem (2).

**Proposition 1.** *The optimal solution to problem* (2) *is given as follows*
$$p_b^n = \frac{\exp(\eta \| g_b^n \|^2)}{\sum_{b=1}^{B} \exp(\eta \| g_b^n \|^2)}, \quad (3)$$
*which is the sampling probability of block $b$ at iteration $n$.*

**Practical implementation.** The sampling probability in (3) is impractical to implement, as $g_b^n = [\nabla f(\theta^n)]_b$ represents a full-batch block gradient, which is inaccessible during LLM optimization. In block-wise LLM training, each sampled block is typically updated for $T$ steps using the Adam optimizer before switching to another block. Let $g_b^{n,t} := [\nabla F(\theta^{n,t}; \xi^{n,t})]_b$ denote the $b$-th block stochastic gradient at outer iteration $n$ and inner update $t$. We approximate the full-batch gradient

norm $\|g_b^n\|^2$ using the empirical average $\frac{1}{T}\sum_{t=1}^T \|g_b^{n,t}\|^2$, resulting in a practical sampling strategy:

$$p_b^n = \frac{\exp\left(\eta G_b^n\right)}{\sum_{b=1}^B \exp\left(\eta G_b^n\right)} \quad \text{where} \quad G_b^n = \begin{cases} \beta G_b^{n-1} + (1-\beta)\frac{1}{T}\sum_{t=1}^T \|g_b^{n,t}\|^2 & \text{If } b \text{ is sampled;} \\ G_b^{n-1} & \text{otherwise.} \end{cases} \quad (4)$$

Instead of relying solely on the most recent $T$ mini-batch block stochastic gradients, $G_b^n$ aggregates all historical stochastic gradients to approximate the full-batch block gradient norm $\|g_b^n\|^2$ in (4). To eliminate the impact of differences in the number of parameters across blocks on gradient norm calculation, we actually use the **scaled gradient norm** in practice rather than the original gradient norm, and the specific definition is provided in Appendix B.2. We thus address **Question 2** by answering how to effectively sample blocks.

**Remark 1** (MEMORY AND COMPUTATION OVERHEAD). *The computation and storage cost to maintain $\{G_b^n\}_{b=1}^B$ is negligible compared to that of the block stochastic gradients. See Sections G.3 and F.5 for details.*

**Remark 2** (CLARIFICATION ON "LAYER", "MODULE", AND "BLOCK"). *A layer is a standard transformer component (e.g., with MHA and FFN), a module is a fine-grained subcomponent within a layer (e.g., $W_q$, $W_k$, $W_v$), and a block is a flexible optimization unit in block coordinate descent that can be a layer, a module, or a group of modules.*

### 2.3 Partition the Weights into Fine-Grained Modules Instead of Coarse Layers

Current approaches [37, 36, 75, 41, 47] often treat transformer layers as homogeneous structures, neglecting the distinct roles and contributions of their internal components—such as multi-head attention mechanisms, feed-forward networks, and normalization layers. As shown in Fig. 1, different modules within the same transformer layer exhibit significantly varying gradient norms, underscoring their differing levels of importance. Uniformly updating all modules within a layer risks over-adapting less critical components while under-training more essential ones, ultimately leading to suboptimal performance.

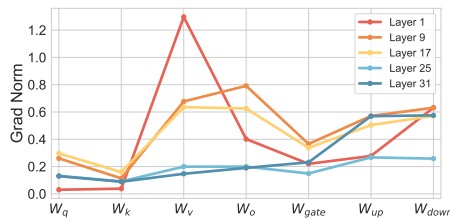

Figure 1: The gradient norm of different modules in different layers when fine-tuning LLaMA3-8B.

Motivated by this observation, we propose decomposing layers into distinct internal modules. In this work, a module is defined as a matrix parameter within a transformer layer that is associated with a weight gradient. For example, in the multi-head attention mechanism, we identify four modules—$W_q$, $W_k$, $W_v$, and $W_o$—while in the feed-forward network, we define two modules—$W_{up}$ and $W_{down}$. In other words, we treat each module as a sampling block, addressing **Question 2**. Proposition 2 characterizes the advantages of module-wise over layer-wise block sampling.

**Proposition 2.** *Suppose $\theta_b$ in problem (2) contains $K$ modules such that $\theta_b = (\theta_{b,1}, \theta_{b,2}, \ldots, \theta_{b,K})$ which associates with block gradient $g_b = (g_{b,1}, g_{b,2}, \ldots, g_{b,K})$. We further introduce $p_{bj}$ as the sampling probability of the block gradient $g_{b,j}$. Under this setting, problem (2) transforms into*

$$\max_{\{p_{bj}\}} \sum_{b=1}^B \sum_{j=1}^K p_{bj}\|g_{b,j}^n\|^2 - (1/\eta)\text{KL}(p_{bj}, q_{BK}), \quad \text{s.t.} \quad \sum_{b=1}^B \sum_{j=1}^K p_{bj} = 1, \quad p_{bj} \geq 0, \quad (5)$$

*Here, $q_{BK} = 1/(BK)$ is the uniform distribution. Any layer-wise importance sampling (i.e., a solution to problem (2)) is also a feasible solution to (5). Consequently, module-wise importance sampling (i.e., the optimal solution to problem (5)) will yield a larger objective function value than layer-wise importance sampling, making it a superior strategy to layer-wise sampling.*

### 2.4 Module-wise Importance Sampling (MISA) Method

Based on the discussions in Sec. 2.2 and 2.3, we now introduce **M**odule-wise **I**mportance **SA**mpling (**MISA**). The pseudocode for MISA is provided in Algorithm 1. MISA is a double-loop algorithm: the outer loop, indexed by $n$, selects a module, while the inner loop (Lines 7-12) updates the parameters of the selected module $T$ times using the Adam optimizer, with each update indexed by $t$. As shown in Line 1, MISA partitions the model into modules rather than layers. Lines 14-15 illustrate that MISA follows the effective sampling strategy described in Proposition 1. By adjusting $\eta$, we can

balance exploration across all modules with exploitation of the more important ones. In Line 16, the module weight $\theta_{\tau_n}^{n,T}$ undergoes an additional Adam step to obtain $\theta_{\tau_n}^{n+1,0}$, which facilitates the convergence analysis. In Line 17, we clear the optimizer states to ensure consistent memory efficiency throughout the training process.

---

**Algorithm 1 Module-wise Importance Sampling (MISA)**

---

**Require:** $\theta^0, N, T, B, \eta, \alpha, \delta > 0, \beta_1, \beta_2 \in (0,1)$ and the sampling block $\tau_n$ at outer loop $n$
1: Partition the model into $B$ modules (not layers);    ▷ Partition weights into modules;
2: Initialize probability weights $P^1 = (\frac{1}{B}, ..., \frac{1}{B})$;
3: Initialize the module gradient estimate $G_b^0 = 0$ for $b \in [B]$ and let $G^0 = (G_1^0, \cdots, G_B^0)$;
4: **for** $n = 1, ..., N$ **do**
5:    Sample $L_n$ modules (labeled with index $\tau_n$) according to $P^n$ such that the ratio of trainable parameter is less than $\delta$. (See Algorithm 2 for more details);    ▷ Importance sampling;
6:    Initialize $m_{\tau_n}^{n,0} = 0, v_{\tau_n}^{n,0} = 0$
7:    **for** $t = 1, ..., T$ **do**
8:       Sample a batch of data and calculate block stochastic gradient $g_{\tau_n}^{n,t}$ for selected module $\tau_n$;
9:       Update the corresponding first-order and second-order momentum as follows:
10:       $m_{\tau_n}^{n,t} \leftarrow \beta_1 m_{\tau_n}^{n,t-1} + (1-\beta_1) g_{\tau_n}^{n,t}, \quad v_{\tau_n}^{n,t} \leftarrow \beta_2 v_{\tau_n}^{n,t-1} + (1-\beta_2)(g_{\tau_n}^{n,t})^2$
11:       Update the corresponding module as follows:
12:       $\theta_{\tau_n}^{n,t} \leftarrow \theta_{\tau_n}^{n,t-1} - \alpha m_{\tau_n}^{n,t}/(\sqrt{v_{\tau_n}^{n,t} + \varepsilon})$
13:    **end for**
14:    Update $G_b^n$ for each $b \in [B]$ according to (4);    ▷ Track block gradient norm;
15:    Update $p_b^{n+1} \leftarrow \frac{\exp(\eta G_b^n)}{\sum_{j=1}^B \exp(\eta G_j^n)}$ for each $b \in [B]$;    ▷ Update sampling probability;
16:    $\theta_{\tau_n}^{n+1,0} \leftarrow \theta_{\tau_n}^{n,T} - \alpha \frac{\beta_1}{1-\beta_1} \frac{m_{\tau_n}^{n,T}}{\sqrt{v_{\tau_n}^{n,T} + \varepsilon}}$;
17:    $g_{\tau_n}, m_{\tau_n}, v_{\tau_n} \leftarrow None$    ▷ Clear optimizer states;
18: **end for**
19: **Return** $\theta^N, P^N$

---

Table 2: Comparison of existing memory-efficient optimization methods of LLMs.

| | LoRA[24] | IST[64] | GaLore[71] | OwLore[30] | BAdam[37] | LISA[41] | **MISA** |
|---|---|---|---|---|---|---|---|
| Full-rank Update | ✗ | ✗ | ✗ | ✗ | ✓ | ✓ | ✓ |
| Importance-aware | ✗ | ✓ | ✗ | ✓ | ✗ | ✓[1] | ✓ |
| Fine-grained memory | ✓ | ✓ | ✓ | ✓ | ✗ | ✗ | ✓ |
| Gradient acccumulation | ✓ | ✓ | ✗ | ✗ | ✓ | ✓ | ✓ |
| w/o SVD | ✓ | ✓ | ✗ | ✗ | ✓ | ✓ | ✓ |
| Fine-Tuning | ✓ | ✓ | ✓ | ✓ | ✓ | ✓ | ✓ |
| Pre-Training | ✗ | ✗ | ✓ | ✓ | ✓ | ✓ | ✓ |
| Convergence guarantee | ✗ | ✗ | ☹ | ✗ | ☹ | ✗ | ☺ |

[1]: LISA's importance-aware strategy focuses only on the embedding layer and LM-head layer, while transformer layers which account for the majority of model parameters still use uniform sampling.

☹: GaLore's convergence proof[71] holds only when the gradient exhibits a highly specialized structure, and BAdam's analysis[37] applies exclusively to the full-gradient regime. Their settings in the theoretical guarantees rely on restrictive assumptions.

☺: MISA's framework is based on the standard assumptions for analyzing non-convex optimization problems, making it more relevant to practical applications.

Table 2 provides a detailed comparison of existing optimization methods of LLMs. Notably, although LISA finds the embedding and LM-head layers to be very important, MISA does not train them in fine-tuning tasks because their parameters are too large (proportional to vocabulary size), and training them would lead to a significant increase in memory consumption.

## 2.5  Memory Analysis

We perform detailed memory analysis of MISA and present a comparative overview of memory consumption between block-wise optimization and subspace optimization methods, such as LoRA and Galore. Our findings demonstrate MISA's significant potential for long-sequence fine-tuning tasks. As illustrated in Fig. 2, for the LLaMA3-8B model, MISA significantly outperforms LoRA in terms of memory efficiency when the sequence length becomes sufficiently large. Notably, when $r = 16$, the trainable parameters in LoRA constitute only about 0.5% of the total parameters. In Appendix F, we demonstrate that (1) Layer-wise method acheives lower peak mem-

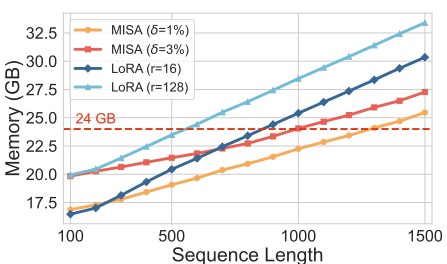

Figure 2: Comparison of peak memory consumption using MISA and LoRA fine-tuning on LLaMA3-8B across various sequence length.

ory than subspace method such as LoRA and GaLore in the long-sequence setting. (2) Layer-wise method can update more weights than LoRA in memory-constrained scenarios. (3) MISA achieves greater memory efficiency compared to the layer-wise method when the sampling ratio threshold $\delta$ is small.

## 3 Convergence Analysis

This section presents the convergence analysis for the proposed MISA method.

**Limitations in existing analysis.** Conventional BCD methods focus on the convergence of block-wise GD or SGD [59, 61, 22], but they do not generalize to the Adam optimizer. More recent studies [37, 74, 41] establish convergence results for layer-wise Adam, but only under restrictive conditions—such as assuming noiseless gradients or allowing just a single update per sampled block. These assumptions, however, diverge significantly from practical LLM training, where Adam is used with stochastic gradients and multiple updates are performed for each sampled block.

**New challenges.** Conventional BCD methods rely on the assumption that block gradients are unbiased estimators of the full gradient, an assumption that holds only when a single update is performed per sampled block—due to the preservation of unbiasedness through random block selection. However, our repeated block updating strategy introduces two key challenges: (1) Biased Gradient Estimates. Performing multiple updates on the same block breaks the randomness of block selection assumed in conventional BCD, resulting in biased estimates of the full gradient. (2) Bias-Noise Interplay. The introduced gradient bias interacts with inherent gradient noise, potentially amplifying estimation errors that would otherwise be mitigated by unbiased estimators.

**Analysis highlights.** Our work tackles these challenges via three innovations: (1) Bias Propagation Analysis. We derive novel recursive relations (Corollary 2) that characterize how gradient bias and stochastic noise accumulate over successive block updates. (2) Bridging Block and Full Gradients. We establish fundamental connections between block-level gradients and the full gradient (Corollary 3), despite the loss of unbiasedness. The additional Adam step in Line 15 of Algorithm 1 plays a critical role in ensuring a smooth transition from local block updates to the optimization of global variables. (3) Convergence Framework. We develop new analytical tools to jointly control gradient bias and amplified stochastic noise, resulting in rigorous convergence guarantees (Theorem 1).

**Main results.** We theoretically justify the convergence of MISA as follows.

**Theorem 1** (Informal). *Assume the loss function is L-smooth, the stochastic gradient is unbiased and bounded, the gradient variance is upper bounded (see Appendix E for detailed assumptions). With an appropriate learning rate $\alpha$, MISA achieves the following convergence rate (proof is in Appendix E):*

$$\frac{1}{N} \sum_{n=1}^{N} \mathbb{E}[\|\nabla f(\theta^{n,0})\|^2] = \mathcal{O}\left(\frac{1}{\sqrt{NT}} + \frac{1}{N}\right)$$

*where $N$ and $T$ denote the numbers of outer and inner-loop iterations, respectively.*

**Remark 3.** *According to Theorem 1, MISA converges for any number $T$ of inner local block updates, which constitutes a key advantage of our analysis. In contrast, the analyses in [59, 61, 74] are limited to the case $T = 1$, while the analysis in [48, 5] requires $T$ to be sufficiently large to ensure each block is adequately updated.*

# 4 Experiments

This section evaluates the performance of MISA across various fine-tuning and pre-training tasks. All our experiments are conducted on RTX 4090 24GB. We also conducted detailed ablation experiments, as presented in Appendix D.

## 4.1 Commonsense Reasoning

**Settings.** To validate the efficiency of MISA, we fine-tuned the LLaMA3-8B [14] and Qwen2.5-7B [63] models on the Commonsense Reasoning tasks. The dataset consists of eight subsets: BoolQ [6], PIQA [3], SIQA [51], HellaSwag [65], WinoGrande [50], ARC [7], and OBQA [40]. Following the settings in [25], we combined the training data from all eight tasks into a single training set for fine-tuning and then evaluated each model separately on the eight test sets. The baselines included LoRA [24], DoRA [35], and BCD-based methods, namely LISA [41] and BAdam [37]. We also reported GPU memory consumption for each method to ensure a thorough comparison. No additional memory-saving techniques, such as checkpointing [4] or flash attention [9], were employed.

**Results.** Table 3 presents the results for the commonsense tasks. Notably, DoRA's additional memory consumption arises from activations, while LISA incurs extra memory usage from fine-tuning both the embedding layer and the language modeling head layer. MISA outperforms all baseline methods on both the LLaMA-3-8B and Qwen2.5-7B models. Furthermore, with the trainable parameter ratio set to 1%, MISA achieves performance comparable to LoRA while saving approximately 10% of memory. In particular, MISA demonstrates significant improvements over BAdam and LISA in the Qwen2.5-7B model.

Table 3: This table compares fine-tuning methods on Qwen2.5-7B models across eight commonsense reasoning tasks. "Hella." refers to HellaSwag [65], and "Wino." to Winogrande [50]. The first row shows ChatGPT results from Zero-shot CoT [58] using the GPT-3.5-turbo API. **Refer to Table 1 for results of LLaMA3-8B.**

| Model | Method | Mem.(GB) | BoolQ | PIQA | SIQA | Hella. | Wino. | ARC-e | ARC-c | OBQA | Avg.↑ |
|---|---|---|---|---|---|---|---|---|---|---|---|
| Qwen2.5-7B | FT | 140.5 | 75.3 | 91 | 81.4 | 96.2 | 91.5 | 96.1 | 89.1 | 93.4 | 89.3 |
| | LoRA | 34.2 | 75.3 | 90.1 | 80.8 | **96.2** | 86.7 | **96.3** | 89 | 91.4 | 88.2 |
| | DoRA | 55.6 | 74.9 | 90.4 | 81.2 | 96.1 | 87.1 | 96.1 | 88.9 | 91.9 | 88.3 |
| | LISA | 57.9 | 75 | 90.3 | 80.6 | 95.2 | 86.2 | 95.8 | 89.1 | 89.8 | 87.8 |
| | BAdam | 33.7 | 74.5 | 90.3 | 80.8 | 95.5 | 86.1 | 96.1 | **89.6** | 91.8 | 88.1 |
| | **MISA($\delta = 1\%$)** | **30.3** | 74 | 90.1 | **82.4** | 95.5 | 87.1 | 96.1 | 89.3 | 90.2 | 88.1 |
| | **MISA($\delta = 3\%$)** | 34 | **75.4** | **90.5** | 81.6 | 95.7 | **87.2** | **96.3** | 89.5 | **92** | **88.5** |

## 4.2 Math Reasoning

**Settings.** To evaluate MISA on math reasoning tasks, we tested LLaMA3-8B and Qwen2.5-7B on four arithmetic benchmarks: GSM8K [8], SVAMP [42], AQUA [34], and MAWPS [28], following [25] for dataset settings. The models were fine-tuned on MATH10K [25], which combines training data from AQUA, GSM8K, and MAWPS, incorporating LM-generated chain-of-thought [58] steps.

**Results.** Table 4 presents the performance of various fine-tuning methods on LLaMA3-8B and Qwen2.5-7B. MISA outperforms all baselines on both models. Notably, MISA achieves a significant performance boost over LoRA and BAdam, which have similar GPU memory consumption. Compared to DoRA, MISA attains comparable performance with greater memory efficiency.

Table 4: Comparison of fine-tuning methods on LLaMA3-8B and Qwen2.5-7B across four arithmetic reasoning tasks. The left table presents results for LLaMA3-8B, while the right table displays results for Qwen2.5-7B. The unit of memory consumption is GB.

| Method | Mem. | GSM8K | SVAMP | AQuA | MAWPS | Avg.↑ | Mem. | GSM8K | SVAMP | AQuA | MAWPS | Avg.↑ |
|---|---|---|---|---|---|---|---|---|---|---|---|---|
| | | | | LLaMA3-8B | | | | | | Qwen2.5-7B | | |
| LoRA | 35.6 | 70.1 | 78.1 | 45.7 | **93.7** | 71.9 | 34.3 | 80.6 | 86.6 | 65.7 | 92.4 | 81.3 |
| DoRA | 54 | 70.9 | 78.4 | 50.3 | 93.4 | 73.3 | 54.8 | 80.1 | 86.7 | 68.1 | 92.5 | 81.9 |
| LISA | 54.2 | 68 | 76.7 | 49.7 | 88.3 | 70.7 | 55.6 | 77.6 | 86.8 | **70.1** | 89.8 | 81.1 |
| BAdam | 33.5 | 69.5 | 77.6 | 47.6 | 93.3 | 72 | 33.6 | 80.2 | 85.9 | 65.1 | 92.4 | 80.9 |
| **MISA($\delta = 1\%$)** | **30.4** | 68.5 | 77.3 | 49.3 | 90.1 | 71.3 | **29.8** | 77.5 | 85 | 65 | 89.5 | 79.3 |
| **MISA($\delta = 3\%$)** | 34.1 | **71.3** | **78.5** | **51.2** | 93.3 | **73.6** | 33.6 | **81** | **88.1** | 66.1 | **92.9** | **82** |

## 4.3 Instruction Tuning

To demonstrate MISA's efficiency in instruction-following fine-tuning, we fine-tuned TinyLLaMA[67], LLaMA2-7B[54] and Mistral-7B[26] on the Alpaca GPT-4 dataset, which consists of 52K samples generated by GPT-4 based on inputs from the Alpaca dataset [53]. Model performance was evaluated on MMLU[21], MMLU-pro[57] and MT-Bench [72], As shown in Table 5, MISA outperforms all baselines in most evaluation benchmarks, demonstrating its robustness and efficiency for instruction fine-tuning across different models. In addition, we compared the validation loss of MISA with other BCD methods, BAdam and LISA. As shown in Figure 3, the x-axis represents training time in minutes. MISA and LISA have similar total training time, while BAdam is faster than both. Despite this, MISA exhibits much better convergence compared to LISA and BAdam.

Table 5: Comparison of different methods on MMLU, MMLU-pro and MT-Bench. Results of LISA, GaLore, and LoRA in MMLU and MT-Bench are cited from LISA[41], while others are reproduced by us. MT-Bench evaluation was conducted using GPT-4. Memory was tested with batch size 2.

| Model | Method | Mem.(GB) | MMLU (5-shot) | MMLU-pro (5-shot) | MT-Bench |
|---|---|---|---|---|---|
| | LoRA | 5.00 | 25.81 | 11.32 | 1.90 |
| | GaLore | 5.00 | 25.21 | 10.96 | 2.61 |
| TinyLLaMA | LISA | 5.92 | **26.02** | 11.59 | 2.57 |
| | BAdam | 4.49 | 25.27 | 11.44 | 2.42 |
| | **MISA** | 4.36 | 25.40 | **11.65** | **2.73** |
| | LoRA | 20.48 | 45.50 | 20.57 | 4.45 |
| | GaLore | 19.25 | 45.56 | 20.19 | 4.63 |
| LLaMA2-7B | LISA | 22.79 | 46.21 | **20.85** | 4.94 |
| | BAdam | 19.83 | 45.61 | 20.68 | 4.81 |
| | **MISA** | 19.72 | **46.27** | 20.69 | **5.13** |
| | LoRA | 21.96 | 61.78 | 32.96 | 4.41 |
| | GaLore | 20.87 | 57.87 | 31.87 | 4.36 |
| Mistral-7B | LISA | 26.90 | 62.09 | 32.83 | 4.85 |
| | BAdam | 21.21 | 62.11 | 32.79 | 5.03 |
| | **MISA** | 21.18 | **62.90** | **33.44** | **5.19** |

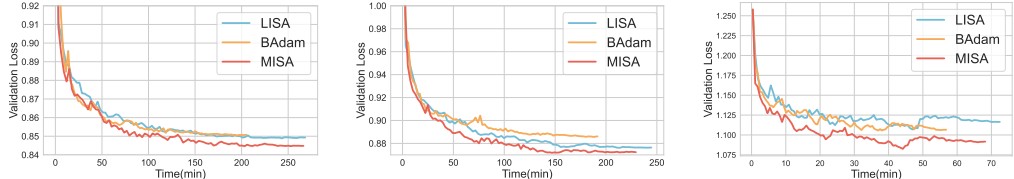

Figure 3: Validation loss of LISA, BAdam, and MISA across three epochs of fine-tuning Mistral-7B (left), LLaMA2-7B (middle) and TinyLLaMA (right) on the Alpaca-GPT4 dataset. The x-axis represents training time (minutes).

## 4.4 Pre-training

We evaluate MISA for pre-training LLaMA models [54] on the C4 dataset [46]. We pre-trained the 130M and 350M variant of LLaMA2 [33]. Unlike fine-tuning, where the embedding and output layers remain frozen, we trained them using the standard Adam optimizer. The number of training tokens is 2.75B. We report results for Adam, GaLore, and MISA. Table 6 reports the validation perplexity after 52K training steps. With a high-rank subspace (i.e., $r = 256$, $\delta = 25\%$), MISA outperforms GaLore and Adam on the 130M model. While on the 350M model, MISA outperforms GaLore and is slightly behind Adam. As shown in Fig. 4, MISA converges better than GaLore, and is similar with Adam's performance in the later training stage. MISA's strong performance indicates that it can be viewed as a regularization of Adam, and many gradients are redundant during full-parameter training.

Table 6: Validation perplexity of pre-training LLaMA 350M model on C4 dataset.

| | | Adam | GaLore | | MISA | |
|---|---|---|---|---|---|---|
| | | | r=32 | r=256 | $\delta$=3% | $\delta$=25% |
| **LLaMA 130M** | Perplexity | 24.63 | 44.88 | 28.27 | 36.07 | 23.81 |
| | Memory(GB) | 6.03 | 5.73 | 5.85 | 5.09 | 5.87 |
| **LLaMA 350M** | Perplexity | 21.30 | 39.43 | 24.34 | 30.62 | 22.11 |
| | Memory(GB) | 13.58 | 11.81 | 12.14 | 9.99 | 11.63 |

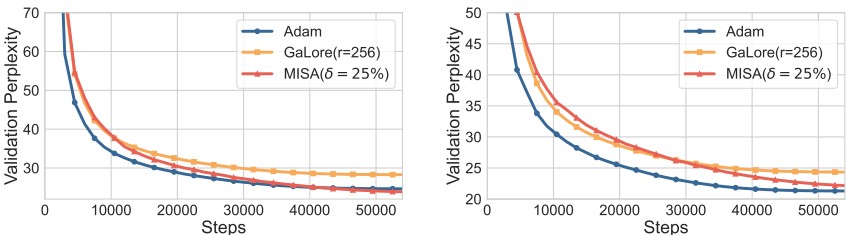

Figure 4: Pre-training dynamics for LLaMA 130M (left) and LLaMA 350M (right) on the C4 dataset.

## 5 Conclusion and Limitation

**Conclusions.** In this paper, we propose MISA, a novel memory-efficient training method. MISA decomposes each layer into smaller modules and employs an importance sampling mechanism for module training. We provide a detailed memory analysis demonstrating that MISA achieves high memory efficiency in long-sequence fine-tuning. Additionally, we show that MISA attains a convergence rate of $\mathcal{O}(1/\sqrt{K})$, where $K$ is the total number of updates.

**Limitations.** Due to resource constraints, we pre-trained MISA at a small scale, and whether MISA can scale to large-scale LLMs pre-training (e.g., 7B, 70B, or larger models) remains to be validated. Additionally, MISA has only been verified on text-modal Transformer-based LLMs (e.g., LLaMA series, Qwen2.5 series, Mistral series) and has not been extended to multi-modal models or non-Transformer architectures, so its adaptability across different modalities and model structures needs further exploration.

## 6 Acknowledgements

Kun Yuan is supported by the National Natural Science Foundation of China (NSFC) under Grants W2441021, 92370121 and 12301392. Tao Yao is supported by the National Natural Science Foundation of China (NSFC) under Grants W2441021, 72371172, 72342023, and 71929101.

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

# Appendix for "*MISA*: Memory-Efficient LLMs Optimization with Module-wise Importance Sampling"

## Contents

# A  Related Work

**Parameter-efficient fine-tuning.** Adapter-based methods introduce new modules (e.g., fully connected layers) into the backbone model and update only these modules during training [43, 23, 19, 56, 20], incurring additional computational costs during inference. Prompt-based methods prepend random soft tokens to the input (typically as a prefix) and train their embeddings while keeping the backbone LLM frozen [31, 18, 73, 44], also introducing inference overhead.

LoRA and its variants, including QLoRA [13], DoRA [35], and AdaLoRA [68], have become state-of-the-art PEFT methods. LoRA injects low-rank matrices into linear layers to estimate weight updates while keeping the backbone LLM parameters frozen. Unlike adapter- and prompt-based methods, LoRA has no additional inference overhead, as the low-rank matrices can be merged into the model weights. To address the limitations of the low-rank structure, recent methods such as ReLoRA [32], MoRA [27], and HiRA [1] have explored high-rank updates based on LoRA.

**Memory-efficient optimization.** LOMO [39] eliminates optimizer and gradient memory costs during training with standard SGD. AdaLOMO [38] extends LOMO by integrating a fused backward operation with an adaptive learning rate per parameter . GaLore [71] enhances memory efficiency by projecting gradients into a low-rank subspace. Quantization techniques are widely applied to model parameters, gradients, optimizer states, and intermediate activations [11, 29, 10]. Adam-mini [70] reduces memory usage by optimizing learning rate allocation in Adam.

**Layer-wise optimization.** Inspired by classical Block Coordinate Descent (BCD), layer-wise optimization has emerged as a promising approach for fine-tuning large language models. Methods such as LIFT [75], HIFT [36], and BAdam [37] update LLM layers sequentially, achieving performance comparable to parameter-efficient fine-tuning (PEFT) methods while significantly reducing computational costs. LISA [41] identified a skewed weight-norm distribution across layers in LoRA and introduced a randomized layer update strategy, keeping the embedding and language modeling head layers activated. Similarly, BlockLLM [47] prioritizes frequent updates to layers with larger gradient norms. However, none of these approaches incorporate module-wise importance sampling.

**Varying importance of modules in LLM fine-tuning.** Previous studies have shown that different layers contribute unequally to LLM performance. LayerSkip [15], LayerDrop [16], and Layer-wise Model Pruning [17] demonstrated that the overparameterization of pre-trained models leads to layer redundancy, enabling high performance to be maintained even after removing certain layers. IST [64] and ILA [52] further highlighted that layer importance varies during LLM fine-tuning. Additionally, studies on PEFT have revealed that different modules within transformer blocks exhibit varying ranks [2, 68]. OwLore[30] identifies layer importance by detecting outliers in the model weights. However, these insights have not yet motivate module-wise importance sampling in LLM optimization.

# B  Algorithm Details

## B.1  Module selection strategy

---

**Algorithm 2 Module selection**

---

**Require:** Active set $\tau_n$, Non-active set $B$, Sampling threshold $\delta$, Probability distribution $P^n$, Total model parameters $n_{model}$, Counter $S$.

1: **Initialize:** $\tau_n = \emptyset$, $B = \{M_1, M_2, ..., M_B\}$, $S = 0$
2: **while** $B \neq \emptyset$ **do**
3:      Sample a module $m$ from $B$ based on the probability distribution $P^n$.
4:      $B = B - m$.
5:      **if** $S + |m| < \delta n_{model}$ **then**
6:          $\tau_n = \tau_n + t$
7:      **end if**
8: **end while**
9: **Return:** $\tau_n$

---

### B.2 Definition of the scaled gradient norm

For a given module $m_i$, we define its scaled gradient norm be the ratio of the Frobenius norm of the gradient to the square root of the number of parameters $\frac{\|g_i\|_F}{\sqrt{|m_i|}}$ in practice, which can be considered a definition of normalization.

## C Additional Experiments

### C.1 Memory Efficiency When Scaling To 70B Model

Fig. 5 demonstrates that when training a 70B model, MISA is still more memory-efficient than LoRA under long sequence length settings, and the use of flash-attention[9] has minimal impact on overall memory consumption.

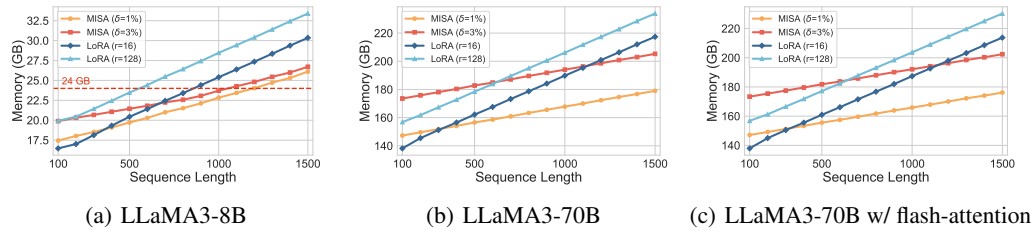

|            (a) LLaMA3-8B            |            (b) LLaMA3-70B            |     (c) LLaMA3-70B w/ flash-attention     |

Figure 5: Comparison of peak memory consumption on LLaMA3-8B and LLaMA3-70B. (c) used flash-attention technique.

### C.2 Enhancing LoRA with MISA

A key advantage of LoRA over MISA is its ability to train separate adapters for different downstream tasks. Instead of storing full model weights, LoRA only requires storage for task-specific adapters, significantly reducing storage costs. Since MISA is orthogonal to LoRA and can be used alongside it, we explored a hybrid approach combining both methods. Given a model weight $W$ with LoRA applied, it is represented as $W' = W + AB$, where $A$ and $B$ are treated as separate modules. We freeze the original model parameters and consider the LoRA adapters as the total set of trainable parameters $n_{\text{LoRA}}$. MISA dynamically activates LoRA adapters while ensuring that the number of active parameters at each step remains below $n_{\text{LoRA}} \times \delta$. Furthermore, we found that optimizer states are not the primary source of memory consumption for LoRA + MISA. Thus, we retain optimizer states for all LoRA adapters without clearing them. Our results show that MISA enhances full LoRA's performance while further reducing memory usage. As shown in Fig. 6, activating no more than 30% of LoRA parameters achieves or even surpasses full LoRA's performance while saving approximately 7.9% of memory. We also compared IST [64] with MISA (see Table 7) and MISA outperformed IST.

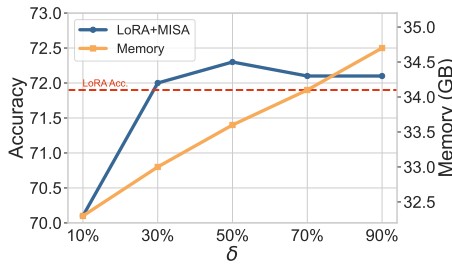

Figure 6: Accuracy and memory consumption of LoRA + MISA fine-tuning with varying $\delta$ on math reasoning tasks.

Table 7: Comparison of MISA and IST on LLaMA-7B and LLaMA3-8B across eight commonsense reasoning tasks. Results of LoRA+IST are cited from IST paper.

| Model | Method | BoolQ | PIQA | SIQA | Hella. | Wino. | ARC-e | ARC-c | OBQA | Avg.↑ |
|-------|--------|-------|------|------|--------|-------|-------|-------|------|-------|
| LLaMA-7B | LoRA | 68.9 | 80.7 | 77.4 | 78.1 | 78.8 | 77.8 | 61.3 | 74.8 | 74.7 |
| | LoRA+IST | 68.7 | 81.7 | 77.3 | 82.7 | 78.7 | 80.6 | 62.4 | 80.0 | 76.5 |
| | **LoRA+MISA** | 70.4 | 81.5 | 78.9 | 83.4 | 78.9 | 81.2 | 63.1 | 79.7 | **77.1** |
| LLaMA3-8B | LoRA | 70.8 | 85.2 | 79.7 | 92.5 | 84.9 | 88.9 | 78.7 | 84.4 | 82.5 |
| | LoRA+IST | 72.7 | 88.3 | 80.5 | 94.7 | 84.4 | 89.8 | 79.9 | 86.6 | 84.6 |
| | **LoRA+MISA** | 74.4 | 88.9 | 81.8 | 95.3 | 86.1 | 91.3 | 81.4 | 86.5 | **85.7** |

## C.3 Computation Efficiency

A training step includes forward propagation, backward propagation, and optimizer computations (including parameter updates and any algorithm-specific operations). Table 8 reports the average time per step for each method. GaLore incurs the highest optimizer computation overhead because it performs a full-model SVD (with very large time complexity) every 2000 steps. For MISA, we further conduct **detailed operation-level profiling** (under the same hardware and training configurations as Table 8) to quantify the overhead of its unique components (e.g., module sampling, gradient norm tracking). Results show that these components introduce negligible additional time (accounting for less than 0.05% of the total per-step time), as their key operations (e.g., gradient norm aggregation, sampling probability calculation) are either lightweight or amortized over multiple inner-loop steps. Thus, MISA's computation efficiency is only behind BAdam's, while it also achieves superior convergence with respect to training time (see Fig. 3).

Table 8: Comparison of training time for fine-tuning LLaMA2-7B on the Alpaca-GPT4 dataset. The Forward, Backward, and Optimizer time represent the average time per step. The last column shows the overall training time of 3 epochs.

| Method | Forward(ms) | Backward(ms) | Optimizer(ms) | Averged cost per step(ms) |
|--------|-------------|--------------|---------------|---------------------------|
| LoRA | 70.3 | 104.5 | 4.5 | 179.3 |
| GaLore | 35.7 | 140.5 | 267.4 | 443.6 |
| BAdam | 34.3 | 42.9 | 1.5 | 78.7 |
| LISA | 36.4 | 84.1 | 2.9 | 123.4 |
| **MISA** | 35.2 | 74.6 | 1.8 | 111.6 |

# D Ablation Study

## D.1 Ablation Study: Impact of Clearing Optimizer States

In line 17 of algorithm1, we clear the optimizer state to save memory consumption. We explored a method to maintain the optimizer states of active layers during training. When a layer is frozen, its complete optimizer state is preserved and offloaded to CPU memory. These saved states are then precisely restored to GPU when the layer is reactivated in subsequent training cycles.

As shown in Fig. 7, preserving optimizer states has no significant impact on fine-tuning loss but brings significant extra pretraining perplexity. However, we don't have a particularly good explanation for this phenomenon. Our guess is that in the early stage of pre-training, when the model is still in the coarse-tuning stage, the dependence on the history of momentum is not that strong, so the clearing optimizer may be able to avoid local minima and thus improve optimization.

## D.2 Ablation Study: Impact of Hyperparameters

We primarily investigated the effects of three hyperparameters: the learning rate, MISA's $\delta$, and MISA's $\eta$. $\eta$ controls the trade-off between importance sampling and uniform sampling. When $\eta = 0$,

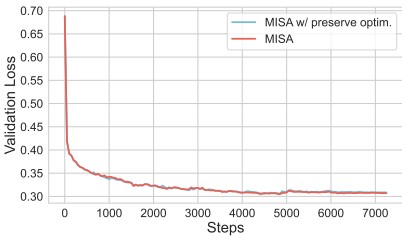 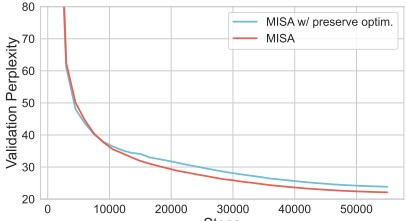

Figure 7: Fine-tuning loss (left) and pre-training perplexity (right) of LLaMA3-8B. Fine-tuning used MATH10K dataset, and pre-training used C4 dataset. MISA discards the optimizer states when switching activated layers, while "MISA w/ preserve optim." indicates retaining the optimizer states.

MISA reduces to uniform sampling. $\delta$ determines the number of parameters updated at each step and scales proportionally with memory consumption.

Fig. 8 presents the impact of $\eta$ and $\delta$. In practice, variations in $\eta$ have a minor effect on model accuracy, whereas MISA is highly sensitive to the learning rate. An excessively large learning rate significantly degrades performance of the model.

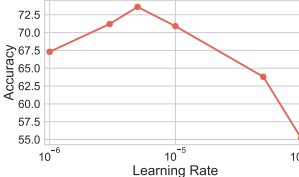 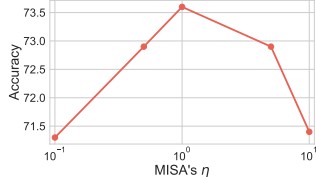

Figure 8: The accuracy of LLaMA3-8B on math reasoning tasks with different learning rate and $\eta$.

Fig. 9 illustrates the validation loss over three epochs of fine-tuning LLaMA2-7B on the Alpaca-GPT4 dataset for different $\delta$. We observe that larger $\delta$ lead to more severe overfitting in the third epoch, further supporting the interpretation of MISA as a regularization method of full parameter fine-tuning.

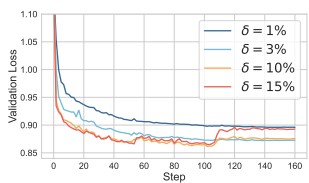

Figure 9: Validation loss of LLaMA2-7B on Alpaca-GPT4 dataset.

## D.3 Impact of Inner-Loop Iteration T

We conduct an ablation study to evaluate the sensitivity of MISA to the inner-loop iteration count $T$ (defined in Algorithm 1, representing the number of Adam updates for a sampled module before switching to another module). Experiments are conducted on the Mistral-7B model fine-tuned on the Alpaca-GPT4 dataset, with other hyperparameters fixed as: learning rate = 1e-5, $\delta = 3\%$, $\eta = 0.5$, and batch size = 16. Table 9 reports the validation loss and MMLU score across different $T$ values.

As shown in Table 9, varying $T$ from 5 to 500 does not significantly affect the convergence of validation loss or the MMLU performance. This observation aligns with the findings in BAdam [37] and LISA [41], confirming that MISA's convergence is robust to the choice of $T$—a result supported by our theoretical analysis in Theorem 1, where MISA is proven to converge for any number of inner-loop updates $T$. Notably, when $T \geq 200$, we observe a slight degradation in MMLU score (e.g., 46.01 at $T = 200$ and. 46.27 at $T = 50$), which may stem from over-fitting to the sampled module during extended inner-loop updates. Thus, we recommend setting $T = 30$–$50$ in practice to balance computational efficiency and performance.

Table 9: Ablation Study of the Inner-Loop Iteration $T$ for the Mistral-7B Model on Alpaca-GPT4

| $T$ | Validation Loss ($\downarrow$) | MMLU (5-shot, $\uparrow$) |
|---|---|---|
| 5 | 0.877 | 46.22 |
| 15 | 0.874 | 46.23 |
| 30 | 0.871 | 46.17 |
| 50 | 0.873 | 46.27 |
| 100 | 0.877 | 46.19 |
| 200 | 0.881 | 46.01 |
| 500 | 0.879 | 45.89 |

## D.4 Ablation Study: Sampling Strategy

To validate the necessity of MISA's sampling strategy, we explored alternative approaches. The Uniform strategy samples modules randomly without computing importance scores, effectively replacing BAdam's layer-wise optimization with module-wise optimization. The Top-K strategy selects the most important modules while ensuring that the total sampled parameters remain below $\delta$ of all parameters. In contrast, the Bottom-K strategy selects the least important modules under the same $\delta$ constraint. As shown in Table 10, the Uniform and Top-K strategies achieve similar performance but remain inferior to MISA, while the Bottom-K strategy significantly reduces accuracy. We observed that with the Top-K strategy, many modules are never sampled, which may explain its suboptimal performance. The effectiveness of MISA can be attributed to its balanced approach, combining elements of both the Uniform and Top-K strategies. It prioritizes modules with higher importance scores while ensuring that all modules have a chance to be sampled.

Table 10: Results of different sampling strategies for fine-tuning LLaMA3-8B on math and commonsense reasoning tasks. The reported scores indicate the average accuracy across all tasks.

| Strategy | Math | Commonsense |
|---|---|---|
| **MISA** | **73.6** | **86.6** |
| Uniform | 71.1(-2.5) | 85.9(-0.7) |
| Top-K | 71.2(-2.4) | 86(-0.6) |
| Bottom-K | 69.6(-4.0) | 82.1(-4.5) |

## D.5 Ablation Study: Importance Scoring Methods

In (4), we identify each module's importance score via its gradient norm. To validate its effectiveness, we also evaluated two alternative scoring functions: the module's weight norm and its total parameter count. Table 11 compares those importance scoring methods and demonstrates that MISA's sampling strategy significantly outperforms the alternatives, thereby validating the efficacy of MISA.

Table 11: Comparison of different scoring function in MISA. The accuracy of commonsense resoning and math reasoning benchmarks are averaged across all tasks.

| Model | Method | MMLU | MMLU-pro | Commonsense | Math |
|---|---|---|---|---|---|
| | Weight Norm | 64.5 | 35.9 | 85.7 | 71.9 |
| LLaMA3-8B | Number of Parameters | 63.7 | 36.2 | 85.9 | 72.7 |
| | **MISA(Gradient Norm)** | 65 | 36.5 | 86.6 | 73.6 |

## D.6 Ablation Study: Impact of Each Module

To assess the impact of each module, we applied MISA to individual modules while keeping all others frozen. Fig. 10 shows that the effectiveness of fine-tuning follows the order: $W_q, W_k < W_v < W_o, W_{gate}, W_{up}, W_{down}$. Interestingly, when applying MISA to all modules, the sampling frequency does not strictly follow this order, as shown in Fig. 11. We think there may be three

reasons: (1) Sampling frequency does not directly reflect importance—more critical layers may require less fine-tuning for optimal performance. (2) When training multiple modules simultaneously, the combined importance of those modules may not be simply the sum of each module's individual importance. (3) Module sizes vary significantly; for instance, $W_{gate}, W_{up}, W_{down}$ contain over ten times as many parameters as $W_k$ and $W_v$. Small-size modules are less likely to exceed the sampling threshold $\delta$, leading to more frequent sampling.

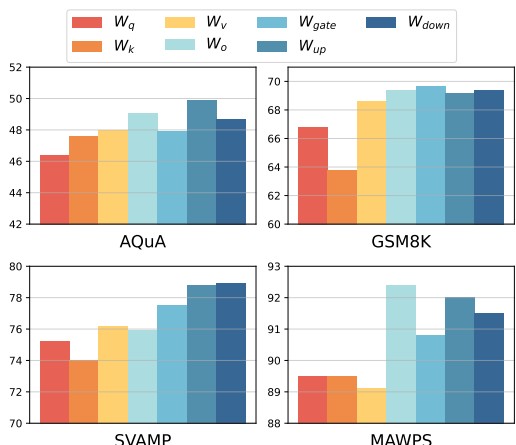

Figure 10: Accuracy comparison on math reasoning tasks when sample each individual module separately.

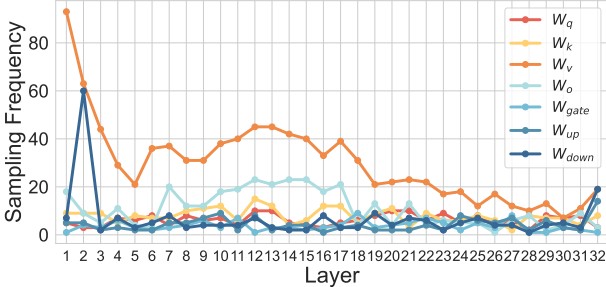

Figure 11: The frequency of module sampling across all layers when applying MISA to LLaMA3-8B.

Therefore, we further applied both uniform and MISA sampling strategies to individual modules in isolation, in order to assess which modules benefit most from MISA. As shown in Table 12, when fine-tuning $W_q, W_{gate}, W_{down}$ and $W_{down}$, MISA achieves substantially larger accuracy gains than uniform sampling.

Table 12: Average accuracy of LLaMA3-8B on math reasoning benchmark when fine-tuning only 1 module type with uniform and MISA sampling strategies. Red indicates an improvement of more than 1%.

| Model | Method | $W_q$ | $W_k$ | $W_v$ | $W_o$ | $W_{gate}$ | $W_{up}$ | $W_{down}$ |
|---|---|---|---|---|---|---|---|---|
| LLaMA3-8B | Uniform | 70 | 67.1 | 70.5 | 71.2 | 69.2 | 71.4 | 70.5 |
| | **MISA** | 69.5 | 68.8 | 70.5 | 71.7 | 71.5 | 72.5 | 72.1 |

# E   Theoretical Analysis.

## E.1   Notation

- Let $f : \mathbb{R}^d \to \mathbb{R}$ be the objective function to minimize.

- Let $f^* := \inf_{\theta \in \mathbb{R}^d} f(\theta)$ be the optimal value of $f$.

- Let $\nabla f$ denote the estimator for the gradient of $f$ at the queried points.

- Let $\alpha$ denote the learning rate.

- Let $n$ denote the block epoch, $n = 1, 2, ..., N$.

- Let $t$ denote the inner iterations for a certain block, $t = 0, 1, ..., T$.

- Let $\xi^{n,t}$ denote the sampled data at $n$-th block epoch and $t$-th inner iteration.

- Let $\tau_n$ denote the sampled block at $n$-th block epoch.

- Let $b$ denote the block's ID, $b = 1, 2, ..., B$.

- Let $\theta_{\tau_n}^{n,t}$ denote the parameter when selected module $\tau_n$ at $n$-th block epoch and t-th inner adam step.

- Let $g_{\tau_n}^{n,t}$ denote the partial derivative conditioned on data $\xi^{n,t}$ at $n$-th block epoch and $t$-th inner adam step.

- Let $\tilde{G}_{\tau_n}^{n,T} = \frac{1}{T} \sum_{t=1}^{T} \|g_{\tau_n}^{n,t}\|^2$ be the cumulative gains during a sampling period.

- Let $\Gamma_{\tau_n}^{n,t} = \text{Diag}^{-1/2}(\tilde{v}_{\tau_n}^{n,t} + \varepsilon, \|v_{\tau_n}^{n,t-1}\|_{\min} + \varepsilon)$, $\tilde{v}_{\tau_n}^{n,t} = \max\left(v_{\tau_n}^{n,t}, \|\tilde{v}_{\tau_n}^{n,t-1}\|_{\max}\right)$.

- Let $\theta^{n,t}$ denote the full parameter at n-th block epoch and t-th inner adam step.
$$\theta^{n,t} := \left(\theta_1^{n-1,T}, \theta_2^{n-1,T}, ..., \theta_{\tau_n-1}^{n-1,T}, \theta_{\tau_n}^{n,t}, \theta_{\tau_n+1}^{n-1,T}, ..., \theta_B^{n-1,T}\right).$$

- Let $\mathcal{F}_t$ denote the filtration containing all historical variables at and before computing $\theta_{\tau_n}^{n,t}$.
$$\mathcal{F}_t := \left\{\theta_{\tau_n}^{n,t}, \xi^{n,t} g_{\tau_n}^{n,t}, m_{\tau_n}^{n,t-1}, v_{\tau_n}^{n,t-1}, \theta_{\tau_n}^{n,t-2}, ..., \xi^{n,1}, g_{\tau_n}^{n,1}, m_{\tau_n}^{n,0}, v_{\tau_n}^{n,0}, \theta_{\tau_n}^{n,0}\right\}$$

- Let $\mathbb{E}_t$ denote the expectation conditioned on $\mathcal{F}_t$.

To enhance the smoothness of the proof procedure, we utilized three techniques.

## 1. Second-order Momentum Inheritance

To establish Lemma 1, we utilize a second-order momentum inheritance technique. Upon completing the $(n-1)$-th block epoch, we initialize the second-order momentum for the $n$-th block epoch as:
$$v_{\tau_n}^{n,0} = \|\tilde{v}_{\tau_{n-1}}^{n-1,T}\|_{\max} I \quad \text{for} \quad n \geq 2$$
This initialization strategy maintains memory efficiency, with its additional overhead being negligible compared to gradient storage requirements.

## 2. Additional Momentum Step

The additional momentum step arises naturally from the variable transformation in (11). This carefully designed operation prevents momentum wastage in $\tau_n$ during the $n$-th block epoch, significantly simplifying the analysis at block transition points. The step is defined as:
$$\theta_{\tau_n}^{n+1,0} \leftarrow \theta_{\tau_n}^{n,T} - \alpha \frac{\beta_1}{1 - \beta_1} \frac{m_{\tau_n}^{n,T}}{\sqrt{v_{\tau_n}^{n,T} + \varepsilon}}$$

## 3. Single-module Gradient Propagation

While MISA typically activates multiple modules simultaneously, our theoretical analysis focuses on the single-module case for clarity. The multi-module scenario can be represented by introducing a diagonal projection matrix $Q_n$, transforming the stochastic gradient in Algorithm 3 to:

$$g_{L_n}^{n,t} = Q_n \left. \frac{\partial}{\partial \theta^{n,t-1}} f\left(\theta^{n,t-1}\right) \right|_{\xi^{n,t}}$$

where $L_n$ denotes the set of activated modules at the $n$-th step. Our analysis naturally extends to this generalized case.

---

**Algorithm 3 MISA:Analytical View**

---

**Require:** $\theta^0, N, T, B, \eta, \alpha, \delta > 0, \beta_1, \beta_2 \in (0,1)$ and the sampling block $\tau_n$ at outer loop $n$

1: Partition the model into $B$ modules (not layers);        ▷ Partition weights into modules;
2: Initialize probability weights $P^1 = (\frac{1}{B}, ..., \frac{1}{B})$;
3: Initialize the module gradient estimate $G_b^0 = 0$ for $b \in [B]$ and let $G^0 = (G_1^0, \cdots, G_B^0)$;
4: **for** $n = 1, ..., N$ **do**
5:     Sample a module $\tau_n$ according to $P^n$;        ▷ Importance sampling;
6:     Initialize $m_{\tau_n}^{n,0} = 0, v_{\tau_n}^{1,0} = 0, v_{\tau_n}^{n,0} = \|\tilde{v}_{\tau_{n-1}}^{n-1,T}\|_{max} I$ if $n \geq 2$;
7:     **for** $t = 1, ..., T$ **do**
8:        Sample a batch of data $\xi^{n,t} \sim D$
9:        Calculate block stochastic gradient $g_{\tau_n}^{n,t} = \left. \frac{\partial}{\partial \theta_{\tau_n}^{n,t-1}} f\left(\theta_{\tau_n}^{n,t-1}\right) \right|_{\xi^{n,t}}$ for selected module $\tau_n$;
10:        Update the corresponding first-order and second-order momentum as follows:
11:        $m_{\tau_n}^{n,t} \leftarrow \beta_1 m_{\tau_n}^{n,t-1} + (1-\beta_1) g_{\tau_n}^{n,t}, \quad v_{\tau_n}^{n,t} \leftarrow \beta_2 v_{\tau_n}^{n,t-1} + (1-\beta_2) (g_{\tau_n}^{n,t})^2$
12:        Update $\tilde{v}_{\tau_n}^{n,t}$ as follows:
13:        $\tilde{v}_{\tau_n}^{n,t} = \max\left(v_{\tau_n}^{n,t}, \|\tilde{v}_{\tau_n}^{n,t-1}\|_{\max}\right)$        ▷ AMSGrad-type normalization;
14:        Update the corresponding module as follows:
15:        $\theta_{\tau_n}^{n,t} \leftarrow \theta_{\tau_n}^{n,t-1} - \alpha m_{\tau_n}^{n,t}/(\sqrt{\tilde{v}_{\tau_n}^{n,t}} + \varepsilon)$
16:     **end for**
17:     Update $G_b^n$ for each $b \in [B]$ according to (4);        ▷ Track block gradient norm;
18:     Update $p_b^{n+1} \leftarrow \frac{\exp(\eta G_b^n)}{\sum_{j=1}^B \exp(\eta G_j^n)}$ for each $b \in [B]$;        ▷ Update sampling probability;
19:     $\theta_{\tau_n}^{n+1,0} \leftarrow \theta_{\tau_n}^{n,T} - \alpha \frac{\beta_1}{1-\beta_1} \frac{m_{\tau_n}^{n,T}}{\sqrt{v_{\tau_n}^{n,T}} + \varepsilon}$;        ▷ Additional momentum step;
20:     $g_{\tau_n}, m_{\tau_n}, v_{\tau_n} \leftarrow None$        ▷ Clear optimizer states;
21: **end for**
22: **Return** $\theta^N, P^N$

---

**Assumption 1.** *The loss function is $\overline{L}$-smooth. And when restricted on $i$-th block, it is $L_i$-smooth.*

$$\|\nabla L\left(\theta^1\right) - \nabla L\left(\theta^2\right)\| \leq \overline{L}\|\theta^1 - \theta^2\|, \quad \left\| \left.\frac{\partial L}{\partial \theta_i}\right|_{\theta_i^1} - \left.\frac{\partial L}{\partial \theta_i}\right|_{\theta_i^2} \right\| \leq L_i\|\theta_i^1 - \theta_i^2\|, \ i = 1, ..., B.$$

*Let $L = \max\left\{\overline{L}, L_i\right\}_{i=1,...,B}$.*

**Assumption 2.** *(Unbiased and bounded stochastic gradient). For all inner iterates $t \geq 1$ at $n$-th block epoch, the stochastic gradient $g_{\tau_n}^{n,t}$ is unbiased and uniformly bounded by some constant $G \geq 0$.*

$$\mathbb{E}_t\left[g_{\tau_n}^{n,t}\right] = \nabla f\left(\theta_{\tau_n}^{n,t-1}\right), \quad \|g_{\tau_n}^{n,t}\| \leq G.$$

**Assumption 3.** *(Bounded variance). For all inner iterates $t$ at $n$-th block epoch, the variance of the stochastic gradient $g_{\tau_n}^{n,t}$ is uniformly bounded by some constant $\sigma^2 \geq 0$.*

$$\mathbb{E}_t\left[\|g_{\tau_n}^{n,t} - \nabla f\left(\theta_{\tau_n}^{n,t-1}\right)\|^2\right] \leq \sigma^2$$

**Proposition 3.** *(Proof of proposition 1) Consider the optimization problem*

$$\max_{p} \sum_{i=1}^{B} G_i^n p_i - \eta \mathrm{KL}(p_i, q_B)$$

$$s.t. \ p_i \geq 0, \ \sum_{i=1}^{B} p_i = 1,$$

*the solution is*

$$p_i^{n+1} = \frac{\exp\left(\eta_0 G_i^n\right)}{\sum_{b=1}^{B} \exp\left(\eta_0 G_i^n\right)}$$

*where $\eta_0 = \frac{1}{\eta}$.*

*Proof.* We consider the equivalence problem

$$\max_{p^n} \eta_0 \sum_{i=1}^{B} G_i^n p_i - \mathrm{KL}(p_i, q_B)$$

Where $\eta_0 = \frac{1}{\eta}$.

We construct the Lagrangian function $\mathcal{L}(P, \lambda)$ of the optimization problem

$$\mathcal{L}(P, \lambda) = \eta_0 \sum_{i=1}^{B} p_i G_i^n - \sum_{i=1}^{B} p_i log \frac{p_i}{q_B} + \lambda(\sum_{i=1}^{B} p_i - 1)$$

We take the partial derivative of $p_i$ and obtain

$$\frac{\partial \mathcal{L}}{\partial p_i} = \log q_B - \log p_i - 1 + \eta_0 G_i^n + \lambda = 0, \quad i = 1, 2, ..., B$$

Which is equivalent to

$$p_i = q_B \exp\left(\eta_0 G_i^n + \lambda - 1\right), \quad i = 1, 2, ..., B.$$

According to the normalization condition, we have

$$1 = \sum_{i=1}^{B} p_i = \sum_{i=1}^{B} q_B \exp\left(\eta_0 G_i^n + \lambda - 1\right)$$

Let $Z = \exp(1 - \lambda)$, we have

$$Z = \sum_{i=1}^{B} q_B \exp(\eta_0 G_i^n)$$

Which implies

$$p_i^{n+1} = \frac{q_B \exp\left(\eta_0 G_i^n\right)}{\sum_{i=1}^{B} q_B \exp\left(\eta_0 G_i^n\right)} = \frac{\exp\left(\eta_0 G_i^n\right)}{\sum_{i=1}^{B} \exp\left(\eta_0 G_i^n\right)}$$

□

**Proposition 4.** *(Proof of proposition 2 ) Suppose that in each layer we have $K$ modules such that $\theta_i = (\theta_{i_1}, \theta_{i_2}, ..., \theta_{i_K})$, the sampling probability of each layer be $p_i$, each module be $p_{i,j}$, the gain of each module be $G_{i,j}$ at $n$-th step and certain iteration, we have*

$$\frac{1}{K} \sum_{i=1}^{B} G_i^n p_i \leq \sum_{i=1}^{B} \sum_{j=1}^{K} G_{i,j}^n p_{i,j}$$

*Proof.* We consider $p_i = \sum_{j=1}^{K} p_{i,j}$, which means the sum of each module's sampling probability is equal to the sampling probability of the corresponding layer.

Noting that in layer-wise setting, the sampling probability of each module is $\frac{p_i}{K}$.

Next, we will prove

$$G_i^n \frac{p_i}{K} \leq \sum_{j=1}^{K} G_{i,j}^n p_{i,j}$$

Noting that gain $G_i^n$ is equal to the sum of each individual module gains $G_{i,j}^n$, we have

$$\frac{\sum_{j=1}^{K} G_{i,j}^n}{K} \leq \sum_{j=1}^{K} G_{i,j}^n \hat{p}_{i,j} \tag{6}$$

Where $\hat{p}_{i,j} = \frac{p_{i,j}}{p_i}, \sum_{j=1}^{K} \hat{p}_{i,j} = 1$.

By symmetry, we can assume without loss of generality that $G_{i,j}^n$ is ordered as follows.

$$G_{i,1}^n \leq G_{i,2}^n \leq ... \leq G_{i,K}^n$$

From Proposition 3, it's obvious that

$$\hat{p}_{i,1} \leq \hat{p}_{i,2} \leq ... \leq \hat{p}_{i,K}$$

The left side of inequality (6) is arithmetic average, the ride side is weighted average of $G_{i,j}^n$.

From Chebyshev's inequality, inequality (6) holds. Summing over $i$, we obtain

$$\frac{1}{K} \sum_{i=1}^{B} G_i^n p_i \leq \sum_{i=1}^{B} \sum_{j=1}^{K} G_{i,j}^n p_{i,j}$$

$\square$

**Corollary 1.** *The sampling probability $\{p_i^n\}_{i=1,2,...,B}^{n \in \mathbb{N}}$ in algorithm 3 has a uniform lower bound $\pi$.*

*Proof.* From Assumption 2, for $\forall\, i \in [1, B], n \in \mathbb{N}^*$, we have

$$\tilde{G}_i^{n,T} = \frac{\sum_{t=1}^{T} \|g_i^{n,t}\|^2}{T} \leq \frac{TG^2}{T} = G^2$$

We obtain

$$G_i^N = (1 - \beta)\, G_i^{N-1} + \beta \tilde{G}_i^{N,T} \leq (1 - \beta)\, G_i^{N-1} + \beta G^2 \tag{7}$$

Let $p = 1 - \beta, q = \beta G$.

Updating equation (7) for $K$ times, we obtain

$$G_i^{N+K} \leq \frac{q}{1-p} + p^{N+k-1} \left( G_i^N - \frac{q}{1-p} \right)$$

Noting that $0 < p < 1$, for $\forall k \in \mathbb{N}^*$, we obtain

$$G_i^{N+K} \leq G^2 + (1 - \beta)^N \left( G_i^N - G^2 \right)$$

Let $\pi_i^* = \max_{1 \leq n \leq N} \left\{ G_i^n, G^2 + (1 - \beta)^N \left( G_i^N - G^2 \right) \right\}$

We have

$$\frac{1}{Be^{\eta \pi_i^*}} \leq \frac{\exp\left(\eta G_i^n\right)}{\sum_{j=1}^{B} \exp\left(\eta G_j^n\right)} = p_i^n$$

We get the lower bound $\pi_i = \frac{1}{Be^{\eta \pi_i^*}}$

Let $\pi = min\{\pi_i\}$, which is the uniform lower bound of $\{p_i^n\}_{i=1,2,...,B}^{n \in \mathbb{N}}$. $\square$

**Lemma 1.** *Let $\Gamma_{\tau_n}^{n,t} = Diag^{-1/2}(\tilde{v}_{\tau_n}^{n,t} + \varepsilon, \|v_{\tau_n}^{n,t-1}\|_{\min} + \varepsilon),\ \Delta\Gamma_{\tau_n}^{n,t} = \Gamma_{\tau_n}^{n,t-1} - \Gamma_{\tau_n}^{n,t}, t = 1, 2, ..., T$*

*We have the following upper bound estimation*

$$\sum_{n=1}^{N}\sum_{t=1}^{T}\|\Delta\Gamma_{\tau_n}^{n,t}\| \le \frac{2}{\sqrt{\varepsilon}}, \quad \sum_{n=1}^{N}\sum_{t=1}^{T}\|\Delta\Gamma_{\tau_n}^{n,t}\|^2 \le \frac{2}{\varepsilon}.$$

*Where $\Delta\Gamma_{\tau_n}^{n,0} = \Gamma_{\tau_{n-1}}^{n-1,T} - \Gamma_{\tau_n}^{n,0}$.*

*Proof.* From algorithm 3, we have $v_{\tau_n}^{n,0} = \|\tilde{v}_{\tau_{n-1}}^{n-1,T}\|_{max}I, n \ge 2$.

For notational convenience, we denote $\Gamma_t$, $\tilde{v}_t$ as $\Gamma_{\tau_n}^{n,t}$, $\tilde{v}_{\tau_n}^{n,t}$ in this proof.

$$\begin{aligned}
\Gamma_t &= \mathrm{Diag}^{-1/2}(\tilde{v}_t + \varepsilon, \|\tilde{v}_t\|_{\min} + \varepsilon) \\
&\preceq \mathrm{Diag}^{-1/2}(\|\tilde{v}_t\|_{\min} + \varepsilon) \\
&\preceq \frac{1}{\sqrt{\|\tilde{v}_{t-1}\|_{\max} + \varepsilon}}I \\
&= \mathrm{Diag}^{-1/2}(\|\tilde{v}_{t-1}\|_{\max} + \varepsilon) \\
&\preceq \mathrm{Diag}^{-1/2}(\tilde{v}_{t-1} + \varepsilon, \|\tilde{v}_{t-1}\|_{\min} + \varepsilon) = \Gamma_{t-1}
\end{aligned}$$

which implies that $\Delta\Gamma_t = \Gamma_{t-1} - \Gamma_t$ is positive semi-definite. Hence, $\|\Delta\Gamma_t\| = \lambda_{max}(\Delta\Gamma_t) \ge 0$

Using the convexity of $\lambda_{\max}$ over symmetric matrices, we have

$$\begin{aligned}
\sum_{t=1}^{T}\|\Delta\Gamma_t\| &= \sum_{t=1}^{T}\lambda_{\max}(\Gamma_{t-1} - \Gamma_t) \\
&\le \sum_{t=1}^{T}\left(\lambda_{\max}(\Gamma_{t-1}) + \lambda_{\max}(-\Gamma_t)\right) \\
&= \sum_{t=1}^{T}\left(\lambda_{\max}(\Gamma_{t-1}) - \lambda_{\min}(\Gamma_t)\right) \\
&= \sum_{t=1}^{T}\left(\frac{1}{\sqrt{\|\tilde{v}_{t-1}\|_{\min} + \varepsilon}} - \frac{1}{\sqrt{\|\tilde{v}_t\|_{\max} + \varepsilon}}\right) \\
&= \sum_{t=1}^{T}\left(\frac{1}{\sqrt{\|\tilde{v}_{t-1}\|_{\min} + \varepsilon}} - \frac{1}{\sqrt{\|\tilde{v}_{t-1}\|_{\max} + \varepsilon}} + \frac{1}{\sqrt{\|\tilde{v}_{t-1}\|_{\max} + \varepsilon}} - \frac{1}{\sqrt{\|\tilde{v}_t\|_{\max} + \varepsilon}}\right) \\
&\le \sum_{t=1}^{T}\left(\frac{1}{\sqrt{\|\tilde{v}_{t-1}\|_{\min} + \varepsilon}} - \frac{1}{\sqrt{\|\tilde{v}_t\|_{\min} + \varepsilon}}\right) + \sum_{t=1}^{NT}\left(\frac{1}{\sqrt{\|\tilde{v}_{t-1}\|_{\max} + \varepsilon}} - \frac{1}{\sqrt{\|\tilde{v}_t\|_{\max} + \varepsilon}}\right) \\
&= \frac{1}{\sqrt{\|\tilde{v}_0\|_{\min} + \varepsilon}} - \frac{1}{\sqrt{\|\tilde{v}_T\|_{\min} + \varepsilon}} + \frac{1}{\sqrt{\|\tilde{v}_0\|_{\max} + \varepsilon}} - \frac{1}{\sqrt{\|\tilde{v}_T\|_{\max} + \varepsilon}}
\end{aligned}$$

For the second sum of squared norms, notice that for scalars $a \geq b \geq 0$, it holds that $(a - b)^2 \leq (a + b)(a - b) = a^2 - b^2$. Therefore, the derivation can be repeated without the square roots:

$$\sum_{t=1}^{T} \|\Delta \Gamma_t\|^2 = \sum_{t=1}^{T} (\lambda_{\max}(\Gamma_{t-1} - \Gamma_t))^2$$

$$\leq \sum_{t=1}^{T} (\lambda_{\max}(\Gamma_{t-1}) + \lambda_{\max}(-\Gamma_t))^2 = \sum_{t=1}^{T} (\lambda_{\max}(\Gamma_{t-1}) - \lambda_{\min}(\Gamma_t))^2$$

$$\leq \sum_{t=1}^{T} (\lambda_{\max}(\Gamma_{t-1}))^2 - (\lambda_{\min}(\Gamma_t))^2$$

$$= \sum_{t=1}^{T} \left( \frac{1}{\|\tilde{v}_{t-1}\|_{\min} + \varepsilon} - \frac{1}{\|\tilde{v}_t\|_{\max} + \varepsilon} \right)$$

$$= \sum_{t=1}^{T} \left( \frac{1}{\|\tilde{v}_{t-1}\|_{\min} + \varepsilon} - \frac{1}{\|\tilde{v}_{t-1}\|_{\max} + \varepsilon} + \frac{1}{\|\tilde{v}_{t-1}\|_{\max} + \varepsilon} - \frac{1}{\|\tilde{v}_t\|_{\max} + \varepsilon} \right)$$

$$\leq \sum_{t=1}^{T} \left( \frac{1}{\|\tilde{v}_{t-1}\|_{\min} + \varepsilon} - \frac{1}{\|\tilde{v}_t\|_{\min} + \varepsilon} \right) + \sum_{t=1}^{T} \left( \frac{1}{\|\tilde{v}_{t-1}\|_{\max} + \varepsilon} - \frac{1}{\|\tilde{v}_t\|_{\max} + \varepsilon} \right)$$

$$= \frac{1}{\|\tilde{v}_0\|_{\min} + \varepsilon} - \frac{1}{\|\tilde{v}_T\|_{\min} + \varepsilon} + \frac{1}{\|\tilde{v}_0\|_{\max} + \varepsilon} - \frac{1}{\|\tilde{v}_T\|_{\max} + \varepsilon}$$

Applying the above estimation, we obtain

$$\sum_{n=1}^{N} \sum_{t=1}^{T} \|\Delta \Gamma_{\tau_n}^{n,t}\| \leq \sum_{n=1}^{N} \left( \frac{1}{\sqrt{\|\tilde{v}_{\tau_n}^{n,0}\|_{\min}} + \varepsilon} - \frac{1}{\sqrt{\|\tilde{v}_{\tau_n}^{n,T}\|_{\min}} + \varepsilon} + \frac{1}{\sqrt{\|\tilde{v}_{\tau_n}^{n,0}\|_{\max}} + \varepsilon} - \frac{1}{\sqrt{\|\tilde{v}_{\tau_n}^{n,T}\|_{\max}} + \varepsilon} \right)$$

$$\leq \sum_{n=1}^{N} \left( \frac{2}{\sqrt{\|\tilde{v}_{\tau_n}^{n,0}\|_{\min}} + \varepsilon} - \frac{2}{\sqrt{\|\tilde{v}_{\tau_n}^{n,T}\|_{\max}} + \varepsilon} \right)$$

$$= \frac{2}{\sqrt{\|\tilde{v}_{\tau_1}^{1,0}\|_{\min}} + \varepsilon} - \sum_{n=1}^{N-1} \left( \frac{2}{\sqrt{\|\tilde{v}_{\tau_n}^{n,T}\|_{\max}} + \varepsilon} - \frac{2}{\sqrt{\|\tilde{v}_{\tau_{n+1}}^{n+1,0}\|_{\min}} + \varepsilon} \right) - \frac{2}{\sqrt{\|\tilde{v}_{\tau_N}^{N,T}\|_{\max}} + \varepsilon}$$

$$\leq \frac{2}{\sqrt{\|\tilde{v}_{\tau_1}^{1,0}\|_{\min}} + \varepsilon} \leq \frac{2}{\sqrt{\varepsilon}}$$

Where the last inequality is because $v_{\tau_{n+1}}^{n+1,0} = \|\tilde{v}_{\tau_n}^{n,T}\|_{max} I$. Similarly, we obtain

$$\sum_{n=1}^{N} \sum_{t=1}^{T} \|\Delta \Gamma_{\tau_n}^{n,t}\|^2 \leq \frac{2}{\varepsilon}.$$

Which completes the proof. □

**Lemma 2.**

$$\mathbb{E}_t[\|m_{\tau_n}^{n,t}\|] \leq G, \quad \sum_{t=1}^{T} \mathbb{E}_t \left[ \|\frac{\beta_1 m_{\tau_n}^{n,t-1}}{1 - \beta_1}\|^2 \right] \leq (\frac{\beta_1}{1 - \beta_1})^2 G^2 T, \quad \mathbb{E}_t \left[ \|\tilde{v}_{\tau_n}^{n,t}\|_{\max} \right] \leq G^2$$

*Proof.*

$$\|m_{\tau_n}^{n,t}\| = \|(1 - \beta_1) \sum_{i=1}^{t} \beta_1^{t-i} g_{\tau_n}^{n,i}\| \leq (1 - \beta_1) \sum_{i=1}^{t} \beta_1^{t-i} G \leq G$$

Taking expectation, we obtain

$$\mathbb{E}_t[\|m_{\tau_n}^{n,t}\|] \leq G \tag{8}$$

From equation ([8](#)), we have

$$\|\frac{\beta_1 m_{\tau_n}^{n,t-1}}{1-\beta_1}\|^2 = (\frac{\beta_1}{1-\beta_1})^2 \|m_{\tau_n}^{n,t-1}\|^2 \leq (\frac{\beta_1}{1-\beta_1})^2 G^2$$

We get the bound

$$\sum_{t=1}^{T} \mathbb{E}_t\left[\|\frac{\beta_1 m_{\tau_n}^{n,t-1}}{1-\beta_1}\|^2\right] \leq (\frac{\beta_1}{1-\beta_1})^2 G^2 T$$

Next, we bound $\|v_{\tau_n}^{n,t}\|$

$$\|v_{\tau_n}^{n,t}\| = \|(1-\beta_2)\sum_{i=1}^{t} \beta_2^{t-i}(g_{\tau_n}^{n,i})^2\| \leq (1-\beta_2)\sum_{i=1}^{t} \beta_2^{t-1}G^2 \leq G^2.$$

We obtain

$$\mathbb{E}_t[\|\tilde{v}_{\tau_n}^{n,t}\|_{\max}] \leq G^2.$$

$\square$

**Corollary 2.** *Suppose Assumptions [1](#), [2](#), [3](#) are satisfied. When the learning rate meets the condition that $\alpha \leq \min\left\{\sqrt{\frac{\varepsilon}{4C_0 L}}, \frac{\sqrt{\varepsilon}}{C_1 L}\right\}$, inner iterations at $n$-th block epoch satisfies:*

$$\mathbb{E}_t[f(x_{\tau_n}^{n,T}) - f(x_{\tau_n}^{n,0})] \leq -\frac{\alpha}{2C_0}\sum_{t=0}^{T-1}\mathbb{E}_t[\|\nabla_{\tau_n}f(\theta_{\tau_n}^{n,t})\|^2] + \frac{T\alpha^2 L\sigma^2}{\varepsilon} + \frac{T\alpha^3 C_1^2 L^2 C_0 G^2}{\varepsilon^2}$$

$$+2\alpha(1+C_1)G^2\sum_{t=1}^{T}\mathbb{E}_t\left[\|\Delta\Gamma_{\tau_n}^{n,t}\|\right] + \frac{2\alpha^2 C_1^2 LG^2}{\varepsilon} \tag{9}$$

*Proof.* First, we define a series $\left\{x_{\tau_n}^{n,t}\right\}$, $t = 0, 1, ..., T$

$$x_{\tau_n}^{n,t} = \theta_{\tau_n}^{n,t} - \alpha\frac{\beta_1 \Gamma_{\tau_n}^{n,t} m_{\tau_n}^{n,t}}{1-\beta_1}$$

Noting that for $t = 1, 2, ..., T$, we have

$$m_{\tau_n}^{n,0} = 0, \quad x_{\tau_n}^{n,0} = \theta_{\tau_n}^{n,0}, \quad \theta_{\tau_n}^{n,t} = \theta_{\tau_n}^{n,t-1} - \alpha\Gamma_{\tau_n}^{n,t} m_{\tau_n}^{n,t} \tag{10}$$

We will prove the following recurrence relation

$$x_{\tau_n}^{n,t} = x_{\tau_n}^{n,t-1} - \alpha\Gamma_{\tau_n}^{n,t} g_{\tau_n}^{n,t} + \alpha\Delta\Gamma_{\tau_n}^{n,t}\frac{\beta_1}{1-\beta_1}m_{\tau_n}^{n,t-1}, \quad t = 1, 2, ..., T \tag{11}$$

For notational convenience, we denote $\theta_t, g_t, m_t, \Gamma_t, \nabla f(\theta_t)$ as $\theta_{\tau_n}^{n,t}, g_{\tau_n}^{n,t}, m_{\tau_n}^{n,t}, \Gamma_{\tau_n}^{n,t}, \nabla f(\theta_{\tau_n}^{n,t})$ below.

From the definition of $\left\{x_i^{n,t}\right\}$, we have

$$x_t = \theta_t - \alpha\Gamma_t\frac{\beta_1}{1-\beta_1}m_t$$

$$= \theta_{t-1} - \alpha\Gamma_t m_t - \alpha\Gamma_t\frac{\beta_1}{1-\beta_1}m_t$$

$$= \theta_{t-1} - \alpha\Gamma_t\frac{1}{1-\beta_1}m_t$$

$$= \theta_{t-1} - \alpha\Gamma_t\frac{\beta_1 m_{t-1} + (1-\beta_1)g_t}{1-\beta_1}$$

$$= \theta_{t-1} - \alpha\Gamma_t g_t - \alpha\Gamma_t\frac{\beta_1 m_{t-1}}{1-\beta_1}$$

Noting that

$$x_{t-1} = \theta_{t-1} - \alpha \Gamma_{t-1} \frac{\beta_1}{1 - \beta_1} m_{t-1}$$

We obtain

$$x_t = x_{t-1} - \alpha \Gamma_t g_t + \alpha \Delta \Gamma_t \frac{\beta_1}{1 - \beta_1} m_{t-1}$$

Next, we apply smoothness assumption1 of the loss function $f$ over the iterates $x_{\tau_n}^{n,t}$, $t = 1, ..., T$.

$$f(x_t) \leq f(x_{t-1}) + \langle \nabla f(x_{t-1}), x_t - x_{t-1} \rangle + \frac{L}{2} \|x_t - x_{t-1}\|^2.$$

Taking expectation, we obtain

$$\mathbb{E}_t[f(x_t)] - \mathbb{E}_t[f(x_{t-1})] \leq -\alpha \mathbb{E}_t \left[ \langle \nabla f(x_{t-1}), \Gamma_t g_t \rangle \right] + \alpha \mathbb{E}_t \left[ \left\langle \nabla f(x_{t-1}), \Delta \Gamma_t \left( \frac{\beta_1}{1 - \beta_1} m_{t-1} \right) \right\rangle \right]$$

$$+ \frac{\alpha^2 L}{2} \mathbb{E}_t \left[ \left\| \Gamma_t g_t - \Delta \Gamma_t \left( \frac{\beta_1}{1 - \beta_1} m_{t-1} \right) \right\|^2 \right]$$

Expanding further, we have

$$\mathbb{E}_t[f(x_t)] - \mathbb{E}_t[f(x_{t-1})] \leq -\alpha \mathbb{E}_t \left[ \langle \nabla f(\theta_{t-1}), \Gamma_t g_t \rangle \right] \quad \text{(I)}$$
$$+ \alpha \mathbb{E}_t \left[ \langle \nabla f(\theta_{t-1}) - \nabla f(x_{t-1}), \Gamma_t g_t \rangle \right] \quad \text{(II)}$$
$$+ \alpha \mathbb{E}_t \left[ \left\langle \nabla f(x_{t-1}), \Delta \Gamma_t \left( \frac{\beta_1}{1 - \beta_1} m_{t-1} \right) \right\rangle \right] \quad \text{(III)}$$
$$+ \frac{\alpha^2 L}{2} \mathbb{E}_t \left[ \left\| \Gamma_t g_t - \Delta \Gamma_t \left( \frac{\beta_1}{1 - \beta_1} m_{t-1} \right) \right\|^2 \right] \quad \text{(IV)}$$

We will estimate each part step by step.

**Bounding term I**

$$I = -\alpha \mathbb{E}_t \left[ \langle \nabla f(\theta_{t-1}), \Gamma_{t-1} g_t \rangle \right] + \alpha \mathbb{E}_t \left[ \langle \nabla f(\theta_{t-1}), \Delta \Gamma_t g_t \rangle \right]$$
$$\leq -\alpha \mathbb{E}_t \left[ \langle \nabla f(\theta_{t-1}), \Gamma_{t-1} \nabla L(\theta_{t-1}) \rangle \right] + \alpha G^2 \mathbb{E}_t \left[ \|\Delta \Gamma_t\| \right]$$
$$\leq -\alpha \lambda_{\min}(\Gamma_{t-1}) \mathbb{E}_t \left[ \|\nabla f(\theta_{t-1})\|^2 \right] + \alpha G^2 \mathbb{E}_t \left[ \|\Delta \Gamma_t\| \right]$$
$$\leq -\frac{\alpha}{C_0} \mathbb{E}_t \left[ \|\nabla f(\theta_{t-1})\|^2 \right] + \alpha G^2 \mathbb{E}_t \left[ \|\Delta \Gamma_t\| \right]$$

Where $\frac{1}{C_0} = \frac{1}{\sqrt{G^2 + \varepsilon}} \leq \lambda_{\min}(\Gamma_{t-1})$, the first inequality is because assumption 2, the second inequality is because lemma 2.

**Bounding term II**

$$II = \alpha \mathbb{E}_t \left[ \langle \nabla f(\theta_{t-1}) - \nabla f(x_{t-1}), \Gamma_{t-1} g_t \rangle \right] - \alpha \mathbb{E}_t \left[ \langle \nabla f(\theta_{t-1}) - \nabla f(x_{t-1}), \Delta \Gamma_t g_t \rangle \right]$$

$$\leq \alpha \mathbb{E}_t \left[ \langle \nabla f(\theta_{t-1}) - \nabla f(x_{t-1}), \Gamma_{t-1} \nabla f(\theta_{t-1}) \rangle \right] + \alpha^2 L \mathbb{E}_t \left[ \left\| \Gamma_{t-1} \left( \frac{\beta_1 m_{t-1}}{1 - \beta_1} \right) \right\| \|\Delta \Gamma_t g_t\| \right]$$

$$\leq \frac{\alpha \rho}{2\varepsilon} \mathbb{E}_t \left[ \|\nabla f(\theta_{t-1})\|^2 \right] + \frac{\alpha}{2\rho} \mathbb{E}_t \left[ \|\nabla f(\theta_{t-1}) - \nabla f(x_{t-1})\|^2 \right] + \frac{\alpha^2 C_1 L G^2}{\sqrt{\varepsilon}} \mathbb{E}_t \left[ \|\Delta \Gamma_t\| \right]$$

$$\leq \frac{\alpha \rho}{2\varepsilon} \mathbb{E}_t \left[ \|\nabla f(\theta_{t-1})\|^2 \right] + \frac{\alpha^3 L^2}{2\rho} \mathbb{E}_t \left[ \left\| \Gamma_{t-1} \left( \frac{\beta_1 m_{t-1}}{1 - \beta_1} \right) \right\|^2 \right] + \frac{\alpha^2 C_1 L G^2}{\sqrt{\varepsilon}} \mathbb{E}_t \left[ \|\Delta \Gamma_t\| \right]$$

$$\leq \frac{\alpha \rho}{2\varepsilon} \mathbb{E}_t \left[ \|\nabla f(\theta_{t-1})\|^2 \right] + \frac{\alpha^3 C_1^2 L^2 G^2}{2\rho \varepsilon} + \frac{\alpha^2 C_1 L G^2}{\sqrt{\varepsilon}} \mathbb{E}_t \left[ \|\Delta \Gamma_t\| \right]$$

where $C_1 = \frac{\beta_1}{1-\beta_1}$ , $\rho$ is a positive constant, we used the fact $\|\Gamma_t\| = \frac{1}{\sqrt{\|\bar{v}_t\|_{\min}+\varepsilon}} \leq \frac{1}{\sqrt{\varepsilon}}$, the first and third inequality is because assumption 1, the second inequality is because Cauchy–Schwarz inequality.

**Bounding term III**

$$
\begin{aligned}
\text{III} &= \alpha\mathbb{E}_t\left[\left\langle \nabla f(x_{t-1}), \Delta\Gamma_t\left(\frac{\beta_1}{1-\beta_1}m_{t-1}\right)\right\rangle\right]\\
&\leq \alpha\mathbb{E}_t\left[\left\langle \nabla f(\theta_{t-1}), \Delta\Gamma_t\left(\frac{\beta_1}{1-\beta_1}m_{t-1}\right)\right\rangle\right] + \alpha\mathbb{E}_t\left[\left\langle \nabla f(x_{t-1}) - \nabla f(\theta_{t-1}), \Delta\Gamma_t\left(\frac{\beta_1 m_{t-1}}{1-\beta_1}\right)\right\rangle\right]\\
&\leq \alpha\mathbb{E}_t\left[\|\nabla f(\theta_{t-1})\|\|\Delta\Gamma_t\frac{\beta_1 m_{t-1}}{1-\beta_1}\|\right] + \alpha^2 L\mathbb{E}_t\left[\|\Gamma_{t-1}\left(\frac{\beta_1 m_{t-1}}{1-\beta_1}\right)\|\|\Delta\Gamma_t\frac{\beta_1 m_t}{1-\beta_1}\|\right]\\
&\leq \alpha C_1 G^2 \mathbb{E}_t\left[\|\Delta\Gamma_t\|\right] + \frac{\alpha^2 C_1^2 LG^2}{\sqrt{\varepsilon}}\mathbb{E}_t\left[\|\Delta\Gamma_t\|\right]
\end{aligned}
$$

Where the third inequality is because assumption 1.

**Bounding term IV**

$$
\begin{aligned}
\mathbf{IV} &= \frac{\alpha^2 L}{2}\mathbb{E}_t\left[\left\|\Gamma_t g_t - \Delta\Gamma_t\left(\frac{\beta_1}{1-\beta_1}m_{t-1}\right)\right\|^2\right]\\
&\leq \alpha^2 L\mathbb{E}_t[\|\Gamma_t g_t\|^2] + \alpha^2 L\mathbb{E}_t\left[\left\|\Delta\Gamma_t\left(\frac{\beta_1}{1-\beta_1}m_{t-1}\right)\right\|^2\right]\\
&\leq \frac{\alpha^2 L}{\varepsilon}\mathbb{E}_t\left[\|g_t - \nabla f(\theta_{t-1}) + \nabla f(\theta_{t-1})\|^2\right] + \alpha^2 L\mathbb{E}_t\left[\|\Delta\Gamma_t\left(\frac{\beta_1 m_{t-1}}{1-\beta_1}\right)\|^2\right]\\
&\leq \frac{\alpha^2 L}{\varepsilon}\left(\mathbb{E}_t\left[\|\nabla f(\theta_{t-1})\|^2\right] + \sigma^2\right) + \alpha^2 C_1^2 LG^2\mathbb{E}_t\left[\|\Delta\Gamma_t\|^2\right]\\
&\leq \frac{\alpha^2 L}{\varepsilon}\mathbb{E}_t\left[\|\nabla f(\theta_{t-1})\|^2\right] + \frac{\alpha^2 L\sigma^2}{\varepsilon} + \alpha^2 C_1^2 LG^2\mathbb{E}_t\left[\|\Delta\Gamma_t\|^2\right]
\end{aligned}
$$

Where we used assumption 3 that $g_t$ is unbiased with bounded variance $\sigma^2$.

We obtain the bound of each part

$$
\mathbf{I} \leq -\frac{\alpha}{C_0}\mathbb{E}_t\left[\|\nabla f(\theta_{t-1})\|^2\right] + \alpha G^2\mathbb{E}_t\left[\|\Delta\Gamma_t\|\right]
$$

$$
\mathbf{II} \leq \frac{\alpha\rho}{2\varepsilon}\mathbb{E}_t\left[\|\nabla f(\theta_{t-1})\|^2\right] + \frac{\alpha^3 C_1^2 L^2 G^2}{2\rho\varepsilon} + \frac{\alpha^2 C_1 LG^2}{\sqrt{\varepsilon}}\mathbb{E}_t\left[\|\Delta\Gamma_t\|\right]
$$

$$
\mathbf{III} \leq \alpha C_1 G^2\mathbb{E}_t\left[\|\Delta\Gamma_t\|\right] + \frac{\alpha^2 C_1^2 LG^2}{\sqrt{\varepsilon}}\mathbb{E}_t\left[\|\Delta\Gamma_t\|\right]
$$

$$
\mathbf{IV} \leq \frac{\alpha^2 L}{\varepsilon}\mathbb{E}_t\left[\|\nabla f(\theta_{t-1})\|^2\right] + \frac{\alpha^2 L\sigma^2}{\varepsilon} + \alpha^2 C_1^2 LG^2\mathbb{E}_t\left[\|\Delta\Gamma_t\|^2\right]
$$

Substituting each part into the above single-step estimation, we obtain

$$
\mathbb{E}_t[f(x_t) - f(x_{t-1})] \leq \left(-\frac{\alpha}{C_0} + \frac{\alpha^2 L}{\varepsilon} + \frac{\alpha\rho}{2\varepsilon}\right)\mathbb{E}_t[\|\nabla f(\theta_{t-1})\|^2] + \frac{\alpha^2 L\sigma^2}{\varepsilon} + \frac{\alpha^3 C_1^2 L^2 G^2}{2\rho\varepsilon}
$$

$$
+ \left(\alpha(1+C_1)G^2 + \frac{\alpha^2(1+C_1)C_1 LG^2}{\sqrt{\varepsilon}}\right)\mathbb{E}_t\left[\|\Delta\Gamma_t\|\right] + \alpha^2 C_1^2 LG^2\mathbb{E}_t\left[\|\Delta\Gamma_t\|^2\right]
$$

Updating $T$ times, we obtain

$$
\mathbb{E}_t[f(x_T) - f(x_0)] \leq \left(-\frac{\alpha}{C_0} + \frac{\alpha^2 L}{\varepsilon} + \frac{\alpha\rho}{2\varepsilon}\right)\sum_{t=0}^{T-1}\mathbb{E}_t[\|\nabla f(\theta_t)\|^2] + \frac{T\alpha^2 L\sigma^2}{\varepsilon} + \frac{T\alpha^3 C_1^2 L^2 G^2}{2\rho\varepsilon}
$$

$$+ \left( \alpha(1+C_1)G^2 + \frac{\alpha^2(1+C_1)C_1 L G^2}{\sqrt{\varepsilon}} \right) \sum_{t=1}^{T} \mathbb{E}_t \left[ \|\Delta \Gamma_t\| \right] + \alpha^2 C_1^2 L G^2 \sum_{t=1}^{T} \mathbb{E}_t \left[ \|\Delta \Gamma_t\|^2 \right].$$

Choosing $\rho = \frac{\varepsilon}{2C_0}$, $\alpha \leq \min \left\{ \sqrt{\frac{\varepsilon}{4C_0 L}}, \frac{\sqrt{\varepsilon}}{C_1 L} \right\}$, applying lemma1, we obtain:

$$\mathbb{E}_t[f(x_T) - f(x_0)] \leq -\frac{\alpha}{2C_0} \sum_{t=0}^{T-1} \mathbb{E}_t[\|\nabla f(\theta_t)\|^2] + \frac{T\alpha^2 L \sigma^2}{\varepsilon} + \frac{T\alpha^3 C_1^2 L^2 C_0 G^2}{\varepsilon^2}.$$

$$+ 2\alpha(1+C_1)G^2 \sum_{t=1}^{T} \mathbb{E}_t \left[ \|\Delta \Gamma_t\| \right] + \frac{2\alpha^2 C_1^2 L G^2}{\varepsilon}$$

.

$\square$

**Corollary 3.** *For each inner iteration index $T_1 = 0, 1, ..., T$ at $n$-th block epoch, the expectation of $\|\nabla_{\tau_n} f \left( \theta^{n,0} \right)\|^2$ is bounded by $\|\nabla_{\tau_n} f \left( \theta_{\tau_n}^{n,T_1} \right)\|^2$*

$$\frac{\mathbb{E}_t \left[ \|\nabla_{\tau_n} f \left( \theta^{n,0} \right)\|^2 \right] - \alpha^2 T^2 C_2}{2} \leq \mathbb{E}_t \left[ \|\nabla_{\tau_n} f \left( \theta_{\tau_n}^{n,T_1} \right)\|^2 \right] \tag{12}$$

*Where $C_2 = \frac{2L^2 G^2}{\varepsilon}$.*

*Proof.*

$$\mathbb{E}_t \left[ \|\nabla_{\tau_n} f \left( \theta^{n,0} \right)\|^2 \right] \leq 2\mathbb{E}_t \left[ \|\nabla_{\tau_n} f \left( \theta_{\tau_n}^{n,T_1} \right)\|^2 \right] + 2\mathbb{E}_t \left[ \|\nabla_{\tau_n} f \left( \theta^{n,0} \right) - \nabla_{\tau_n} f \left( \theta_{\tau_n}^{n,T_1} \right)\|^2 \right]$$

$$\leq 2\mathbb{E}_t \left[ \|\nabla_{\tau_n} f \left( \theta_{\tau_n}^{n,T_1} \right)\|^2 \right] + 2\mathbb{E}_t \left[ \|\nabla f \left( \theta^{n,0} \right) - \nabla f \left( \overline{\theta}_{\tau_n}^{n,T_1} \right)\|^2 \right]$$

$$\leq 2\mathbb{E}_t \left[ \|\nabla_{\tau_n} f \left( \theta_{\tau_n}^{n,T_1} \right)\|^2 \right] + 2L^2 \mathbb{E}_t \left[ \|\theta^{n,0} - \overline{\theta}_{\tau_n}^{n,T_1}\|^2 \right]$$

Where

$$\overline{\theta}_{\tau_n}^{n,T_1} = \left( \theta_1^{n-1,T}, ..., \theta_{\tau_n-1}^{n-1,T}, \theta_{\tau_n}^{n,T_1}, \theta_{\tau_n+1}^{n-1,T} ..., \theta_B^{n-1,T} \right)$$

$$\theta^{n,0} = \left( \theta_1^{n-1,T}, ..., \theta_{\tau_n-1}^{n-1,T}, \theta_{\tau_n}^{n,0}, \theta_{\tau_n+1}^{n-1,T} ..., \theta_B^{n-1,T} \right),$$

the second inequality is because the partial derivative vector $\nabla_{\tau_n} f \left( \theta_{\tau_n}^{n,T_1} \right)$ is part of the gradient vector $\nabla f \left( \overline{\theta}_{\tau_n}^{n,T_1} \right)$.

Next, we will bound $\mathbb{E}_t \left[ \|\theta^{n,0} - \overline{\theta}_{\tau_n}^{n,T_1}\|^2 \right]$.

$$\mathbb{E}_t \left[ \|\theta^{n,0} - \overline{\theta}_{\tau_n}^{n,T_1}\|^2 \right] = \alpha^2 \mathbb{E}_t \left[ \|\sum_{j=1}^{T_1} \Gamma_{\tau_n}^{n,j} m_{\tau_n}^{n,j}\|^2 \right]$$

$$\leq \frac{T_1 \alpha^2}{\varepsilon} \sum_{j=1}^{T_1} \mathbb{E}_t \left[ \|m_{\tau_n}^{n,j}\|^2 \right]$$

$$\leq \frac{T_1^2 \alpha^2 G^2}{\varepsilon} \leq \frac{T^2 \alpha^2 G^2}{\varepsilon}$$

Where we used assumption 3 and lemma 2, the first inequality is because Cauchy-shwarz inequality.

We obtain

$$\mathbb{E}_t \left[ \|\nabla_{\tau_n} f \left( \theta^{n,0} \right)\|^2 \right] \leq 2\mathbb{E}_t \left[ \|\nabla_{\tau_n} f \left( \theta_{\tau_n}^{n,T_1} \right)\|^2 \right] + \frac{2T^2 \alpha^2 L^2 G^2}{\varepsilon}.$$

Let $C_2 = \frac{2L^2 G^2}{\varepsilon}$, we have

$$\frac{\mathbb{E}_t \left[ \|\nabla_{\tau_n} f \left( \theta^{n,0} \right)\|^2 \right] - \alpha^2 T^2 C_2}{2} \leq \mathbb{E}_t \left[ \|\nabla_{\tau_n} f \left( \theta_{\tau_n}^{n,T_1} \right)\|^2 \right].$$

$\square$

## E.2 Proof of theorem 1

**Theorem 2.** *(MISA Convergence) Suppose that Assumptions 1, 2, 3 are fulfilled, learning rate satisfies $\alpha \le \min\left\{\sqrt{\frac{\varepsilon}{4C_0 L}}, \frac{\sqrt{\varepsilon}}{C_1 L}\right\}$, MISA converges at the rate of $\mathcal{O}(\frac{1}{\sqrt{NT}} + \frac{1}{N})$.*

$$\frac{\sum_{n=1}^{N} \mathbb{E}_t[\|\nabla f(\theta^{n,0})\|^2]}{N} \le \frac{1}{\sqrt{N}} \frac{4C_0\sqrt{\Delta_0[TL\sigma^2 + 2LG^2(\frac{\beta_1}{1-\beta_1})^2]}}{T\pi\sqrt{\varepsilon}} + \mathcal{O}(\frac{1}{N}).$$

*Where $\Delta_0 = f(x^{1,0}) - f(x^*)$, $C_1 = \frac{\beta_1}{1-\beta_1}$, $C_0 = \frac{1}{\sqrt{G^2+\varepsilon}}$.*

*Proof.* From algorithm 3, we have

$$\theta_{\tau_{n+1}}^{n+1,0} = \theta_{\tau_n}^{n,T} - \alpha\frac{\beta_1}{1-\beta_1}\Gamma_{\tau_n}^{n,T}m_{\tau_n}^{n,T}$$

From the definition of the sequence $\left\{x_{\tau_n}^{n,T}\right\}$ in (10), we obtain

$$\mathbb{E}_t[f(x_{\tau_n}^{n+1,0})] = \mathbb{E}_t[f(\theta_{\tau_n}^{n+1,0})] = \mathbb{E}_t[f(x_{\tau_n}^{n,T})]$$

Which implies

$$\mathbb{E}_t[f(x_{\tau_n}^{n+1,0})] = \mathbb{E}_t[f(x_{\tau_n}^{n,T})] \tag{13}$$

Substituting (12) and (13) into (9) ,we obtain

$$\mathbb{E}_t[f(x_{\tau_n}^{n+1,0}) - f(x_{\tau_n}^{n,0})] \le -\frac{\alpha T}{4C_0}\mathbb{E}_t[\|\nabla_{\tau_n}f(\theta^{n,T})\|^2] + \frac{T\alpha^2 L\sigma^2}{\varepsilon} + \frac{T\alpha^3 C_1^2 L^2 C_0 G^2}{\varepsilon^2}.$$

$$+2\alpha(1+C_1)G^2\sum_{t=1}^{T}\mathbb{E}_t\left[\|\Delta\Gamma_t\|\right] + \frac{2\alpha^2 C_1^2 LG^2}{\varepsilon} + \frac{\alpha^3 T^3 C_2}{4C_0}$$

It's obvious that the norm of full gradient is equal to the sum of individual block's gradient norms.

$$\|\nabla f\left(\theta^{n,T}\right)\|^2 = \sum_{b=1}^{B}\|\nabla_b f\left(\theta^{n,T}\right)\|^2$$

Taking expectation, we have

$$\mathbb{E}_t\left[\|\nabla f\left(\theta^{n,T}\right)\|^2\right] = \sum_{b=1}^{B}\mathbb{E}_t\left[\|\nabla_b f\left(\theta^{n,T}\right)\|^2\right]$$

Using the conclusion of Corollary 2 and taking expectation with respect to $\tau_n$, we have

$$\mathbb{E}_{t,\tau_n}[f(x^{n+1,0}) - f(x^{n,0})] = \sum_{b=1}^{B}p_b^n\mathbb{E}_t[f(x_b^{n+1,0}) - f(x_b^{n,0})]$$

$$\le -\frac{\alpha T}{4C_0}\sum_{b=1}^{B}P_b^n\mathbb{E}_t[\|\nabla_b f(\theta^{n,0})\|^2] + \frac{T\alpha^2 L\sigma^2}{\varepsilon} + \frac{T\alpha^3 C_1^2 L^2 C_0 G^2}{\varepsilon^2}$$

$$+ 2\alpha(1+C_1)G^2\sum_{b=1}^{B}P_b^n(\sum_{t=1}^{T}\mathbb{E}_t\left[\|\Delta\Gamma_b^{n,t}\|\right]) + \frac{2\alpha^2 C_1^2 LG^2}{\varepsilon} + \frac{\alpha^3 T^3 C_2}{4C_0}$$

$$\le -\frac{\alpha T}{4C_0}\pi\mathbb{E}_t[\|\nabla f(\theta^{n,0})\|^2] + \frac{T\alpha^2 L\sigma^2}{\varepsilon} + \frac{T\alpha^3 C_1^2 L^2 C_0 G^2}{\varepsilon^2}$$

$$+ 2\alpha(1+C_1)G^2\sum_{b=1}^{B}P_b^n(\sum_{t=1}^{T}\mathbb{E}_t\left[\|\Delta\Gamma_b^{n,t}\|\right]) + \frac{2\alpha^2 C_1^2 LG^2}{\varepsilon} + \frac{\alpha^3 T^3 C_2}{4C_0}$$

$$\leq -\frac{\alpha T}{4C_0}\pi\mathbb{E}_t[\|\nabla f(\theta^{n,0})\|^2] + \frac{T\alpha^2 L\sigma^2}{\varepsilon} + \frac{T\alpha^3 C_1^2 L^2 C_0 G^2}{\varepsilon^2}$$

$$+ 2\alpha(1+C_1)G^2\sum_{t=1}^{T}\mathbb{E}_t\left[\|\Delta\Gamma_{b_n}^{n,t}\|\right] + \frac{2\alpha^2 C_1^2 LG^2}{\varepsilon} + \frac{\alpha^3 T^3 C_2}{4C_0}$$

Where $b_n = \underset{b\in\{1,2,...,B\}}{arg\max}\left\{\sum_{t=1}^{T}\mathbb{E}_t\left[\|\Delta\Gamma_b^{n,t}\|\right]\right\}$, $\{P_b^n\}$ is the sampling probability of each block at $n$-th block epoch, the first inequality is because Corollary 2, the second inequality is because Corollary 1.

Let $D_0 = \frac{\pi}{4C_0}, D_1 = \frac{L\sigma^2}{\varepsilon}, D_2 = \frac{C_1^2 L^2 C_0 G^2}{\varepsilon^2}, D_3 = 2(1+C_1)G^2, D_4 = \frac{2C_1^2 LG^2}{\varepsilon}, D_5 = \frac{C_2}{4C_0}$, we obtain

$$\mathbb{E}_t[f(x^{n+1,0}) - f(x^{n,0})] \leq -\alpha T D_0\mathbb{E}_t[\|\nabla f(\theta^{n,0})\|^2] + T\alpha^2 D_1 + T\alpha^3 D_2 + \alpha D_3\sum_{t=1}^{T}\mathbb{E}_t\left[\|\Delta\Gamma_{b_n}^{n,t}\|\right]$$

$$+ D_4\alpha^2 + D_5\alpha^3 T^3$$

Updating the above equation for $N$ times, we obtain

$$\mathbb{E}_t[f(x^{N+1,0}) - f(x^{1,0})] \leq -\alpha T D_0\sum_{n=1}^{N}\mathbb{E}_t[\|\nabla f(\theta^{n,0})\|^2] + NT\alpha^2 D_1 + NT\alpha^3 D_2$$

$$+ \alpha D_3\sum_{n=1}^{N}\sum_{t=1}^{T}\mathbb{E}_t\left[\|\Delta\Gamma_{b_n}^{n,t}\|\right] + ND_4\alpha^2 + ND_5\alpha^3 T^3$$

From lemma 1, we know

$$\sum_{n=1}^{N}\sum_{t=1}^{T}\mathbb{E}_t\left[\|\Delta\Gamma_{b_n}^{n,t}\|\right] \leq \frac{2}{\sqrt{\varepsilon}}$$

Let $D_6 = -(f(x^*) - f(x^{1,0}))$, we obtain

$$\frac{\sum_{n=1}^{N}\mathbb{E}_t[\|\nabla f(\theta^{n,0})\|^2]}{N} \leq \frac{D_6}{\alpha NTD_0} + \frac{\alpha D_1}{D_0} + \frac{\alpha^2 D_2}{D_0} + \frac{2D_3}{NTD_0\sqrt{\varepsilon}} + \frac{\alpha D_4}{TD_0} + \frac{T^2\alpha^2 D_5}{D_0}$$

$$= \frac{D_6}{\alpha NTD_0} + \alpha\frac{TD_1 + D_4}{TD_0} + \alpha^2\frac{(D_2 + T^2 D_5)}{D_0}$$

Choosing $\alpha = \sqrt{\frac{D_6}{N(TD_1+D_4)}}$, we get the conclusion

$$\frac{\sum_{n=1}^{N}\mathbb{E}_t[\|\nabla f(\theta^{n,0})\|^2]}{N} \leq \frac{2\sqrt{D_6(TD_1+D_4)}}{\sqrt{N}TD_0} + \mathcal{O}(\frac{1}{N})$$

$$= \frac{1}{\sqrt{N}}\frac{4C_0\sqrt{\Delta_0[TL\sigma^2 + 2LG^2(\frac{\beta_1}{1-\beta_1})^2]}}{T\pi\sqrt{\varepsilon}} + \mathcal{O}(\frac{1}{N}).$$

$\square$

# F   Memory Analysis

The memory analysis presented in this study is conducted under frozen embedding layer and language modeling head layer configurations, with memory quantification comprehensively defined as the cumulative composition of parameters, gradients, optimizer states, and intermediate activation.

**Notation.** We suppose that the model follows the standard transformer architecture, where each layer consists of six modules, i.e., $W_Q \in \mathbb{R}^{h\times h}$, $W_K \in \mathbb{R}^{h\times h}$, $W_V \in \mathbb{R}^{h\times h}$, $W_O \in \mathbb{R}^{h\times h}$, $W_1 \in \mathbb{R}^{h\times 4h}$, $W_2 \in \mathbb{R}^{4h\times h}$. Let $L$ denote the number of the transformer layers, and $a$

denotes the number of attention heads. $v$ is the vocabulary size of the model. $s$ refer to the sequence length. $b$ represents the training batch size. The embedding representation dim is $h$.

Table 13: **Parameter dimension**

| Parameter | Property | Memory consumption |
|---|---|---|
| $X$ | Activation | $s * h * b = bsh$ |
| $W_Q$ | Model parameter | $a * (h * h/a) = h^2$ |
| $Q$ | Activation | $a * (s * h/a) * b = bsh$ |
| $W_K$ | Model parameter | $a * (h * h/a) = h^2$ |
| $K$ | Activation | $s * (h * h/a) * b = bsh$ |
| $W_V$ | Model parameter | $a * (h * h/a) = h^2$ |
| $V$ | Activation | $a * (s * h/a) * b = bsh$ |
| $S$ | Activation | $a * s * s * b = abs^2$ |
| $O$ | Activation | $a * (s * h/a) * b = bsh$ |
| $W_O$ | Model parameter | $a * (h * h/a) = h^2$ |
| $U$ | Activation | $s * h * b = bsh$ |
| $Z$ | Activation | $s * h * b = bsh$ |
| $Z_1$ | Activation | $s * 4h * b = 4bsh$ |
| $W_1$ | Model parameter | $h * 4h = 4h^2$ |
| $\mathbb{I}_{ZW_1+b_1>0}$ | Activation | $s * 4h * b = 4bsh$ |
| $W_2$ | Model parameter | $4h * h = 4h^2$ |

### F.1 Memory analysis of layer-wise optimization method

In this section, we will give detailed memory analysis for layer-wise optimization methods in backward propagation based on LLaMA's standard architecture.

**Lemma 3.** $\frac{\partial f}{\partial U} = g_1\left(\frac{\partial f}{\partial Z}, U\right), \quad \frac{\partial f}{\partial A} = g_2\left(\frac{\partial f}{\partial S}, S\right)$

*Where $Z = Layernorm(U)$, $S = softmax(A)$.*

*Proof.* First, we compute $\frac{\partial f}{\partial U}$.

The computational flow of $Z = LayerNorm(U)$ can be written as

$$\mu = \frac{1}{h} \sum_{i=1}^{h} U_{:,i}$$

$$\sigma = \sqrt{\frac{1}{h} \sum_{i=1}^{h} (U_{:,i} - \mu)^2 + \delta}$$

$$Z = \frac{U - \mu}{\sigma}$$

Taking the partial derivative of $U$, we obtain

$$\frac{\partial f}{\partial U_{ij}} = \frac{1}{\sigma_j} \left( \frac{\partial f}{\partial Z_{ij}} - \frac{1}{h} \sum_{k=1}^{h} \left\{ \frac{\partial f}{\partial Z_{ik}} - \frac{U_{ij} - \sigma_j}{\sigma_j^2} \frac{\partial f}{\partial Z_{ik}} (U_{ik} - \mu_j) \right\} \right)$$

We observe that the computation of $\frac{\partial f}{\partial U}$ depends solely on $\frac{\partial f}{\partial Z}$ and $U$. For notational convenience, we denote $\frac{\partial f}{\partial U}$ as $g_1\left(\frac{\partial f}{\partial Z}, U\right)$.

Second, we compute $\frac{\partial f}{\partial A}$.

Let $S_{ij} = \frac{e^{A_{ij}}}{\sum_{k=1}^{s} e^{A_{ik}}}$, we have

$$\frac{\partial S_{ij}}{\partial A_{mn}} = \begin{cases} 0, & if \ m \neq i \\ -\frac{e^{A_{im}} e^{A_{ij}}}{\left(\sum_{k=1}^{s} e^{A_{ik}}\right)^2} = -S_{in} S_{ij}, & if \ m = i \ and \ n \neq j \\ \frac{e^{A_{ij}} \left(\sum_{k \neq j} e^{A_{ik}}\right)}{\left(\sum_{k=1}^{s} e^{A_{ik}}\right)^2} = S_{ij} \left(1 - S_{ij}\right), & if \ m = i \ and \ n = j \end{cases}$$

We obtain

$$\frac{\partial f}{\partial A_{ij}} = \sum_{k \neq j} \frac{\partial f}{\partial S_{ik}} \left(-S_{ik} S_{ij}\right) + \frac{\partial f}{\partial S_{ij}} S_{ij} \left(1 - S_{ij}\right)$$

$$= S_{ij} \left( \frac{\partial f}{\partial S_{ij}} - \sum_{k=1}^{s} \frac{\partial f}{\partial S_{ik}} S_{ik} \right)$$

Which indicates

$$\frac{\partial f}{\partial A} = g_2 \left( \frac{\partial f}{\partial S}, S \right).$$

$\square$

Below, we present the forward and backward propagation flows of certain transformer layer.

**Forward**

$Q = XW_Q$

$K = XW_K$

$V = XW_V$

$A = \frac{QK^T}{\sqrt{d_k}}$

$S = softmax(A)$

$O = SV$

$Attn = OW_0$

$U = Attn + X$

$Z = LayerNorm(U)$

$Z_1 = ReLU(ZW_1 + b_1)$

$Z_2 = Z_1 W_2 + b_2$

**Backward**

$\frac{\partial f}{\partial W_Q} = X^T \frac{\partial f}{\partial Q}$

$\frac{\partial f}{\partial W_K} = X^T \frac{\partial f}{\partial K}$

$\frac{\partial f}{\partial X} = \frac{\partial f}{\partial V} W_V^T, \frac{\partial f}{\partial W_V} = X^T \frac{\partial f}{\partial V}$

$\frac{\partial f}{\partial Q} = \frac{\partial f}{\partial A} K^T \frac{1}{\sqrt{d_k}}, \frac{\partial f}{\partial K} = (\frac{\partial f}{\partial A})^T Q \frac{1}{\sqrt{d_k}}$

$\frac{\partial f}{\partial A} = g_2 \left( \frac{\partial f}{\partial S}, S \right)$

$\frac{\partial f}{\partial S} = \frac{\partial f}{\partial O} V^T, \frac{\partial f}{\partial V} = S^T \frac{\partial f}{\partial O}$

$\frac{\partial f}{\partial O} = \frac{\partial f}{\partial Attn} W_0^T, \frac{\partial f}{\partial W_0} = O^T \frac{\partial f}{\partial Attn}$

$\frac{\partial f}{\partial Attn} = \frac{\partial f}{\partial U}$

$\frac{\partial f}{\partial U} = g_1 \left( \frac{\partial f}{\partial Z}, U \right)$

$\frac{\partial f}{\partial Z} = \frac{\partial f}{\partial Z_1} \odot \mathbb{I}_{ZW_1 + b_1 > 0} W_1^T, \frac{\partial f}{\partial W_1} = Z^T \frac{\partial f}{\partial Z_1}$

$\frac{\partial f}{\partial Z_1} = \frac{\partial f}{\partial Z_2} W_2^T, \frac{\partial f}{\partial W_2} = Z_1^T \frac{\partial f}{\partial Z_2}$

Next, we will analyze layer-wise optimization methods' memory based on the forward and backward propagation flow of a transformer layer.

**Activation memory of frozen layers.** To enable backpropagation, the activation gradients of a Transformer layer must be stored, including the tensors $Q, V, K, S, U$ and $\mathbb{I}_{ZW_1 + b_1 > 0}$. The total activation memory is calculated as: $3bsh + abs^2 + 5bsh = abs^2 + 8bsh$.

**Activation memory of activated layers.** In the layer-wise optimization method, the weight matrices of unfrozen layers $W_Q, W_K, W_V, W_O, W_1, W_2$ require computation, necessitating the storage of $X, Q, K, V, O, S, U, \mathbb{I}_{ZW_1 + b_1 > 0}, Z$ and $Z_1$. The total activation memory here is: $5bsh + abs^2 + 10bsh = abs^2 + 15bsh$.

**Parameter memory of layer-wise optimization method.** Irrespective of which Transformer layer is activated, the model parameters must always be stored. This includes the weight matrices of a full

Transformer layer $W_Q, W_K, W_V, W_O, W_1, W_2$. The total parameter memory is: $L(4h^2 + 8h^2) = 12h^2L$.

**Optimizer state and gradient memory of frozen layer.** For frozen layers, neither optimizer states nor gradients need to be stored, resulting in zero memory usage for these components.

**Optimizer state and gradient memory of activated layer.** Activated layers require storing both gradients and optimizer states. The total memory for this is $3 \times 12h^2 = 36h^2$.

Table 14: **Memory of layer-wise optimization method**

| Layer State | Parameter Memory | Activation Memory | Optimizer State and Gradient Memory |
|---|---|---|---|
| Activated layer | $12h^2L$ | $abs^2 + 15bsh$ | $36h^2$ |
| Frozen layer | $12h^2L$ | $abs^2 + 8bsh$ | $0$ |

In practice, we assume that the activated layer's ID is $i$, we obtain the following conclusion

$$Parameter : 12h^2L$$
$$Optimizer\ State : 24h^2$$
$$Gradient : 12h^2$$
$$Activation : (L - i)(abs^2 + 8bsh) + (abs^2 + 15bsh)$$
$$Total : (L - i)(abs^2 + 8bsh) + (abs^2 + 15bsh) + 12h^2L + 36h^2$$

The rationale for the activation memory is as follows: when the activated layer ID is $i$, backpropagation only needs to propagate to the $i$-th layer. Consequently, the activations of layers $j = 1, 2, ..., i - 1$ can be saved.

Based on this, we derive the peak memory of the layer-wise method.
$$L(abs^2 + 8bsh) + 7bsh + 12h^2L + 36h^2.$$

## F.2 Memory analysis of MISA and LoRA

In this section, we focus on the scenario where only a specific module within a layer is activated at each step. Based on this, we derive a memory comparison between the module-wise optimization method and LoRA.

### F.2.1 Memory analysis of Module-wise BCD

Based on the computation flow described above, activating different individual modules incurs additional memory consumption compared to the module-wise optimization method.

Table 15: **Memory of Module-wise BCD**

| Unfrozen Module | Extra Storage | Extra Activation Memory | Extra Optimizer State And Gradient Memory |
|---|---|---|---|
| $W_Q$ | $X, W_Q$ | $bsh$ | $3h^2$ |
| $W_K$ | $X, W_K$ | $bsh$ | $3h^2$ |
| $W_V$ | $X, W_V$ | $bsh$ | $3h^2$ |
| $W_O$ | $O, W_O$ | $bsh$ | $3h^2$ |
| $W_1$ | $Z, W_1$ | $bsh$ | $12h^2$ |
| $W_2$ | $Z_1, W_2$ | $4bsh$ | $12h^2$ |

Let us consider the case where the activated module is $W_Q$ as an example. The memory analysis yields the following components:

- **Activation memory of frozen modules**

From the layer-wise memory analysis, we know that for a frozen layer, the activation memory requirement is $abs^2 + 8bsh$.

- **Activation memory of the activated module**
  To enable backpropagation through $W_Q$, we need to store the additional activation $X$, resulting in a total memory of $abs^2 + 9bsh$.

- **Parameter memory in layer-wise optimization**
  The layer-wise approach requires storing all parameters of each layer, amounting to $12h^2L$ memory.

- **Optimizer state and gradient memory for frozen layers**
  Similar to the layer-wise optimization case, the module-wise approach requires no additional memory for frozen layers.

- **Optimizer state and gradient memory for the activated layer**
  Only the optimizer states and gradients for $W_Q$ need to be stored, requiring $3h^2$ memory.

The peak memory consumption for the module-wise optimization method, when $W_Q$ is the activated module, is therefore:
$$L(abs^2 + 8bsh) + bsh + 12h^2L + 3h^2$$

### F.2.2   Memory analysis of LoRA

Consider a weight matrix $W_Q$ with LoRA adaptation, which can be expressed as:
$$W_Q = W_0 + B_Q A_Q, \quad \text{where} \quad B_Q \in \mathbb{R}^{h \times r}, \ A_Q \in \mathbb{R}^{r \times \frac{h}{a}}$$

The backward propagation gradients are computed as:
$$\frac{\partial \mathcal{L}}{\partial B_Q} = \frac{\partial \mathcal{L}}{\partial W_Q} A_Q^\top$$

$$\frac{\partial \mathcal{L}}{\partial A_Q} = B_Q^\top \frac{\partial \mathcal{L}}{\partial W_Q}$$

This formulation leads to the following memory requirements:

- **Embedding layer activations**: Requires storing parameters of size $sh$.
- **Adapter gradients and optimizer states**: For each layer's adapters $A_Q$ and $B_Q$, this requires $3aL \left( \frac{2hr}{a} \right) = 6hrL$ memory.
- **Full model parameters**: The base model parameters require $12h^2L$ memory.
- **Activation storage**: During backpropagation through $W_Q$, LoRA needs to store additional activations $X$, $A_Q$, and $B_Q$, requiring $L(abs^2 + 9bsh + 2hr)$ memory.

The peak memory consumption when applying LoRA to $W_Q$ is therefore:
$$L \left( abs^2 + 9bsh + 12h^2 + 8hr \right)$$

From the analysis above, we can similarly generalize the memory storage of different modules.

### F.3   Memory analysis of GaLore

Consider activating $W_Q$ as an example, GaLore maintains a projection matrix $P_Q \in \mathbb{R}^{h \times r}$ for $W_Q$ and optimizer states in the projected subspace of shape $r \times h$. It holds that:

- GaLore needs to store gradient, projection matrix and optimizer state of each layer's $W_Q$, which needs $4hrL$ memory storage.
- GaLore needs to store full model parameters, which needs $12h^2L$ memory storage.
- For backward propagation, GaLore needs to store extra activation $X$, leading to $L(abs^2 + 9bsh)$ activation memory in total.

We obtain the peak memory of GaLore when the activated module is $W_Q$:

$$L(abs^2 + 9bsh + 12h^2 + 4hr).$$

From the analysis above, we can similarly generalize the memory storage of different modules. Then we obtain the comparison between LoRA, GaLore and Module-wise BCD.

Table 16: **Peak memory comparison between LoRA, GaLore and Module-wise BCD**

| Activating Module | LoRA | GaLore | Modulewise-BCD |
|---|---|---|---|
| $W_Q$ | $L(abs^2 + 9bsh$ $+12h^2 + 8hr)$ | $L(abs^2 + 9bsh$ $+12h^2 + 4hr)$ | $L(abs^2 + 8bsh$ $+12h^2) + bsh + 3h^2$ |
| $W_K$ | $L(abs^2 + 9bsh$ $+12h^2 + 8hr)$ | $L(abs^2 + 9bsh$ $+12h^2 + 4hr)$ | $L(abs^2 + 8bsh$ $+12h^2) + bsh + 3h^2$ |
| $W_V$ | $L(abs^2 + 9bsh$ $+12h^2 + 8hr)$ | $L(abs^2 + 9bsh$ $+12h^2 + 4hr)$ | $L(abs^2 + 8bsh$ $+12h^2) + bsh + 3h^2$ |
| $W_O$ | $L(abs^2 + 9bsh$ $+12h^2 + 4hr)$ | $L(abs^2 + 9bsh$ $+12h^2 + 8hr)$ | $L(abs^2 + 8bsh$ $+12h^2) + bsh + 3h^2$ |
| $W_1$ | $L(abs^2 + 9bsh$ $+12h^2 + 20hr)$ | $L(abs^2 + 9bsh$ $+12h^2 + 13hr)$ | $L(abs^2 + 8bsh$ $+12h^2) + bsh + 12h^2$ |
| $W_2$ | $L(abs^2 + 12bsh$ $+12h^2 + 20hr)$ | $L(abs^2 + 12bsh$ $+12h^2 + 13hr)$ | $L(abs^2 + 8bsh$ $+12h^2) + 4bsh + 12h^2$ |
| All | $L(abs^2 + 15bsh$ $+12h^2 + 72hr)$ | $L(abs^2 + 15bsh$ $+12h^2 + 42hr)$ | $L(abs^2 + 8bsh$ $+12h^2) + 7bsh + 36h^2$ |

## F.4 Memory analysis of MISA

Let $x$, $y$, and $z$ denote the number of activated modules for $W_Q/W_K/W_V$, $W_1$, and $W_2$, respectively. Based on Table 15, we derive the memory estimation for MISA as follows:

- **Activation Memory**
  Compared to layer-wise optimization, MISA requires additional activation storage of:
  $$bshx + bshy + 4bshz$$

- **Parameter Memory**
  The full model parameters require:
  $$12h^2 L$$

- **Optimizer States and Gradients**
  The memory needed for storing optimizer states and gradients of activated modules is:
  $$3h^2 x + 12h^2 y + 12h^2 z$$

Given a trainable parameter ratio threshold $\delta$, where the number of trainable parameters is $12h^2 L\delta$, we formulate the optimization problem to determine the maximum memory consumption $M$:

$$
\begin{aligned}
\text{Constraint:} \quad & h^2 x + 4h^2(y + z) \leq 12h^2 L\delta \\
\text{Objective:} \quad & M = L(abs^2 + 8bsh + 12h^2) + bsh(x + y + 4z) \\
& + h^2(3x + 12y + 12z)
\end{aligned}
$$

The optimal solution yields the peak memory consumption of MISA:

$$L\left(abs^2 + 8bsh + 12h^2 + 12bsh\delta + 36h^2\delta\right) \tag{14}$$

## F.5 Memory analysis of MISA's indicators

In this subsection, we denote $B$ be the total number of modules.

According to equation 4, MISA needs to maintain $G_b^n$, $p_b^n$.

- **Memory of $G_b^n$:** For the frozen block $b_0$, $G_0^n$ follows the previous step. For the unfrozen block $b$, since its gradient has been saved, we only need to store $\frac{1}{T}\sum_{t=1}^{T}\|g_b^{n,t}\|^2$, which is negligible. Therefore, the total memory overhead is $O(B)$.

- **Memory of $p_b^n$:** According to the definition of $p_b^n$ in equation 4, the previous one $p_b^{n-1}$ can be discarded at $n$-th sampling period. Therefore, the memory occupation is $O(B)$ .

Therefore, the storage of the importance indicator $O(B)$ is negligible compared with the gradients, which is $O(h^2L)$.

**Memory efficiency even as model scales.** The memory overhead of storing the smoothed historical gradient norm $G_b$ is $O(B)$, which is negligible compared to the $O(h^2L)$ overhead of the gradients themselves (with a relative ratio of $O(1/h^2)$ when $L = O(B)$, as in LLaMA models). As model dimensions $h$ and $L$ increase, this ratio decays quadratically, making the overhead of storing $G_b$ negligible.

### F.6 Peak memory comparison between MISA and layer-wise method

**Lemma 4.** *Let $\delta$ be the trainable parameter ratio threshold, when $\delta < \frac{7bs+36h}{12bsL+36hL}$, MISA is more memory efficient than layer-wise method.*

*Proof.* From conclusion (14), we only need to prove

$$L(abs^2 + 8bsh + 12h^2 + 12bsh\delta + 36h^2\delta) < L(abs^2 + 8bsh + 12h^2) + 7bsh + 36h^2$$

$$\Longleftrightarrow$$

$$\delta < \frac{7bs + 36h}{12bsL + 36hL}$$

It's obvious that when $\delta < \frac{1}{L}$, memory of MISA is always smaller than layer-wise method. $\qquad\square$

### F.7 Peak memory comparison between layer-wise method and LoRA

**Lemma 5.** *As sequence length gets larger, layer-wise optimization method will become more memory efficient than LoRA and Galore. When $s > \frac{36h-72rL}{7bL-7b}$, peak memory of layer-wise optimization method is always smaller than LoRA and Galore.*

*Proof.* In this proof, we only consider the case when LoRA target and Galore target are all modules.

From table 16, when peak memory of layer-wise method is smaller than LoRA and Galore, we have

$$L\left(abs^2 + 8bsh + 12h^2\right) + 7bsh + 36h^2 < L\left(abs^2 + 15bsh + 12h^2 + 42hr\right)$$

$$\Longleftrightarrow$$

$$7bsh + 36h^2 < 7bshL + 42rhL$$

$$\Longleftrightarrow$$

$$36h - 42rL < s\left(7bL - 7b\right)$$

$$\Longleftrightarrow$$

$$s > \frac{36h - 42rL}{7bL - 7b}$$

$$\square$$

### F.8 Paras/Peak-Memory comparison between layer-wise method and LoRA

**Lemma 6.** *In the memory-consistent case, layer-wise optimization method can update more parameters than LoRA when the inequality holds $h > \frac{3rL}{2}$.*

*Proof.* From table16, we only need to prove

$$\frac{18rhL}{L\left(abs^2 + 15bsh + 12h^2 + 72rh\right)} < \frac{12h^2}{L\left(abs^2 + 8bsh + 12h^2\right) + 7bsh + 36h^2}$$

$$\iff$$

$$3r\left[L\left(abs^2 + 8bsh + 12h^2\right) + 7bsh + 36h^2\right] < 2h\left(abs^2 + 15bsh + 12h^2 + 72rh\right)$$

$$\iff$$

$$abs^2\left(2h - 3rL\right) + bsh\left[30h - 24rL - 21r\right] + h^2(36r + 24h - 36rL) > 0$$

When $r < \frac{2h}{3L}$ holds, each part of the last inequality is greater than 0, which completes the proof. $\quad\square$

## G  Computation Analysis

In this subsection, we adopt the same notation and setting as those in the memory analysis. It is well known that subspace optimization algorithms, such as LoRA, are not computationally efficient. Therefore, in this section, we will only analyze the computational complexity of layer-wise and module-wise optimization methods. Here we present the computation analysis in detail. We will analyze the exact floating point operations (FLOPs) required for **backward propagation** through a single Transformer layer. To begin with, we should remind that the matrices production with size $m \times n$ and $n \times r$ need $2mnr$ FLOPs.

From the forward and backward propagation flows and analysis in F.1, We can estimate the computational overhead under layer-wise setting.

Let:

- $b$: batch size
- $s$: sequence length
- $h$: hidden dimension
- $a$: number of attention heads
- $d = h/a$: per-head dimension

$$W_Q, W_K, W_V \in \mathbb{R}^{h \times h}, \quad W_O \in \mathbb{R}^{h \times h},$$
$$W_1 \in \mathbb{R}^{h \times 4h}, \quad W_2 \in \mathbb{R}^{4h \times h}$$

**Backward FLOPs per layer**

**1. Feed-Forward Network (FFN)**

$$\text{Grad } \partial W_2 : 2bs \cdot 4h \cdot h = 8bsh^2$$
$$\text{Grad } \partial Z_1 : 2bs \cdot h \cdot 4h = 8bsh^2$$
$$\text{Grad } \partial W_1 : 2bs \cdot h \cdot 4h = 8bsh^2$$
$$\text{ReLU mask} : 4bsh$$
$$\textbf{Total} : \boxed{24bsh^2 + 4bsh}$$

**2. Output Projection $W_O$**

$$\text{Grad } \partial W_O : 2bsh^2$$
$$\text{Grad } \partial O : 2bsh^2$$
$$\textbf{Total} : \boxed{4bsh^2}$$

**3. Q/K/V Projection**

$$\text{Each of } \partial W_Q, \partial W_K, \partial W_V : 2bsh^2$$
$$\textbf{Total} : \boxed{6bsh^2}$$

**4. Attention Score and Softmax**

$$QK^\top : 2bs^2h$$
$$\text{Grad w.r.t } Q, K : 2bs^2h$$
$$\text{Softmax backward} : 2bas^2$$
$$\textbf{Total} : \boxed{4bs^2h + 2bas^2}$$

**5. Attention Output Context (SV)**

$$S^\top \partial O : 2bs^2h$$
$$\partial SV^\top : 2bs^2h$$
$$\textbf{Total} : \boxed{4bs^2h}$$

**6. Layer-Norm Backward**

$$\boxed{10bsh}$$

## Total Backward FLOPs (One Layer)

$$\boxed{34bsh^2 + 8bs^2h + 2bas^2 + 14bsh}$$

### G.1 Computation analysis of layer-wise optimization methods

We assume only one layer is unfrozen at each iteration under a block coordinate descent (BCD) strategy.

**Activated layer**

For an activated layer, we cannot omit the computation cost of the entire layer during backpropagation.

Thus, the total FLOPs without weight gradient computations is:

$$\boxed{34bsh^2 + 8bs^2h + 2bas^2 + 14bsh}$$

**Frozen layer**

For a frozen layer, we can omit part of the computational cost of calculating the weight gradients during backpropagation.

We exclude the following terms:

- $W_Q, W_K, W_V$: $6bsh^2$
- $W_O$: $2bsh^2$
- $W_1, W_2$: $16bsh^2$

Thus, the total FLOPs without weight gradient computations is:

$$\boxed{10bsh^2 + 8bs^2h + 2bas^2 + 14bsh}$$

**Total computation cost**

$$\boxed{(L-1)(10bsh^2 + 8bs^2h + 2bas^2 + 14bsh) + 34bsh^2 + 8bs^2h + 2bas^2 + 14bsh}$$

### G.2 Computation analysis of module-wise optimization methods

Let the total number of activated modules $W_Q, W_K, W_V, W_O$ be $x$, number of modules $W_1, W_2$ be $y$, the expectation of computation cost be $C$, from analysis of G, we obtain the simple optimization Problem.

$$h^2 x + 4h^2 y \leq 12h^2 L\delta$$

$$C = L(10bsh^2 + 8bs^2h + 2bas^2 + 14bsh) + 2bsh^2 x + 8bsh^2 y$$

We obtain

$$\boxed{C_{max} = L(10bsh^2 + 8bs^2h + 2bas^2 + 14bsh) + 24bsh^2 L\delta}$$

It is obvious that when $\delta < \frac{1}{L}$, module-wise methods demonstrates greater computational efficiency compared to layer-wise optimization methods.

### G.3 Computation analysis of MISA's indicators

In implementation, we first update $G_b^n$ and then compute $p_b^n$ according to Equation (4).

- **Computation overhead of $G_b^n$:** The dominant term of the computation cost is the calculation for $\|g_b^{n,t}\|^2$, which needs $O(h^2)$ flops.

- **Computation overhead of $p_b^n$:** The computation overhead of $p_b^n$ only depends on the number of blocks $B$, which needs $O(B)$ flops.

Therefore, the computational overhead of the importance indicators $O(B+h^2)$ is negligible compared with the gradients, which is $O(bsh^2)$.

**Computational efficiency even as model scales.** The computation of the smoothed historical gradient norm $G_b$ introduces an overhead of $O(h^2)$. In contrast, the computational overhead for gradients is $O(bsh^2)$, yielding a relative ratio of $O(1/bs)$. As the model scales up, (i.e., with increasing $h$ and $L$), this ratio remains stable, making the overhead of computing $G_b$'s overhead negligible.

## H   Broader Impacts

This paper presents work that aims to advance the field of machine learning. The proposed method, MISA, is designed to efficiently train LLMs with low memory consumption. This will facilitate further research regarding memory-efficient LLM training in the future. We believe that this work will not cause significant societal consequences.

## I   Experimental Hyperparameters

For all baselines and our proposed method, we conducted extensive hyperparameter searches. The learning rate was searched in {2e-4, 1e-4, 5e-5, 1e-5, 5e-6, 3e-6, 1e-6}. For LoRA and GaLore, we explored ranks in {8, 16, 32}, and we found that a rank of 16 or 32 consistently yielded better performance than 8. For MISA, $\eta$ was searched in {0.1, 0.5, 1}. The table below presents the optimal hyperparameter settings.

The batch size in the following tables is represented as micro batch size $\times$ gradient accumulation.

## I.1 Hyperparameters for commonsense reasoning

| Hyperparameters | LLaMA3-8B | | Qwen2.5-7B | |
|---|---|---|---|---|
| | LoRA | DoRA | LoRA | DoRA |
| Rank | 32 | 16 | 32 | 16 |
| $\alpha$ | 64 | 32 | 64 | 32 |
| Dropout | | 0.05 | | |
| Optimizer | | AdamW | | |
| Learning rate | 1e-4 | 1e-4 | 2e-4 | 2e-4 |
| Batch size | | 4×4 | | |
| Warmup Steps | | 100 | | |
| Epochs | | 3 | | |
| Target module | | $W_q, W_k, W_v, W_{up}, W_{down}$ | | |

Table 17: Hyperparameters of LoRA and DoRA in commonsense reasoning tasks.

| Hyperparameters | LLaMA3-8B | | | Qwen2.5-7B | | |
|---|---|---|---|---|---|---|
| | MISA | BAdam | LISA | MISA | BAdam | LISA |
| Optimizer | | | AdamW | | | |
| Learning rate | 1e-5 | 5e-6 | 3e-6 | 5e-6 | 5e-6 | 3e-6 |
| Batch size | | | 4×4 | | | |
| Warmup Steps | | | 0 | | | |
| Activated parameters | 1%, 3% | 1 layer | 1 layer | 1%, 3% | 1 layer | 1 layer |
| MISA's $\eta$ | 1 | - | - | 1 | - | - |
| T | | | 50 | | | |
| Epochs | | | 3 | | | |

Table 18: Hyperparameters of MISA, BAdam and LISA in commonsense reasoning tasks.

## I.2 Hyperparameters for math reasoning

| Hyperparameters | LLaMA3-8B | | Qwen2.5-7B | |
|---|---|---|---|---|
| | LoRA | DoRA | LoRA | DoRA |
| Rank | 32 | 32 | 32 | 32 |
| $\alpha$ | | 32 | | |
| Dropout | | 0.05 | | |
| Optimizer | | AdamW | | |
| Learning rate | 2e-4 | 2e-4 | 1e-4 | 1e-4 |
| Batch size | | 4×1 | | |
| Warmup Steps | | 0 | | |
| Epochs | | 3 | | |
| Target module | | $W_q, W_v, W_{up}, W_{down}$ | | |

Table 19: Hyperparameters of LoRA and DoRA in math reasoning tasks.

| Hyperparameters | LLaMA3-8B | | | Qwen2.5-7B | | |
|---|---|---|---|---|---|---|
| | MISA | BAdam | LISA | MISA | BAdam | LISA |
| Optimizer | | | AdamW | | | |
| Learning rate | 5e-6 | 5e-6 | 5e-6 | 1e-5 | 1e-5 | 5e-6 |
| Batch size | | | 4×1 | | | |
| Warmup Steps | | | 0 | | | |
| Activated parameters | 1%, 3% | 1 layer | 1 layer | 1%, 3% | 1 layer | 1 layer |
| MISA's $\eta$ | 0.5 | - | - | 1 | - | - |
| T | | | 50 | | | |
| Epochs | | | 3 | | | |

Table 20: Hyperparameters of MISA, BAdam and LISA in math reasoning tasks.

## I.3 Hyperparameters for instruction fine-tuning

| Hyperparameters | LLaMA2-7B | | Mistral-7B | | Tiny-LLaMA | |
|---|---|---|---|---|---|---|
| | LoRA | GaLore | LoRA | GaLore | LoRA | GaLore |
| Rank | 32 | 32 | 32 | 32 | 32 | 32 |
| $\alpha$ | 64 | 64 | 64 | 64 | 64 | 64 |
| Dropout | 0.05 | 0 | 0.05 | 0 | 0.05 | 0 |
| Optimizer | | | AdamW | | | |
| Learning rate | 2e-4 | 3e-6 | 1e-4 | 1e-6 | 2e-4 | 1e-5 |
| Batch size | | | 2×8 | | | |
| Warmup Steps | | | 0 | | | |
| Epochs | | | 3 | | | |
| Target module | | | $W_q, W_k, W_v, W_o, W_{up}, W_{gate}, W_{down}$ | | | |

Table 21: Hyperparameters of LoRA and DoRA on Alpaca GPT4.

| Hyperparameters | LLaMA2-7B | | | Mistral-7B | | | Tiny-LLaMA | | |
|---|---|---|---|---|---|---|---|---|---|
| | MISA | BAdam | LISA | MISA | BAdam | LISA | MISA | BAdam | LISA |
| Optimizer | | | | AdamW | | | | | |
| Learning rate | 1e-5 | 1e-5 | 5e-6 | 1e-5 | 5e-6 | 1e-5 | 5e-5 | 1e-5 | 1e-5 |
| Batch size | | | | 2×8 | | | | | |
| Warmup Steps | | | | 0 | | | | | |
| Activated parameters | 3% | 1 layer | 1 layer | 3% | 1 layer | 1 layer | 4.545% | 1 layer | 1 layer |
| MISA's $\eta$ | 0.5 | - | - | 1 | - | - | 0.5 | - | - |
| T | | | | 50 | | | | | |
| Epochs | | | | 3 | | | | | |

Table 22: Hyperparameters of MISA, BAdam and LISA on Alpaca GPT4.

## I.4 Hyperparameters for pre-training

| Hyperparameters | MISA |
|---|---|
| Optimizer | AdamW |
| Learning rate | 1e-3 |
| Batch size | 32×256 |
| Warmup Steps | 0 |
| MISA's $\delta$ | 3%, 25% |
| MISA's $\eta$ | 300 |
| T | 50 |
| Sequence Length | 256 |
| Training Steps | 52000 |

Table 23: Hyperparameters of MISA on pre-training LLaMA2 350M.

| Hyperparameters | GaLore |
|---|---|
| Optimizer | AdamW |
| Learning rate | 1e-3 |
| Batch size | 32×256 |
| Warmup Steps | 0 |
| GaLore's rank | 32, 256 |
| GaLore's $\alpha$ | 1 |
| T | 50 |
| Sequence Length | 256 |
| Training Steps | 52000 |

Table 24: Hyperparameters of GaLore on pre-training LLaMA2 350M.

