# OpenReview forum: "MISA: Memory-Efficient LLMs Optimization with Module-wise Importance Sampling"
_NeurIPS.cc/2025/Conference — NeurIPS 2025 poster_

### Official Review · Reviewer_ov6T · 2025-06-30

**Clarity:** 3
**Significance:** 2
**Originality:** 2
**Rating:** 4
**Confidence:** 2

**Summary:**

The paper introduces MISA (Module-wise Importance SAmpling) - a memory efficient Transformer optimization scheme. In the method, all model’s components (Q, K, V, O, W_up, W_down, W_gate matrices in all layers) are assigned a uniform probability of being sampled for training. After sampling, the selected components are optimized for a preset number of optimization steps, and after that, their probability of being sampled is updated based on the gradient in the component. This sample-and-train cycle occurs a preset number of times. The memory savings in the method are due to the fact that only the optimizer parameters of the currently selected blocks need to be stored in VRAM.
The method convergence rate is shown via theoretical analysis.
The method’s efficacy is measured in both fine-tuning and pretraining and against numerous baseline methods (LoRA, DoRA, LISA, BAdam, IST).

**Questions:**

1. What is the tradeoff between T and N in Algorithm 1, i.e. if the total number of steps (T*N) is fixed, how could one approach choosing the appropriate T and N?
1. Have the authors considered a constrained scenario in which only some of the blocks are modified, resulting (similarly to LoRA) with a diff that is smaller than the whole model?
1. Have the authors explored the variance in the results inherent due to the sampling mechanism used? Can this stochastic element be eliminated from the algorithm?
1. Why do LoRA and MISA differ in peak memory usage for different context lengths (Figure 2).

**Ethical Concerns:**

["NO or VERY MINOR ethics concerns only"]

**Final Justification:**

I maintain my positive score. The proposed method is a valuable contribution to the community and is proven to work via sufficient experiments. Furthermore, the authors have addressed my concerns and questions thoroughly.
At the same time, the scale of improvements is not groundbreaking, which is hard for me to assign higher score.

**Limitations:**

yes

**Quality:**

3

**Strengths And Weaknesses:**

Strengths:
1. Extensive experiments and ablations
1. The paper is easy to read

Weaknesses:
1. The proposed method (unlike e.g. LoRA) does not save storage space needed for the storing the diff
1. No experiments for modalities different than text
1. No mentioning of non-Transformer architectures (even in Limitations)
1. Very small scale (especially the number of tokens) in the pretraining experiments

---

> ### Author Rebuttal · Authors · 2025-07-31
>
> We are grateful for the reviewer's insightful feedback and constructive comments. We address each point in detail below.
>
> **Question 1. Trade-off between $T$ and $N$** We conduct an ablation study on the effect of $T$ when $T\times N$ is fixed to 9650, as shown in the table below. We found that different choices of $T$ does not affect the convergence of online training loss, which is consistent with the observations reported in the BAdam and LISA.
>
> |T|5|15|30|50|100|200|500|
> |-|-|-|-|-|-|-|-|
> |Validation Loss($\downarrow$)|0.877|0.874|0.871|0.873|0.877|0.881|0.879|
> |MMLU($\uparrow$)|46.22|46.23|46.17|46.27|46.19|46.01|45.89 |
>
> **Question 2. Fine-tune a small set of blocks with MISA**
>
> The reviewer is correct that MISA is a memory-efficient method for full-parameter fine-tuning. This approach aims to match or exceed the performance of standard full fine-tuning, while producing a full model rather than a small, separate adapter.
>
> However, the core principles of MISA are flexible and can be directly applied to the "constrained scenario" the reviewer proposes. Our paper already contains two key experiments that validate the viability and power of using MISA to create or enhance parameter-efficient adapters:
>
>  1. **Creating novel "module-adapters" (Appendix D.5):** We explored a simplified version of the reviewer's idea in our ablation study. As shown in Figure 10, we fine-tuned only a single module type (e.g., only $W_q$ across all layers) while keeping the rest of the model frozen. The significant performance gains observed in this experiment prove that fine-tuning a small, targeted subset of modules is a viable and effective strategy. This validates the core concept of creating a "module-adapter" that would result in a small, storage-efficient "diff" file. To address this, we conducted comparative experiments under three settings: (1) MISA fine-tuning selected modules, (2) MISA fine-tuning all modules, and (3) LoRA fine-tuning all modules. The results are shown in the following table. Since fully fine-tuning all modules is not always optimal for LoRA, its reported $W_{all}$ corresponds to fine-tuning only $W_q$, $W_v$, $W_{up}$, and $W_{down}$ (which yields the best performance), while MISA's $W_{all}$ refers to fine-tuning all modules.
>
> |Method|$W\_v$|$W\_o$|$W\_{\text{gate}}$|$W\_{\text{up}}$ |$W\_{\text{down}}$|$W\_{\text{all}}$|
> |-|-|-|-|-|-|-|
> |MISA|70.5|71.7|71.5|72.5|71.7|**73.6**|
> |LoRA|69.9|71.2|71.5|71.1|70.9|71.9|
>
> As shown in the table, even when only specific individual modules are fine-tuned, MISA still achieves competitive performance compared to LoRA.
>
>  2. **Enhancing existing LoRA adapters with MISA (Appendix C.2):** We can also apply MISA's principles directly to LoRA. In our LoRA+MISA hybrid approach, we partition the LoRA matrices into modules, and use MISA's importance sampling to activate only a fraction of them during training. As shown in Figure 6, this method achieves higher accuracy than standard LoRA while saving ~8% memory. Crucially, the final output is still a standard LoRA adapter, demonstrating that MISA can create more performant adapters with greater training efficiency.
>
> Together, these results confirm the value of the reviewer's suggestion. The MISA framework is not only powerful for full-parameter tuning but can also be used to create novel, efficient "module-adapters" or enhance existing adapter methods.
>
> **Question 3. Variance due to sampling mechanism**
>
> We agree that the stochastic sampling mechanism will inherently introduce some variance across different training runs. To ensure the reliability of our findings, we report MISA's results on LLaMA2-7B from multiple runs with different random seeds. The table below illustrates the averaged performance and standard deviation.
> |Method|MMLU|MMLU-pro|MT-Bench|
> |-|-|-|-|
> |LISA|46.21 $\pm$ 0.12|20.85 $\pm$ 0.09|4.94 $\pm$ 0.14|
> |LoRA|45.50 $\pm$ 0.07|20.57 $\pm$ 0.08|4.45 $\pm$ 0.15|
> |MISA|46.27 $\pm$ 0.09|20.69 $\pm$ 0.16|5.13 $\pm$ 0.22|
>
> To validate that the stochasticity within our proposed importance sampling is crucial to MISA's performance, we conducted a new experiment comparing our method against a deterministic, cyclic sampling baseline. The results presented in the table below show that MISA with stochastic importance sampling significantly outperforms the deterministic approach, confirming that the stochastic element is essential and cannot be removed without impairing performance. Math and commonsense reasoning tasks are conducted on LLaMA3-8B.
> |Method|MMLU|MMLU-pro|MT-Bench|Math Reasoning|Commonsense Reasoning|
> |-|-|-|-|-|-|
> |Ours |46.27|20.69|5.13|73.6|86.6|
> |Cyclic|46.13|20.64|4.89|72.1|85.7|
>
> **Question 4. Peak memory comparison between LoRA and MISA.**
>
> The difference in peak memory usage between LoRA and MISA for varying context lengths (Figure 2) stems from their **distinct approaches to activation memory management**. LoRA, while introducing lightweight adapters per layer, does not reduce overall activation memory; it must store full activation values for backpropagation across all layers (including activations of its specific adapters). In contrast, MISA freezes inactive modules, thereby **eliminating the need to store activations associated with their gradient computations**. As sequence length increases, activation memory becomes the dominant factor in total memory usage, and MISA’s ability to skip activations for frozen modules yields significantly lower memory overhead compared to LoRA.
> We assume the number of layers is $L$, the number of attention heads is $a$, batch size is $b$, sequence length is $s$, dimension of hidden states is $h$, LoRA rank is $r$, and the threshold of trainable parameter ratio is $\delta$. According to our detailed memory analysis in Appendix F, the memory comparison between LoRA and MISA is as follows:
> ||LoRA|MISA|
> |-|-|-|
> |Peak Memory |$L(abs^2 + 15bsh + 12h^2 + 72hr)$|$L(abs^2 + 8bsh + 12h^2 + 12bsh\delta + 36h^2\delta)$|
> |Activation memory|$L(abs^2+15bsh+18hr)$| $L(abs^2+8bsh+12bsh\delta)$|
>
> As shown in the Table above, a direct comparison of the formulas reveals that for long-text tasks, MISA's activation memory is guaranteed to be smaller than that of LoRA when $\delta < 7/12$. In practical scenarios, however, the value of $\delta$ generally stays below $10\\%$, which indicates that MISA boasts significantly higher memory efficiency than LoRA in long-text tasks.
>
> **Weakness 1. MISA does not lead to an adapter**
>
> The reviewer is correct that MISA is a memory-efficient method for full-parameter fine-tuning. This approach aims to match or exceed the performance of standard full fine-tuning, while producing a full model rather than a small, separate adapter. However, the core principles of MISA are flexible and can be directly applied to the "constrained scenario" the reviewer proposes.
> - When MISA is employed to fine-tune only a small subset of (or even a single) modules, it results in what we term "module-adapters." This produces an adapter with parameters comparable to LoRA, as further detailed in our response to Question 2.
> - Furthermore, MISA can be used to enhance LoRA adapters. This MISA + LoRA hybrid approach achieves "LoRA-adapter" functionality while simultaneously saving memory and incurring improved performance, see our response to Question 2.
>
> **Weakness 2. Modality model**
>
> Thank you for highlighting this important point. We acknowledge that our current work focuses exclusively on text modalities and does not include experiments on other modalities. This is indeed a limitation of the present study, which we will explicitly clarify in the limitation section of the paper. We promise to explore multi-modality applications (e.g., image-text, audio-text) as part in the camera-ready revision.
>
> **Weakness 3. Non-Transformer model**
>
> Thank you for pointing out this important oversight. We will clearly state in the limitations that our method is only evaluated on Transformer-based models, and its applicability to non-Transformer architectures is not addressed. **Moreover**, we will highlight exploring non-Transformer frameworks as a key direction for future work.
>
> **Weakness 4. Small scale experiment**
>
> In Section 5 of the manuscript, we have acknowledged the limitations in pretraining due to resource constraints. While not at the 7B+ scale, our pre-training results (Figure 4 and Table 6) on 130M and 350M models provide valuable proof-of-concept. Furthermore, we include an additional pre-training experiment on a 1B model using 13.1B tokens to further demonstrate MISA's utility, shown in the following table. Most importantly, our work's primary focus is on memory-efficient fine-tuning, where MISA's effectiveness and scalability have been extensively validated on large models such as LLaMA3-8B, Qwen2.5-7B, Mistral-7B, and LLaMA3-70B.
>
> |LLaMA 1B|Adam|GaLore(r=512)|MISA($\delta = 25\\%$)
> |-|-|-|-|
> |Validation perplexity($\downarrow$)|15.56|15.64|**15.27**|
> ---
>
> We sincerely thank the reviewer again for their thoughtful comments. We hope the above responses adequately address their concerns and are happy to provide further clarification if needed.

---

> > ### Comment · Reviewer_ov6T · 2025-08-05
> >
> > I would like to thank the authors for their thourough reply. I will maintain my score.

---

> > > ### Author Response · Authors · 2025-08-06
> > >
> > > Thank you so much for your positive feedback on our rebuttal and for maintaining the score. We truly appreciate your careful consideration and the time you've invested in reviewing our work. Your insights and this constructive interaction are really helpful for us to improve the paper. If there's any further discussion needed, we're always ready. Thanks again for your support!

---

### Official Review · Reviewer_HSph · 2025-06-30

**Clarity:** 4
**Significance:** 4
**Originality:** 3
**Rating:** 6
**Confidence:** 4

**Summary:**

The authors propose a new efficient optimization method based on module-level importance sampling. The key idea is to update each module probabilistically based on the computation budget and the importance score which comes from the gradient norm. The authors demonstrate empirically that models can be trained with a fraction of the memory and computation of full model training while achieving competitive performance. A theoretical convergence guarantee is also supplied.

**Questions:**

I found the figure 7 intriguing. Is a similar observation true in full model training, namely that cleaning optimizer state improves performance early in training?

Also is there some intuition for why gradient norm should be divided by the square-root of the number of parameters? It might make sense to have different normalizations in different layers, as per [1].

What is the impact of the number of inner iterations $T$? How sensitive is this hyperparameter and what is the rule of thumb for setting it in practice?

1. A Spectral Condition for Feature Learning (2024) by Yang et al.

**Ethical Concerns:**

["NO or VERY MINOR ethics concerns only"]

**Limitations:**

Yes

**Quality:**

4

**Strengths And Weaknesses:**

The paper is very clear with extensive experiments and ablations. The idea is simple and appears to be scalable. A detailed analysis of the computational and memory requirements are provided with comparisons to many popular baselines like LoRA.

---

> ### Author Rebuttal · Authors · 2025-07-31
>
> We are grateful for the reviewer's insightful feedback and constructive comments. We address each point in detail below.
>
> **Q1. Cleaning optimizer state.**
>
> We thank the reviewer for this very insightful question regarding our observation in Figure 7. We believe this effect is specific to the block-coordinate nature of MISA and would likely not be beneficial in standard full-model training.
>
> Our conjecture for why clearing the optimizer state helps in MISA's pre-training is based on the concept of "stale" momentum.
> - In MISA's block-coordinate framework, a module is updated and then frozen. It may not be selected for an update again for a significant number of iterations. During this long interval, the other active parts of the model change substantially, shifting the overall loss landscape.
> - When the module is eventually re-activated, its stored momentum from many iterations prior is now "stale"—it was calculated based on a very "old" model state. Reintroducing this outdated momentum could provide a poor estimate of the current gradient's direction and potentially hinder optimization.
> - Therefore, clearing the optimizer state acts as a momentum reset for the newly activated block. This forces the optimizer to re-evaluate the gradient direction and track it tightly, which can be beneficial for model training.
>
> This contrasts sharply with standard full-model training (e.g., with Adam):
> - In standard training, all model parameters and their corresponding optimizer states (first and second moments) are updated in every single iteration.
> - The momentum is therefore always "in-time" and provides a valuable, continuously updated estimate of the gradient's trajectory.
> - In this context, clearing the optimizer state would be detrimental, as it would discard this crucial historical information and disrupt the convergence path.
>
> In summary, we believe the benefit observed in Figure 7 is a unique characteristic of block-wise update methods, where optimizer states for inactive blocks can become outdated.
>
> **Q2.1 Scaled gradient norm.**
>
> We thank the reviewer for this insightful question.
>
> The primary motivation for our normalization—dividing the gradient norm by the square root of the number of parameters (as defined in Appendix B.2)—is to create a scale-invariant importance metric.
>
> Without normalization, the gradient norm would be heavily biased towards parameter-dense modules (e.g., FFNs) over smaller ones (e.g., attention heads), simply because they have more parameters contributing to the norm's magnitude. Our chosen normalization effectively measures the average intensity of the gradient signal per parameter, allowing for a more equitable comparison between modules of vastly different sizes. This ensures that our importance score reflects the relative significance of the update for a given module, rather than just its absolute size.
>
> **Q2.2 Different normalizations in different layers.**
>
> We agree with the reviewer that a single, global normalization scheme may not be optimal, and we thank you for pointing us to the excellent work by Yang et al. [1]. The idea of using more sophisticated, layer-specific normalization is indeed valuable. This could involve adapting the normalization based on a module's depth, its functional role (e.g., attention vs. FFN), or other structural properties. We consider this a promising direction for future research and will cite and discuss this work in our revised paper. Thank you again for the inspiring suggestion.
>
> **Q3. Sensitivity to inner iteration $T$.** We conduct an ablation study on the effect of $T$, as shown in the table below. We found that different choices of $T$ does not affect the convergence of online training loss, consistent with the observations reported in the BAdam and LISA.
>
> |T|5|15|30|50|100|200|500
> |-|-|-|-|-|-|-|-|
> |Validation Loss($\downarrow$)|0.877|0.874|0.871|0.873|0.877|0.881|0.879|
> |MMLU($\uparrow$)|46.22|46.23|46.17|46.27|46.19|46.01|45.89 |
>
> **Reference**
>
> [1] Yang, G., Simon, J. B., & Bernstein, J. (2024). A Spectral Condition for Feature Learning.
>
>
> ---
>
> We sincerely thank the reviewer again for their thoughtful comments. We hope the above responses adequately address their concerns and are happy to provide further clarification if needed.

---

> > ### Comment · Reviewer_HSph · 2025-08-06
> >
> > Thank you for the provided clarifications. I plan to maintain my positive score.

---

> > > ### Author Response · Authors · 2025-08-06
> > >
> > > Thank you so much for your positive feedback on our rebuttal and for maintaining the score. We truly appreciate your careful consideration and the time you've invested in reviewing our work. Your insights and this constructive interaction are really helpful for us to improve the paper. If there's any further discussion needed, we're always ready. Thanks again for your support!

---

### Official Review · Reviewer_CWR5 · 2025-07-02

**Clarity:** 1
**Significance:** 2
**Originality:** 3
**Rating:** 4
**Confidence:** 4

**Summary:**

In the proposed method, score based importance sampling is utilized for selecting modules for pretraining and fine-tuning. The authors provide convergence rate under the nonconvex stochastic optimization setting.

**Questions:**

1. In Proposition 2, the authors state that a solution to problem (2) is also a feasible solution to (5). Based on my understanding, the solution to the problem (2) would give the probabilities for the block. However, the solution to problem (5) would give probabilities for the modules. Therefore, solution to problem (2) is not sufficient to solve problem (5).

2. The authors also state that "LISA finds the embedding and LM-head layers to be very important, MISA does not train them in fine-tuning tasks because their parameters are too large." However, I observe that in many of the simulation results, MISA is performing better than LISA, why is that the case?

3. How are you deciding the value for delta in Algorithm 1?

4. In Experiments Section, the authors mention that "we combined the training data from all eight tasks into a single training set for fine-tuning." Why was this combination needed in this paper?

5. In Section 4.3, the authors should also include results with respect to techniques like LoRA to show the effectiveness of the proposed approach.

**Ethical Concerns:**

["NO or VERY MINOR ethics concerns only"]

**Final Justification:**

The authors have satisfactorily addressed my concerns, both theoretical and experimental. While some clarifications (especially around notation and algorithm indexing) should be better integrated into the main text, the core ideas are sound, and the method is well-motivated. I will accordingly increase my score.

**Limitations:**

Yes

**Paper Formatting Concerns:**

The format was followed.

**Quality:**

2

**Strengths And Weaknesses:**

Strengths

Module-wise optimization is motivated by demonstrating that smaller modules of parameters preserve more gradient information. Moreover, this strategy eliminates the need for storing the parameters of an entire layer. Furthermore, the derive the sampling probabilities for the modules as a function of the gradient norm and the convergence rate.

Weaknesses

1. Although the idea of module based optimization is interesting, the presentation lacks clarity. The authors keeping switching between the use of terms like layers, blocks, and modules which makes it confusing to follow. For example, the description in Section 2.3 uses the term layers but Proposition 2 uses modules in the blocks. Moreover, the use of the terms layers and blocks are changed across sections without consistency. This dilutes the impact of the main contributions of the paper.

2. Algorithm 1 is also confusing because as Steps 5-13 are for the modules within a block but which block is being selected is not shown. Steps 5-13 should also depend on the block index.

3. There is inconsistency in the notations as well. For examples, in Sec 2.3, the authors mention that each layer has distinct internal modules where each module is a matrix parameter. However, in Proposition 2, the authors use $\theta_b=(\theta_{b,1},\theta_{b,2},\ldots,\theta_{b,K})$, that encompasses the module parameters, to denote the parameters of a block (assuming this is same as a layer) which is a vector.

4. The paper lacks adequate experiments. For example, techniques like LoRA should also be considered for instruction tuning experiments. Moreover, in several experiments, it is not clear why MISA performs better than LISA in terms of the accuracy.

---

> ### Author Rebuttal · Authors · 2025-07-31
>
> We are grateful for the reviewer's insightful feedback and constructive comments. We address each point in detail below.
>
> **W1. Terminology.** We sincerely thank the reviewer for highlighting this point. Our definitions are:
> - **Layer**: A standard transformer layer (e.g., the entire multi-head attention and FFN). This is the coarse-grained unit of optimization in prior work .
> - **Module**: Our proposed fine-grained optimization unit. A module can be regarded as a component inside the standard transformer layer. For instance, within the multi-head attention mechanism, we identify four such modules: $W_q$, $W_k$, $W_v$, and $W_o$. Similarly, in the feed-forward network, we define two modules: $W_{up}$ and $W_{down}$. **These internal components collectively constitute a transformer layer**.
> - **Block**: The general term for the unit of optimization in the Block Coordinate Descent (BCD) framework. a block typically refers to an entire layer. However, in our work, a block can be either a single module or a collection of several modules.
>
> With these definitions, the writing in Section 2.3 is **unambiguous**, which motivates our shift from a "layer-as-a-block" paradigm to a "module-as-a-block" paradigm. Proposition 2 further demonstrates that using modules as the optimization unit is superior to using layers. In the revision, we will add explicit definitions for "layer," "block," and "module," clarifying their interrelationships and ensuring global consistency throughout the manuscript.
>
> **W2. Algorithm 1.** $\tau_n$ is the block selection index for step $n$. In Step 5 of Algorithm 1, all selected modules constitute the block $\tau_n$. Steps 5-13 of Algorithm 1 then operate on the modules within the block specified by $\tau_n$. For detailed information on block selection, please refer to Algorithm 2 in Appendix B.1.
>
> **W3. Notation.** We'd like to clarify that there's no inconsistency in our notation. Vectorizing module matrices into vectors follows a standard convention in optimization theory.
>
> - While modules are intuitively described as architectural matrices (e.g., the weight matrix $W_q$ in Section 2.3), it is common practice in optimization literature to $\textbf{vectorize these parameters for mathematical analysis}$. This simplification streamlines the notation for gradients and convergence proofs. For instance, most stochastic gradient descent convergence proofs in literature utilize vector representations, even though neural network weights are typically in matrix form.
>
> - In the specific case of Proposition 2, we employ this convention to illustrate the benefits of our fine-grained block-coordinate update. Here, $\theta_b$ represents the parameters of a single layer, which is then decomposed into its constituent modules, denoted as $\theta_b = (\theta_{b,1}, \dots, \theta_{b,K})$. Each $\theta_{b,j}$ corresponds to the vectorized form of that module's matrix parameters. The primary purpose of this proposition is to formally demonstrate that optimizing over these finer-grained module-blocks is a superior strategy compared to optimizing over the coarse layer-block.
>
> **W4.1 Comparison with LoRA.** We would like to respectfully clarify that we **did include LoRA** and other key baselines in our instruction tuning experiments. The results of this comparison are presented in **Table 5 on page 9**. Our results demonstrate that MISA consistently outperforms or performs competitively against LoRA and other state-of-the-art baselines on three different models (TinyLLaMA, LLaMA2-7B, and Mistral-7B) for instruction tuning.
>
> **W4.2 Comparison with LISA.** MISA outperforms LISA for two fundamental reasons.
>  - **Module-wise partitioning.** LISA employs a coarse "layer-as-a-block" approach, treating all components within a transformer layer as homogeneous. In contrast, MISA recognizes **the inherent heterogeneity of these components**, underscoring the necessity of a finer "module-as-a-block" approach. Proposition 2 provides the theoretical justification for this, proving that our finer-grained partitioning enables a superior optimization step compared to any layer-wise strategy.
>  - **Adaptive importance sampling.** While LISA keeps the embedding and LM-head layers active, it samples the transformer layers—which account for the majority of parameters—uniformly at random. This non-adaptive strategy can be inefficient. MISA, however, employs an adaptive importance sampling mechanism (governed by Eq.(4) in the manuscript) that dynamically prioritizes modules with higher utility. This intelligent allocation of computational budget **allows MISA to focus on the most impactful modules** at each step, leading to better convergence.
>
> **Q1. Proposition 2.** The reviewer is correct that a probability distribution over layers, $p_b$, is not directly a probability distribution over modules, $p_{b,j}$. Our statement that "a solution to problem (2) is also a feasible solution to (5)" is meant to convey that any layer-wise sampling strategy can be represented as a valid instance of a module-wise sampling strategy. Let's make this mapping explicit:
>  1. A layer-wise sampling strategy, which is the solution to problem (2), gives a probability $p_b$ for sampling each layer $b$. Implicitly, when layer $b$ is chosen, all $K$ of its modules are updated.
>  2. We can construct an equivalent module-wise probability distribution by distributing the layer's probability uniformly across its modules. That is, we can set the module sampling probability $p_{b,j} = p_b / K$ for all modules $j=1,...,K$ within layer $b$.
>  3. This constructed distribution ${p_{b,j}}$ is a feasible solution for problem (5) because it satisfies the necessary constraints:
>     - $p_{b,j} >= 0$ since $p_b >= 0$.
>     - The probabilities sum to one: $\sum_{b=1}^B\sum_{j=1}^K p_{bj} = \sum_{b=1}^B\sum_{j=1}^K p_{b}/K = \sum_{b=1}^B p_b = 1$
>
> The core purpose of Proposition 2 is to show that the optimization space for module-wise sampling (problem 5) is a superset of the strategies considered by layer-wise sampling (problem 2). Because the layer-wise approach represents just one possible (and likely suboptimal) feasible solution within the module-wise framework, the optimal solution found by MISA in the space of problem (5) is guaranteed to be at least as good as, or better than, the optimal layer-wise solution of problem (2).
>
> **Q2. Comparison with LISA.** Our central finding is that **how** the transformer blocks are fine-tuned is **far more critical** for downstream performance than **whether** the embedding and LM-head layers are updated. MISA's performance advantage stems directly from its two core contributions, which are absent in LISA: Fine-Grained Module Partitioning and Adaptive Importance Sampling (see our response to **W4.2**). We conduct the ablation study to validate these factors. As shown in the following table, using the same granularity as LISA (layer-wise) combined with our proposed importance sampling strategy already yields higher accuracy than LISA.
>
> |Method|Math|Commonsense|
> |-|-|-|
> |LISA|70.7|85.9|
> |Layer-wise + **Importance Sampling**|73.2|86.1|
> |**Module-wise** + Uniform Sampling|72.1|85.7|
> |**Module-wise+Importance Sampling(MISA)**|**73.6**|**86.6**|
>
> **Q3. Value of $\delta$.**  In MISA's experiments, we primarily choose the value of $\delta$ in Algorithm 1 to align with the memory usage of baselines. For instance, setting $\delta=3\\%$ ensures GPU memory consumption matches that of BAdam, while $\delta=1\\%$ achieves lower consumption than all baselines (see Tables 1, 3, 4, etc.). This choice effectively balances efficiency and performance by constraining the number of active modules within hardware memory limits. **In practice, $\delta$ adapts flexibly to the GPU memory constraints of the target hardware**, allowing users to adjust it based on their specific computational environment without compromising the core mechanism of importance sampling. This flexibility renders MISA robust across diverse deployment scenarios, from memory-constrained edge devices to high-end training clusters.
>
> **Q4. Data combination.** The combination of training data from all eight tasks into a single set for fine-tuning follows the experimental setup of LLM-Adapters[1] and DoRA[2]. This is a standard practice in multitask instruction tuning, designed to enable multitask generalization and ensure a fair comparison with baselines like LISA, LoRA, and GaLore under consistent conditions.
>
> **Q5. Comparison with LoRA.** We would like to respectfully clarify that we **did include LoRA** and other key baselines in our instruction tuning experiments. The results of this comparison are presented in **Table 5 on page 9**. Our results demonstrate that MISA consistently outperforms or performs competitively against LoRA and other state-of-the-art baselines on three different models (TinyLLaMA, LLaMA2-7B, and Mistral-7B) for instruction tuning.
>
> **Reference**
>
> [1]Hu Z, Wang L, Lan Y, et al. Llm-adapters: An adapter family for parameter-efficient fine-tuning of large language models[J]. EMNLP, 2023.
>
> [2]Liu S Y, Wang C Y, Yin H, et al. Dora: Weight-decomposed low-rank adaptation. ICML, 2024.
>
> ---
> We sincerely thank the reviewer again for their thoughtful comments. We hope the above responses adequately address their concerns and are happy to provide further clarification if needed.

---

> > ### Author Response · Authors · 2025-08-06
> > **Seeking Your Valuable Insights: A Follow - up on Paper 15661's Rebuttal**
> >
> > Dear reviewer CWR5，
> >
> > I hope this email finds you well.​
> >
> > First and foremost, I want to express my sincere gratitude once again for the time and effort you've already invested in reviewing our paper (ID: 15661). Your initial feedback was incredibly insightful, and I truly appreciate your commitment to the review process.​
> >
> > I understand that you must be extremely busy with multiple responsibilities, but I wanted to gently follow up regarding our rebuttal. We submitted it within the given timeframe, and since we haven't received any further comments or indications of remaining concerns from you, I'm writing to kindly inquire if you've had the opportunity to review it. Your perspective is of utmost importance to us, as it will help us refine our work and ensure it meets the high standards of this conference.​
> >
> > If there's anything we can do to assist you in reviewing our response more efficiently, please don't hesitate to let me know. I would be truly grateful if you could spare some time to share your thoughts with us at your earliest convenience.​Thank you again for your dedication to academic excellence and for your invaluable contributions to this review process. I look forward to hearing from you soon.
> >
> > Best regards,​
> >
> > The authors of paper 15661

---

> > ### Comment · Reviewer_CWR5 · 2025-08-07
> > **Official Comment by Reviewer CWR5**
> >
> > The authors have satisfactorily addressed my concerns, both theoretical and experimental. While some clarifications (especially around notation and algorithm indexing) should be better integrated into the main text, the core ideas are sound, and the method is well-motivated. I will accordingly increase my score.

---

> > > ### Author Response · Authors · 2025-08-07
> > >
> > > Thank you for your valuable feedback and kind recognition of our work. We will carefully integrate the clarifications on notation and algorithm indexing into the revision as suggested. Your insights have been instrumental in strengthening the manuscript, and we greatly appreciate your support.

---

### Official Review · Reviewer_ULBr · 2025-07-04

**Clarity:** 3
**Significance:** 3
**Originality:** 3
**Rating:** 4
**Confidence:** 3

**Summary:**

This paper proposes MISA (Module-wise Importance Sampling Algorithm), a novel and memory-efficient optimization method designed for full-parameter fine-tuning and pre-training of large language models (LLMs). Unlike prior layer-wise or low-rank adaptation approaches such as LoRA, MISA decomposes each transformer layer into smaller modules (e.g., attention heads, feed-forward components). It then computes gradient-based importance scores and samples these modules for update using a softmax-based strategy that balances importance and exploration. The method is theoretically grounded with convergence guarantees under non-convex stochastic settings and is supported by strong empirical performance across commonsense reasoning, mathematical problem solving, instruction tuning, and pre-training tasks.

**Questions:**

* In Algorithm 1 (Line 1) you state: "Partition the model into B modules (not layers)". However Section 3.1 briefly mentions using subcomponents such as Wq, Wk, Wv, Wo, Wup, and Wdown. However, the paper does not justify why this particular partitioning is optimal. Could you clarify how this module granularity was chosen, and whether alternative decompositions (e.g., more fine-grained or head-wise) were explored?
* How often do the module importance scores (i.e., gradient norms) change during training? Are the high-importance modules relatively stable, or do they vary significantly over time? A clearer understanding here would help assess whether recomputing them at every outer loop is computationally justified.
* How often do the module importance scores (i.e., gradient norms) change during training? Are the high-importance modules relatively stable, or do they vary significantly over time? A clearer understanding here would help assess whether recomputing them at every outer loop is computationally justified.
* Have you investigated how sensitive MISA is to the choice of gradient norm as the importance metric?
Specifically, could modules with consistently high variance but low long-term utility be over-sampled?

**Ethical Concerns:**

["NO or VERY MINOR ethics concerns only"]

**Final Justification:**

The authors addressed my main concerns regarding profiling, theoretical justification for the importance metric, and ablation studies on partitioning granularity during the rebuttal. Thus, I would like to increase my score.

**Limitations:**

* The method has only been tested on small-scale pretraining (130M and 350M models). It remains unclear how it performs during full-scale pretraining on 7B or 70B models, which is where memory constraints are most severe.
* The fixed module granularity may restrict adaptability across architectures or training regimes.
* While the paper provides per-step timing comparisons (Table 8), it does not isolate the time spent on MISA-specific operations (e.g., sampling, importance score computation, and multi-step inner updates). A more detailed breakdown of these components would help fully assess where MISA’s computational efficiency gains originate, especially as model size scales.

**Quality:**

3

**Strengths And Weaknesses:**

Strengths:

* The motivation is well-articulated. The paper clearly identifies memory bottlenecks in full fine-tuning and highlights the limitations of existing layer-wise or low-rank methods.
* The idea of fine-grained module-level optimization is compelling. Shifting from coarse layer updates to smaller modules allows for better control over memory and seems to preserve more task-relevant parameters.
* Theoretical contributions are strong. The authors provide a convergence guarantee even when using stochastic gradients, Adam, and repeated updates per module—a realistic and practically relevant setting.

Weaknesses:

* While Section C.3 provides per-step timing comparisons (Table 8) and shows that MISA is more computationally efficient than LoRA and significantly faster than GaLore, the breakdown does not include the cost of repeated module updates ($T$ inner steps) or the overhead from adaptive sampling. While overall step time is reasonable, it remains unclear how much of MISA’s time is spent specifically on its unique operations (e.g., importance computation, sampling, module-specific Adam updates).
* MISA requires maintaining a smoothed historical gradient norm ($\mathbf{G}_b$) for every module, which introduces additional memory and compute overhead—especially as the number of modules increases. Although the authors state that this overhead is negligible, they do not provide a quantitative breakdown (in memory footprint or compute usage), which is particularly important when scaling to larger models or when operating in mixed-precision, low-memory environments.
* The use of gradient norm as a proxy for module importance is somewhat heuristic. In practice, gradient magnitude may not always reflect the true contribution of a module to generalization, especially in overparameterized settings. This could lead to prioritizing noisy or spurious updates.
* The partitioning into modules (e.g., Wq, Wk, Wv, etc.) is static and hardcoded. There is no discussion or ablation on whether this granularity is optimal, nor is there any mechanism to adapt it dynamically based on model behavior during training.

---

> ### Author Rebuttal · Authors · 2025-07-31
>
> We are grateful for the reviewer's insightful feedback and constructive comments. We address each point in detail below.
>
> **W1. Detailed profiling.** The following table shows the detailed profiling, including all unique operations. MISA-related operations are performed every $T$ steps ($T=50$ by default). The table reports the average time consumption per step (in ms), showing that the overhead introduced by MISA is negligible compared to the forward and backward consumption. The profiling is conducted using Qwen2.5-7B on a single RTX PRO 6000.
>
> |Forward|Backward|Adam update|Sample modules|Track block gradient norm|Update sampling probability|Additional Momentum Update|Clear optimizer states|
> |-|-|-|-|-|-|-|-|
> |11.433|24.359|0.591|0.010|0.992|$\le10^{-5}$|0.012|0.002|
>
> **W2. Memory and compute overhead.** We define the hidden dimension as $h$, sequence length as $s$, batch size as $b$, and number of layers as $L$. Each layer has 7 modules, totaling $B=7L$ modules.
> - **Theoretical comparison.** Analyses in Appendices F.5 and G.3 show that computing and storing the smoothed historical gradient norm ($G_b$) incurs an $O(7L)$ memory overhead and $O(h^2)$ computational overhead. In comparison, the memory and computational overheads for gradients are $O(7Lh^2)$ and $O(bsh^2)$, respectively. This results in relative ratios of $O(1/h^2)$ for memory and $O(1/bs)$ for computation. As the model scales (i.e., larger $h$ and $L$), these ratios will remain stable or decrease quadratically, rendering $G_b$'s overhead negligible.
> - **Empirical evidence.** To validate this point, we conducted additional experiments. We trained models of different sizes on the Alpaca-GPT4 dataset with a batch size of 1, under both bf16 and fp32 precision settings. As shown in the table below, the computational and memory overhead of gradient norm is trivial. Storing the gradient norms of all modules requires less than 100 Bytes.
>
> |Model|precision|Avg. Step Time (ms)|Grad. Norm Time (ms)|Total Memory (GB)|Grad. Norm Memory (GB)|
> |-|-|-|-|-|-|
> |Qwen2.5-7B|bf16|37.40|0.99|15.68|$\le 10^{-7}$|
> |Qwen2.5-7B|fp32|126.25|1.96|35.40|$\le 10^{-7}$|
> |Qwen2.5-14B|bf16|78.09|1.89|34.08|$\le 10^{-7}$|
> |Qwen2.5-14B|fp32|260.50|4.15|65.98|$\le 10^{-7}$|
>
> **W3. Gradient norm as module importance.** Using the gradient norm to gauge module importance is not a heuristic; instead, it follows from a mathematical derivation.
> - According to the derivation in Line 130, the decrease in loss function can be lower bounded by
> $$\mathbb{E}[f(\theta^n) - f(\theta^{n+1})] \ge \sum_{b} p_b \\|g_b\\|^2.$$
> To maximize this loss decrease, we should maximize the right-hand side, $\sum_{b} p_b \\|g_b\\|^2$. As shown in Lines 132-134, this implies choosing the block with the largest squared gradient norm. Intuitively, **this strategy prioritizes blocks with larger gradient norms, as they contribute more significantly to optimization progress**.
>
> - To better evaluate the gradient norm in the presence of stochastic noise, this paper applies an **moving average** to the gradient norm (as illustrated in Eq. (4) of the manuscript). This effectively mitigates the impact of noise. Additionally, the  sampling probability in Eq.(4) employs a **softmax operation** on the importance metrics, further smoothing the noise and enhancing training stability.
>
> **W4. Module partition.**
> - **MISA is flexible.** We wish to clarify that MISA supports both coarser and finer-grained block partitioning. The core algorithm development—including the block coordinate descent update (Lines 124-130), importance sampling (Lines 131-141), practical implementation details (Lines 144-155, Algorithm 1), and convergence analysis—does not depend on a specific, concrete partitioning like $W_q, W_k, W_v$. This means our framework can operate effectively with any valid partitioning, unequivocally demonstrating that it is not hardcoded.
>
> - **Justification for the chosen partition.** The partitioning into units like $W_q,W_k,W_v$​ serves as a natural illustrative example in LLMs for two key reasons: (1)These are distinct matrix parameters within a transformer layer. They are inherently associated with weight gradients that participate in forward/backward passes, making them straightforward to compute and manage. (2) As shown in Figure 1, these modules exhibit significant heterogeneity in their gradient norms, which is crucial for the effectiveness of our importance sampling strategy.
>
> - **Partition ablation.** We do not claim our specific partition is optimal. It is possible that better partitions might exist, and to explore this, we conducted an ablation study on partitioning granularity. In the coarse-grained setting, each Transformer layer is divided into the Attention block and the MLP block. In the fine-grained setting, each matrix within the Transformer layer is partitioned into to the size of attention heads($128\times 4096$). Compared to the module-wise granularity, the improvement brought by finer-grained strategies becomes marginal.
>
> |Granularity|Math|Commonsense|
> |-|-|-|
> |Module-wise(MISA)|**73.6**|86.6|
> |Attention and MLP(coarser-grained)|72.5|85.4|
> |Head-wise(finer-grained)|73.1|**86.8**|
>
> **Q1. Module partition.** While we do not claim our specific partition is universally optimal, our primary goal is to demonstrate that this fine-grained module-wise approach is a significant and superior advancement over the layer-wise partitioning used in prior work. Proposition 2 provides theoretical evidence for this superiority , and our strong empirical results confirm it in practice. Please refer to our response to **W4** for more detailed discussion and ablation study.
>
> **Q2 and Q3. Importance score update frequency.**  According to Line 14 in Algorithm 1, the importance score is updated every $T$ inner iterations. We have conducted experiments analyzing how the importance scores change in training. We observed that the scores of several most important modules become stable early in training, while the later modules require a period of training to stabilize. In the later stages of training, the relative importance of modules changes very little. We also observe that when $T$ is too large, the convergence speed in the early training phase slows down. Therefore, it is beneficial to update the module importance scores more frequently at the beginning of training. In practice, however, the overhead of computing importance scores is very low (see **W2**), so more frequent updates of sampling probabilities do not introduce significant overhead. We will include more discussion on this in the revision.
>
> **Q4. Sensitivity to gradient norm.**  We thank the reviewer for this insightful question.
> - **Ablation study on importance metrics.** We investigated the sensitivity of MISA to the choice of importance metric through the following ablation study on LLaMA3-8B. Our results consistently show that the gradient norm yields the best performance when compared to the weight norm and parameter count, thus validating it as a robust choice.
>
>  |Method|MMLU|MMLU-pro|Commonsense|Math|
>  |-|-|-|-|-|
>  |Weight Norm|64.5|35.9|85.7|71.9|
>  |Number of Parameters|63.7|36.2|85.9|72.7|
>  |**MISA(Gradient Norm)**|**65**|**36.5**|**86.6**|**73.6**|
>
> - **Mitigating over-sampling of high-variance modules.** We acknowledge the potential issue of over-sampling modules that exhibit high gradient variance but low long-term utility. **MISA is explicitly designed to counteract this** through its sampling strategy. Our objective function, defined in Eq. (2) of the manuscript, incorporates a KL divergence penalty that regularizes the sampling probabilities. This term **prevents the sampling distribution from deviating excessively from a uniform distribution**, effectively balancing exploitation (prioritizing modules with high gradient norms) and exploration (ensuring all modules have a non-trivial probability of being sampled).
>
> **L1. Test on larger models.** In Section 5 of the manuscript, we have already acknowledged the limitations due to resource constraints. While not at the 7B+ scale, our pre-training results (Figure 4 and Table 6) on 130M and 350M models provide valuable proof-of-concept. Furthermore, we include an additional pre-training experiment on a 1B model to further demonstrate MISA's utility. As demonstrated in the table below, MISA outperforms standard Adam optimization in terms of perplexity in 1B model training. Most importantly, our work's primary focus is on memory-efficient fine-tuning, where MISA's effectiveness and scalability have been extensively validated on large models such as LLaMA3-8B, Qwen2.5-7B, Mistral-7B, and LLaMA3-70B.
> |LLaMA 1B|Adam|GaLore(r=512)|MISA($\delta = 25\\%$)
> |-|-|-|-|
> |Validation perplexity($\downarrow$)|15.56|15.64|**15.27**|
>
> **L2. Module partition granularity.** See the granularity ablation study in our response to **W4** and **Q1**.
>
> **L3. Detailed profiling.** See our response to **W1**.
>
> ---
> We sincerely thank the reviewer once again for their thoughtful comments. We hope the above responses adequately address their concerns and are happy to provide further clarification if needed.

---

> > ### Comment · Reviewer_ULBr · 2025-08-07
> >
> > Thank you for the comprehensive responses. The detailed profiling, theoretical justification for the importance metric, and ablation studies on partitioning granularity have addressed my major concerns. I will accordingly increased my score.

---

> > > ### Author Response · Authors · 2025-08-07
> > >
> > > Thank you for your positive feedback and increased score! Your comments significantly helped improve our manuscript, and we appreciate your recognition of our work. We pledge that these revised contents will be added to the new version of MISA.

---

> > > ### Author Response · Authors · 2025-08-09
> > >
> > > Dear Reviewer ULBr,
> > >
> > > We are pleased to have addressed your concerns. The detailed profiling, theoretical justification for the importance metric, and ablation studies on partitioning granularity will all be incorporated into the main text of the revised version. Thank you very much for your valuable feedback and support.
> > >
> > > Best regards,
> > >
> > > The authors of Paper 15661

---

> ### Author Response · Authors · 2025-08-06
> **Seeking Your Valuable Insights: A Follow - up on Paper 15661's Rebuttal**
>
> Dear reviewer ULBr，
>
> I hope this email finds you well.​
>
> First and foremost, I want to express my sincere gratitude once again for the time and effort you've already invested in reviewing our paper (ID: 15661). Your initial feedback was incredibly insightful, and I truly appreciate your commitment to the review process.​
>
> I understand that you must be extremely busy with multiple responsibilities, but I wanted to gently follow up regarding our rebuttal. We submitted it within the given timeframe, and since we haven't received any further comments or indications of remaining concerns from you, I'm writing to kindly inquire if you've had the opportunity to review it. Your perspective is of utmost importance to us, as it will help us refine our work and ensure it meets the high standards of this conference.​
>
> If there's anything we can do to assist you in reviewing our response more efficiently, please don't hesitate to let me know. I would be truly grateful if you could spare some time to share your thoughts with us at your earliest convenience.​
> Thank you again for your dedication to academic excellence and for your invaluable contributions to this review process. I look forward to hearing from you soon.
>
> Best regards,​
>
> The authors of paper 15661

---

### Note · Authors · 2025-08-13

We sincerely thank the area chair and all reviewers for their efforts in reviewing our submission. Their constructive feedback has greatly strengthened the quality of our work. Below, we provide a summary of our work and our rebuttal to help the decision-making process.
### 1. Summary of Our Work
We introduce Module-wise Importance Sampling (MISA), an optimized method for large language models. By dividing layers into modules and sampling based on importance scores, MISA cuts gradient variance, reaches an $O(1/\sqrt{K})$ convergence rate in non-convex stochastic scenarios, and boosts memory efficiency. Experiments across tasks prove its superiority over existing methods.
### 2. Strengths Recognized by Reviewers
In their comments, the reviewers recognized the following strengths of our work:
- **Well-motivated module-wise partition:** ULBr and CWR5 lauded MISA's fine-grained partitioning for enhanced memory control and gradient preservation.
- **Innovative sampling probability:** CWR5 highlighted the innovative design of sampling probabilities tied to gradient norms.
- **Strong theoretical guarentees:** ULBr commended the convergence guarantee with stochastic gradients and Adam optimizer; CWR5 emphasized deriving sampling from gradient norms and rates as a key technical edge.
- **Clear presentation:** ULBr, HSph, and ov6T praised the paper's lucid motivation, logical flow, and detailed analysis, with HSph and ov6T noting comprehensive experiments and ablation studies.
### 3. All Reviewer's Concerns Have Been Addressed
Our detailed rebuttals and extra experiments addressed all reviewer concerns:

- Reviewer ULBr (3→intended increase): "The detailed profiling, theoretical justification for the importance metric, and ablation studies on partitioning granularity have **addressed my major concerns**. **I will accordingly increased my score**".

- Reviewer CWR5 (3→intended increase): "The authors have satisfactorily addressed my concerns, both theoretical and experimental. **I will accordingly increase my score**"".

- Reviewer HSph (6, unchanged): "Clarifications accepted; **score remains positive**."

- Reviewer ov6T (4, unchanged): "Thorough reply; **score maintained**."

From their final comments, we find that all reviewers’ concerns have been fully addressed. While ULBr is yet to update the score, the intention to increase it is clear.

We trust this summary aids the AC in finalizing the decision and thank the AC for their time and consideration.

---

### Decision · Program_Chairs · 2025-09-17

**Decision:**

Accept (poster)

**Comment:**

This paper proposes MISA (Module-wise Importance Sampling Algorithm), a novel memory-efficient optimization method for full-parameter fine-tuning and pre-training of LLMs. Unlike prior layer-wise approaches that freeze entire transformer blocks, MISA decomposes each layer into smaller modules (e.g., attention heads, feed-forward components) and assigns gradient-based importance scores. A weighted sampling strategy then activates modules for updates, reducing gradient variance and yielding greater memory savings than layer-wise sampling. The method is theoretically grounded with convergence guarantees under non-convex stochastic settings and is validated by strong empirical results across commonsense reasoning, mathematical problem solving, instruction tuning, and pre-training tasks.

The reviewers recognize several key strengths of this work. MISA introduces a well-motivated module-wise partitioning of transformer layers, allowing fine-grained memory control and improved gradient preservation. Its importance-based sampling strategy, tied to gradient norms, is innovative and effectively reduces gradient variance. The method is theoretically grounded, with convergence guarantees under non-convex stochastic optimization, including with adaptive optimizers like Adam. Reviewers also praised the clarity of presentation, including logical flow, detailed analysis, and comprehensive experiments and ablation studies across tasks. Overall, the work is seen as methodologically novel, theoretically sound, and empirically strong, with all reviewer concerns addressed in the rebuttal and supplementary experiments. So we accept this work.